
# An overview of the ORACLES (ObseRvations of Aerosols above CLouds and their intEractionS) project: aerosol-cloud-radiation interactions in the Southeast Atlantic basin

Jens Redemann[1], Robert Wood[2], Paquita Zuidema[3], Sarah J. Doherty[2], Bernadette Luna[4], Samuel E. LeBlanc[5,4], Michael S. Diamond[2], Yohei Shinozuka[7], Ian Y. Chang[1], Rei Ueyama[4], Leonhard Pfister[4], Jun-me Ryoo[4,35], Amie N. Dobracki[3], Arlindo M. da Silva[6], Karla M. Longo[6,7], Meloë S. Kacenelenbogen[4], Connor J. Flynn[1], Kristina Pistone[5,4], Nichola M. Knox[8], Stuart J. Piketh[9], James M. Haywood[10], Paola Formenti[11], Marc Mallet[12], Philip Stier[13], Andrew S. Ackerman[14],

Susanne E. Bauer[14], Ann M. Fridlind[14], Gregory R. Carmichael[15], Pablo E. Saide[16], Gonzalo A. Ferrada[15], Steven G. Howell[17], Steffen Freitag[17], Brian Cairns[14], Brent N. Holben[6], Kirk D. Knobelspiesse[6], Simone Tanelli[18], Tristan S. L'Ecuyer[19], Andrew M. Dzambo[19], Ousmane O. Sy[20], Greg M. McFarquhar[1], Michael R. Poellot[20], Siddhant Gupta[1], Joseph R. O'Brien[20], Anastasios Nenes[21,36,37], Mary Kacarab[21], Jenny P. S. Wong[22], Jennifer D. Small-Griswold[23], Kenneth L.

Thornhill[24,33], David Noone[25,34], James R. Podolske[4], K. Sebastian Schmidt[26], Peter Pilewskie[26], Hong Chen[26], Sabrina P. Cochrane[26], Arthur J. Sedlacek[27], Timothy J. Lang[28], Eric Stith[29], Michal Segal-Rozenhaimer[4,5,30], Richard A. Ferrare[24], Sharon P. Burton[24], Chris A. Hostetler[24], David J. Diner[18], Steven E. Platnick[6], Jeffrey S. Myers[31], Kerry G. Meyer[6], Douglas A. Spangenberg[33], Hal Maring[32], and Lan Gao[1].

[1]University of Oklahoma, Norman, OK, USA

[2]University of Washington, Seattle, WA, USA

[3]University of Miami, Miami, FL, USA

[4]NASA Ames Research Center, Moffett Field, CA, USA

[5]Bay Area Environmental Research Institute, Moffett Field, CA, USA

[6]NASA Goddard Space Flight Center, Greenbelt, MD, USA

[7]Universities Space Research Association, Columbia, MD, USA

[8]Department of Geo-Spatial Sciences and Technology, Namibia University of Science and Technology, Windhoek, Namibia

[9]North-West University, Unit for Environmental Science and Management, Potchefstroom, North-West, South Africa

[10]College of Engineering, Maths and Physical Science, University of Exeter, UK

[11]Laboratoire Interuniversitaire des Systèmes Atmosphériques (LISA, UMR 7583), Creteil, France

[12]Centre National de Recherches Météorologiques, Météo-France-CNRS, Toulouse, France

[13]Atmospheric, Oceanic and Planetary Physics, Department of Physics, University of Oxford, Oxford, UK

[14]NASA Goddard Institute for Space Studies, New York, NY, USA

[15]Center for Global and Regional Environmental Research, University of Iowa, Iowa City, IA, USA

[16]Department of Atmospheric and Oceanic Sciences, and Institute of the Environment and Sustainability, University of California, Los Angeles, CA, USA

[17]Department of Oceanography, University of Hawaii at Manoa, Honolulu, HI, USA



[18]Jet Propulsion Laboratory, California Institute of Technology, Pasadena, CA, USA

[19]Department of Atmospheric and Oceanic Sciences, University of Wisconsin-Madison, Madison, WI, USA

[20]University of North Dakota, Grand Forks, ND, USA

[21]Georgia Institute of Technology, Atlanta, GA, USA

[22]Department of Chemistry and Biochemistry, Mount Allison University, Sackville, Canada

[23]Department of Atmospheric Sciences, University of Hawaii at Manoa, Honolulu, HI, USA

[24]NASA Langley Research Center, Hampton, VA, USA

[25]College of Earth, Ocean, and Atmospheric Sciences, Oregon State University, Corvalis, OR, USA

[26]University of Colorado, Boulder, CO, USA

[27]Brookhaven National Laboratory, Upton, NY

[28]NASA Marshall Space Flight Center, Huntsville, AL, USA

[29]National Suborbital Research Center, Moffett Field, CA, USA

[30]Department of Geophysics and Planetary Sciences, Porter School of the Environment and Earth Sciences, Tel Aviv University, Tel Aviv, Israel

[31]UC Santa Cruz, Santa Cruz, CA, USA

[32]NASA Headquarters, Washington, DC, USA

[33]Science Systems & Applications, Inc., Hampton, VA, USA

[34]Department of Physics, University of Auckland, Auckland, New Zealand

[35]Science and Technology Corporation, Moffett Field, CA, USA

[36]Ecole Polytechnique Federale de Lausanne, Switzerland

[37]Foundation for Research and Technology Hellas, Greece

**Abstract.** Southern Africa produces almost a third of the Earth's biomass burning (BB) aerosol particles, yet the fate
of these particles and their influence on regional and global climate is poorly understood. ORACLES (ObseRvations of Aerosols above CLouds and their intEractionS) is a five-year NASA EVS-2 (Earth Venture Suborbital-2) investigation with three Intensive Observation Periods designed to study key atmospheric processes that determine the climate impacts of these aerosols. During the Southern Hemisphere winter and spring (June-October), aerosol particles reaching 3-5 km in altitude are transported westward over the South-East Atlantic, where they interact with
one of the largest subtropical stratocumulus subtropical stratocumulus (Sc) cloud decks in the world. The representation of these interactions in climate models remains highly uncertain in part due to a scarcity of observational constraints on aerosol and cloud properties, and due to the parameterized treatment of physical processes. Three ORACLES deployments by the NASA P-3 aircraft in September 2016, August 2017 and October 2018 (totaling ~350 science flight hours), augmented by the deployment of the NASA ER-2 aircraft for remote sensing in September 2016
(totaling ~100 science flight hours), were intended to help fill this observational gap. ORACLES focuses on three fundamental science questions centered on the climate effects of African BB aerosols: (a) direct aerosol radiative effects; (b) effects of aerosol absorption on atmospheric circulation and clouds; (c) aerosol-cloud microphysical interactions. This paper summarizes the ORACLES science objectives, describes the project implementation, provides



an overview of the flights and measurements in each deployment, and highlights the integrative modeling efforts from cloud to global scales to address science objectives. Significant new findings on the vertical structure of BB aerosol physical and chemical properties, chemical aging, cloud condensation nuclei, rain and precipitation statistics, and aerosol indirect effects are emphasized, but their detailed descriptions are the subject of separate publications. The main purpose of this paper is to familiarize the broader scientific community with the ORACLES project and the data set it produced.

## 2. Introduction

The radiative and cloud-altering impacts of anthropogenic aerosol particles constitute the largest source of uncertainty in anthropogenic climate forcing (IPCC 2013). Aerosol particles interact directly with solar radiation through scattering and absorption of radiation, which leads to a direct radiative forcing whose sign depends on the ratio of the absorption to the total aerosol extinction and on the albedo of the underlying surface-atmosphere system (Coakley and Chylek, 1975). Heating of the atmosphere by aerosol absorption can induce changes in atmospheric circulation and mixing that can either enhance or decrease cloudiness (Ackerman et al., 2000; Koch and Del Genio, 2010). Aerosol particles also serve as cloud condensation nuclei (CCN), which can enhance cloud albedo by increasing the concentration of cloud droplets and reducing their size when aerosol concentrations increase (Twomey, 1974) and also potentially drive changes in cloud condensate or cloud cover by altering cloud lifetimes (Simpson and Wiggert 1969, Albrecht, 1989, Ackerman et al. 2004, Wood 2007). The magnitudes of all effects and even the sign of the latter two aerosol effects are not well quantified globally (IPCC 2013) and are expected to be geographically and seasonally heterogeneous because the total aerosol forcing is dependent upon the nature and amount of different aerosol species, cloud type and cover, and surface albedo, all of which vary on such scales.

Biomass burning (BB) is one of the largest sources of absorbing aerosol globally (Bond et al., 2013). BB aerosol particles contain black carbon, the most strongly absorbing of all aerosol constituents found in the atmosphere. The sign of the direct aerosol forcing is highly dependent upon the relative vertical locations of aerosol and clouds, with forcing changing from negative to positive when BB aerosol layers overlie low clouds rather than a dark ocean surface (Chand et al., 2009). BB aerosol also contains oxidized organic carbon and other soluble inorganic species that can act as effective CCN if transported into clouds.

The South East Atlantic (SEA) region has some of the highest optical depths of BB aerosol on the planet. It is also the location of large inter-model differences in aerosol forcing assessments (Schulz et al. 2006, Stier et al. 2013, Zuidema et al., 2016). The neighboring Southern African biomass burning (BB) source regions account for almost one third of the Earth's BB emissions (550 TgC yr[-1]; van der Werf et al., 2010) producing optically thick BB aerosol layers that are routinely transported across much of the South Atlantic basin (Chand et al., 2009; Zuidema et al., 2016). While burned areas are decreasing in size globally, burned areas in Africa are increasing, raising interesting questions about BB aerosol interactions with climate in that region in the future (Andela et al., 2017).

The SE Atlantic is also home to one of the Earth's largest subtropical stratocumulus (Sc) cloud decks, which plays a key role in the energetic balance of the region. The physical processes governing the feedbacks between sea surface temperature (SST) and cloud properties in these Sc decks are poorly represented in climate models (Bony and



Dufresne, 2005). In the Austral spring (July to October), the Sc deck interacts with the African BB aerosols that have been transported westward by prevailing mid-tropospheric tropical easterly winds. These aspects of the SE Atlantic attracted several international field experiments on aerosol-cloud-climate interactions in the region. These projects were based out of deployment sites distributed throughout the SE Atlantic (Fig. 1), and were scheduled between 2016 and 2018 to allow for collaborative science. These experiments include the NASA ORACLES project described in

this paper, deploying from Walvis Bay, Namibia in 2016 and São Tomé and Príncipe in 2017 and 2018, as well as the UK CLARIFY (Clouds and Aerosol Radiative Impacts and Forcing), deploying from Ascension Island in 2017, the French AEROCLO-sA (Aerosol Radiation and Clouds in southern Africa) project deploying from Walvis Bay, Namibia in 2017, and the DOE Atmospheric Radiation Measurement Mobile Facility LASIC (Layered Atlantic Smoke Interactions with Clouds) deployment to Ascension Island in 2016-2017, all described in more detail in Sect. 3.4.


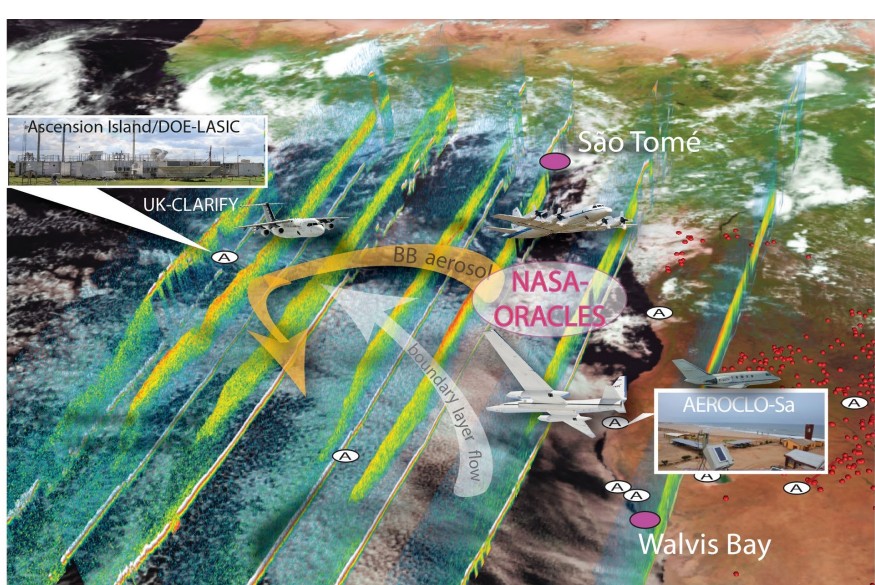

**Figure 1.** Deployment sites for the 2016-2018 ORACLES field experiments and collaborative international deployment activities (see text). The ovals with the letter A indicate new or refurbished AERONET sites.

The Southern and Central African fires producing BB aerosol occur during the warm, dry season over the continent, so emissions are lofted in the convective boundary layer to an altitude of several kilometers. As they advect offshore, the BB aerosol layers form a plume that initially overlays the cloud deck over the Atlantic (Fig. 2; see also Adebiyi et al. 2016, Zuidema et al. 2016, Deaconu et al., 2019), and exerts a direct radiative forcing (RF) whose sign and magnitude depend upon the reflectance and coverage of the clouds below and on the absorptivity of the aerosols (Kiel

and Haywood, 2003; Chand et al., 2009). Depending on the relative vertical location of the aerosols and the cloud deck, cloud condensate may increase or decrease in response to aerosol absorption and subsequent changes in atmospheric stability, relative humidity, and subsidence (semi-direct forcing). Cloud optical thickness and areal





coverage may also be influenced by aerosol-induced changes in cloud microphysics (forcing from aerosol-cloud interactions) when BB aerosols are mixed into the marine boundary layer (MBL). This is expected to occur more

frequently offshore as the MBL deepens in response to warming sea surface temperatures (e.g. Eastman et al., 2017), raising cloud top heights (Zuidema et al., 2009) and easing entrainment of the overlying aerosol, and as BB aerosol layers descend in response to prevailing large scale subsidence (Fig. 2).

Satellite- and model-based assessments of aerosol-cloud-climate interactions in this region (e.g., Chand et al., 2009; Wilcox 2012; Stier et al. 2013; De Graaf et al., 2014; Zhang et al., 2016; Adebiyi and Zuidema, 2018; Kacenelenbogen

et al., 2019; Sayer et al., 2019) indicate that improved observations of aerosol properties and loading, and cloud fraction, albedo, and liquid water path (LWP) are needed to constrain the local aerosol radiative impacts. Such studies are hampered by problematic aerosol retrievals in regions of extensive low clouds and difficulties retrieving cloud microphysical properties underneath dense aerosol layers (Haywood et al., 2004; Coddington et al., 2010; Deaconu et al., 2017). The observations used in these studies often have severe limitations and require significant assumptions

about aerosol and cloud properties (Yu et al., 2012, Yu and Zhang, 2013, Jethva et al., 2014; Knobelspiesse et al., 2015, Meyer et al., 2015, Sayer et al., 2016).

An example of a satellite-based retrieval of both aerosols and clouds is given in Fig. 2, which shows the altitude of aerosol and cloud layers during three months as a function of longitude, as operationally retrieved from the Cloud-Aerosol Lidar with Orthogonal Polarization (CALIOP) space-based lidar. Multiple filters have been applied for quality

assurance of these data. However, the simple message this figure conveys, of an elevated aerosol layer that is typically far above the low cloud, is somewhat misleading because (i) there could be multiple aerosol layers above the uppermost cloud and (ii) the CALIOP-derived aerosol layer base height has been found to be biased high, based on airborne measurements made during ORACLES as well as CALIOP retrievals that have been constrained by above-cloud aerosol optical thickness derived from the CALIOP data. The latter results from the fact that, especially for the

day-time retrievals from CALIOP, there is a significant reduction in signal-to-noise in the presence of optically thick aerosol layers. Hence, the separation between clouds and overlying aerosols (Fig. 2, yellow bars) is also likely biased high (see also Rajapakshe et al., 2017). Such observational uncertainty and the differing conclusions one may draw based on the separation between the BB aerosol layer and the underlying Sc clouds in this region, were a significant contributing impetus for the ORACLES project. We include the CALIOP-derived Fig. 2 here to provide the scientific

information available at the ORACLES proposal stage, which partially motivated the project in the first place and greatly influenced its design.

Surface-based measurements also have limitations. AERONET (Aerosol Robotic Network) sky radiance observations (Holben et al., 1998) are used frequently to tune global model estimates of aerosol absorption (Bond et al., 2013), but

can be routinely performed only from land and in the absence of clouds. Although historically a number of AERONET stations existed near the main African BB sources, just prior to ORACLES, there were no operational AERONET stations in the main BB region, with the exception of Ascension Island far downwind.

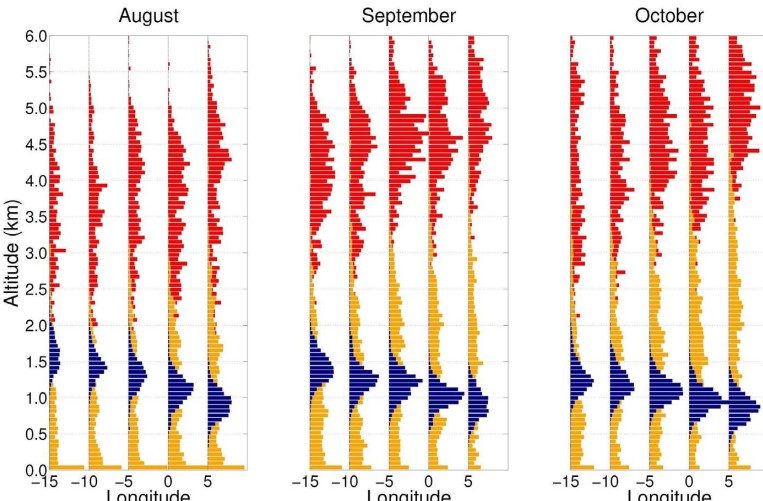

**Figure 2.** Distributions of aerosol top height (red), cloud top height (blue) and the separation between clouds and overlying aerosols (yellow) as a function of longitude, between 10-22.5° S. Observations are taken from the Cloud-Aerosol Lidar with Orthogonal Polarization (CALIOP) Version 3 aerosol profile product from 2006 to 2012 (7 years). See text.

Airborne instruments provide measurements of aerosols and clouds under co-varying meteorological conditions that are currently impossible to obtain from space. High resolution airborne observations, over scales that resolve processes of interest, provide critical constraints for parameterizing aerosol-cloud-climate interactions in models. They can also be used to enhance satellite-based remote sensing, by resolving in situ characteristics and variability within a particular scene, by providing a direct test of retrieved properties, and in the long term by guiding the development of new and improved remote sensing techniques. Because previous efforts to study BB emissions in South Africa (e.g. SAFARI-2K [Swap et al., 2003], TRACE-A [Fishman et al., 1996]) were focused over land or in close proximity to the coastal zone (e.g., Haywood et al., 2003), prior to ORACLES, there was a dearth of measurements over the SE Atlantic Ocean, where the major radiative impacts of BB aerosols are taking place.

In response to the need for new measurement constraints, in 2014 NASA funded the ORACLES project as one of the Earth Venture Suborbital-2 investigations. The goal of ORACLES is to provide a process-level understanding of the role of aerosols in climate by providing observations of all relevant aerosol effects over the SE Atlantic, a region with some of the largest aerosol loadings on the planet that is readily accessible with airborne platforms. The overarching ORACLES science goals are:

1. Determine the impact of African BB aerosol on cloud properties and the radiation balance over the South Atlantic, using state of the art in situ and remote sensing instruments to generate data sets that can also be used to verify and refine current and future observation methods, including instrument concepts with potential for deployment to space.


2.  Acquire a process-level understanding of aerosol-cloud-radiation interactions and resulting cloud adjustments, that can be applied in global models.

In this paper, we provide an overview of all three ORACLES deployments, highlighting aerosol absorptive and cloud-nucleating properties, their vertical distribution relative to clouds, the locations and degree of aerosol mixing into clouds, and cloud changes in response to such mixing. We make an initial assessment of the differences and similarities of the BB plume and cloud properties as observed from the 2016 deployment site (Walvis Bay, Namibia) at the plume's southern edge, and from the 2017 and 2018 deployment site (São Tomé and Príncipe) near the plume's northern edge. We conclude with an outlook for the integrative work we envision to address the overarching science questions regarding aerosol-radiation-climate interactions in the SE Atlantic and how these suborbital observations will aid long-term modeling and satellite remote sensing efforts.

## 3. Project Background

### 3.1 Motivation for 3-year field deployment

Prior to the ORACLES implementation stage, an analysis of satellite data in the study area had revealed pronounced shifts in aerosol altitude, concentration, and optical properties through the July to October BB season. That combined body of work suggested that aerosol loadings peak in September (Fig. 3, see also Adebiyi et al., 2015), whereas single scattering albedo (SSA) increases over the season, reflecting either a change in BB aerosol composition (Eck et al., 2013) or the mix of aerosol types present (Bond et al., 2013). Another striking seasonal change is that, on average, the gap between cloud top and the aerosol layer increases dramatically (Fig. 2), primarily due to higher aerosol layers.

The closer vertical proximity of BB aerosol layers to clouds early in the season (shown in Fig. 2) suggested that studies of aerosol-cloud interactions would be most feasible then, while larger gaps later in the season would suggest weaker indirect effects. Observing and quantifying these seasonal changes and the changing importance of the aerosol semi-direct and indirect effects over the BB season required either an impractically extended deployment or separate deployments spread across the season. The ORACLES team decided on separate deployments in September 2016, August 2017, and October 2018, a decision that was aided by a relative lack of interannual variability in meteorology. This variability was predominantly linked to SST variations known as Benguela Niños that mainly occur in boreal spring, not fall, and are much less frequent than the better-known Pacific El Niños (Rouault, 2012). Interannual variability in fire emissions was expected to be low as well (van der Werf, 2010). As a result, aerosol loading in the ORACLES region was expected to be repeatable, with MODerate Resolution Imaging Spectroradiometer (MODIS) clear-sky Aerosol Optical Depth (AOD) retrievals implying year-to-year variability through the burning season of only 20% of the mean. In reality, recently developed above-cloud AOD retrievals reveal a significant interannual variability in the properties of the above-cloud aerosol plume (see Sect. 4.2-4.3 below); investigations into the particular reasons are ongoing. Finally, a practical consideration for the attempt to cover the BB seasonal cycle with three separate deployments was based on the fact that airborne instrument performance has a tendency to significantly decrease as mission durations extend beyond 4 weeks.

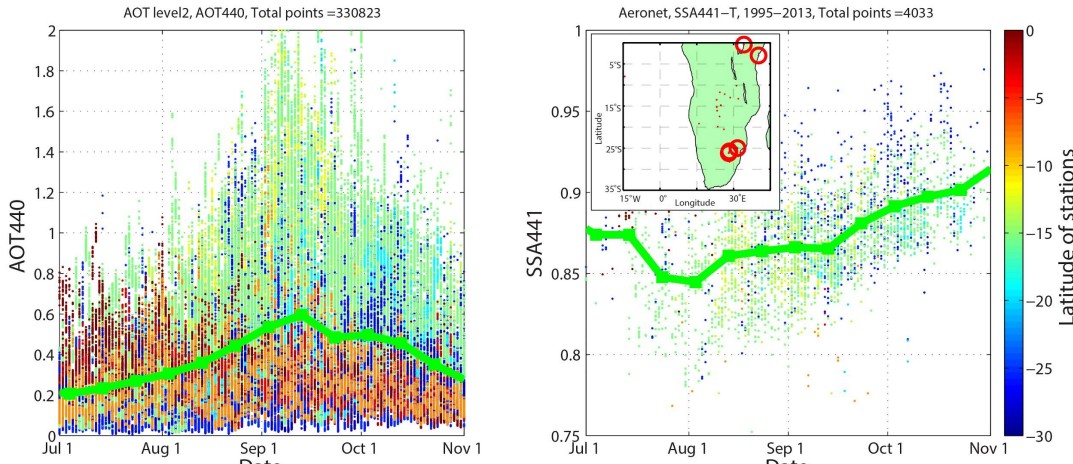

**Figure 3.** Aerosol optical depth (left) from AERONET sites in Southern Africa peaks in early September, while SSA
(right) shows a significant increase between August and November. Red circles in the inset indicate the few stations
operating between 2011 and 2013. Both panels contain data from 1995 to 2013 to represent the state of knowledge
prior to the ORACLES deployments.

### 3.2 Science questions and objectives

ORACLES science questions and related objectives are generally focused on direct, semi-direct, and indirect aerosol
effects on climate. Table 1 summarizes science questions and objectives as originally posed. The general approach for
developing these objectives was to include goals that were highly achievable first, and to increase the complexity of
objectives gradually. The objectives related to science questions 1 and 2, i.e., direct and semi-direct effects, constituted
the "threshold" (for success) science mission. The science objectives associated with question 3, i.e., indirect effect
assessments, were part of the "baseline" science mission, which in NASA terminology indicates the full mission scope.

**Table 1.** A summary of ORACLES science questions and related objectives

| Science Questions | Related science objectives |
|---|---|
| Q1: What is the direct radiative effect of the African biomass burning (BB) aerosol layer in clear and cloudy sky conditions over the SE Atlantic? | O1-1 (Aerosol spatial evolution): Determine the evolution of the BB aerosol microphysical and spectral radiative properties as the aerosol is transported across the South Atlantic. O1-2 (Aerosol-induced radiative fluxes): Measure aerosol-induced spectral radiative fluxes as a function of cloud albedo and aerosol properties. |





| | O1-3 (Seasonal aerosol variation): Assess the key factors that control the seasonal variation in aerosol direct effects. |
|---|---|
| Q2: How does absorption of solar radiation by African biomass burning (BB) aerosol change atmospheric stability, circulation, and ultimately cloud properties? | O2-1 (Relative vertical distribution): Determine the seasonally varying relative vertical distributions of aerosol and cloud properties as a function of distance from shore. <br> O2-2 (Aerosol-cloud heating rates): Constrain aerosol-induced heating rates for aerosol layers above, within and below cloud. <br> O2-3 (Cloud changes due to aerosol-induced heating): Investigate the sensitivity of cloud structure and condensate to aerosol-induced heating rates. |
| Q3: How do BB aerosols affect cloud droplet size distributions, precipitation and the persistence of clouds over the SE Atlantic? | O3-1 (Mixing survey): Survey the location and extent of aerosol mixing into the BL and its seasonal variation. <br> O3-2 (Cloud changes due to aerosol mixing): Measure changes in cloud microphysical properties, albedo and precipitation as a function of aerosol mixing into the BL. <br> O3-3 (Cloud changes due to aerosol-suppressed precipitation): Investigate the sensitivity of cloud structure and condensate to aerosol-induced suppression in precipitation. |

### 3.3 Project implementation

#### 3.3.1 Logistics and Deployment Details

250    The Walvis Bay airport in western Namibia was originally considered the ideal location for ORACLES due to its proximity to the ocean and cloud deck, runway length and hangar size for the ER-2, and due to its use during the SAFARI-2K campaign by the University of Washington CV-580 aircraft. The runway had been extended to ~3,350 m, but certification of the extension was still in progress as ORACLES began. It was our intention to deploy only one aircraft in the first year, and grow the activity after acclimating to the locality and airspace. However, during the

255    ORACLES-2016 deployment planning, the UK CLARIFY team announced plans to fly its airborne assets in 2016 as well. In response, leadership re-ordered ORACLES deployments to bring both aircraft to Walvis Bay in 2016, so as to maximize their impact in a concerted effort with the international partners.





In the event that the facilities were found to be inadequate or unready or unavailable, alternate airfields were investigated. Specifically, Upington (FAUP), South Africa and São Tomé (FPST) were pursued with due diligence until country approval was obtained from Namibia. Upington had no usable hangar, but an extremely long runway, little competing traffic, and the benefit of a longstanding collaborative relationship with the USA and NASA. São Tomé could not accommodate the ER-2 requirements and its airport fell short of NASA standards, but officials were very enthusiastic about a NASA collaboration. Both locations could support a P-3-only deployment. A temporary hangar in Upington, RSA or Windhoek, Namibia might have supported the ER-2. Ascension Island also had no hangar and posed significant constraints to the commercial import/transport of people and equipment. In later years, Ascension was subject to runway construction, but it did serve as an overnight transit stop for the P-3 in 2016 and 2017, and as the target for suitcase (overnight stay) flights in 2017.

ORACLES experiment requirements dictated deployment of up to 80 people (110 in 2016) for three 5-week periods, centered on the months of September 2016, August 2017, and October 2018.

**3.3.2 Choice of Measurement Platforms:  Envisioned versus Realized Capabilities**

The revised ORACLES project implementation plan called for the operation of two aircraft in 2016 and the operation of only one aircraft in 2017 and 2018. The choice to deploy the ER-2 aircraft in one year only was solely based on funding considerations, as its operations are considerably more complicated and costly.

The ORACLES platforms and instruments were selected to efficiently and quantitatively address the science questions outlined above through measurements of radiative fluxes, derivation of heating rates, and observations of aerosol and cloud microphysical and radiative properties, atmospheric thermodynamics and chemistry. The desired flight plans were driven by expected aerosol-cloud features and their interactions within the region, by regional model forecasts and by recent (same-flight or previous-flight) observations. When the P-3 was the sole NASA aircraft deployed (August 2017 and October 2018), Research Scanning Polarimeter (RSP) and High Spectral Resolution Lidar (HSRL-2) instruments (Table A4) were added to its payload to capture relevant science data by flying above, within and below aerosol layers and clouds. When both the P-3 and ER-2 were present (September 2016), the ER-2 served in a remote sensing role, obviating the need for the P-3 to fly above both cloud and aerosol layers.

It was known from the planning stage that the range of the P-3 aircraft was dependent on payload and flight pattern flown; spirals and low altitude flight reduce flight time. Based on DISCOVER-AQ's successful inclusion of spirals from 1000 to 5000 ft during 8-hour flights, ORACLES flights were expected to be similar in duration and character. From Walvis Bay, Namibia, this covered the target science zone (5° S to 35° S, westward from the coast to approximately the prime meridian), transit to Ascension Island for suitcase flights and for coordination with the UK's CLARIFY team, and transit to São Tomé Island, located near the northern edge of the climatological BB plume, with margins for headwinds, profiles, and low-altitude flight.

During the 2016 deployment, it became obvious that there was merit in P-3 flights that extended beyond the originally-estimated 8 hour duration. The NASA P-3 crew accommodated this request as far as crew rest considerations permitted; the average flight duration for local P-3 science flights (excluding transits) in 2016 was 8.3 hours (see Sect.





4.4). In 2017 and 2018, while flying from São Tomé, a larger P-3 flight crew was able to average 9.1 and 8.3 flight

hours respectively for similar flights, with the 2018 average affected by increased safety margins due to expected

inclement weather.

For ORACLES, ER-2 flights were envisioned to be up to 8 hours in duration, similar to the prior Studies of Emissions and Atmospheric Composition, Clouds and Climate Coupling by Regional Surveys (SEAC4RS) campaign. Due to the 2-hour pre-flight 'hands off' period and pilot 12-hour duty day, there were concerns that weather delays (e.g., low

ceilings) would result in flight duration limitations and therefore limitations in the final geographical coverage. The ER-2 deployed only once for ORACLES, to Walvis Bay, Namibia during September 2016. The payload consisted of Enhanced MODIS Airborne Simulator (eMAS), Airborne Multiangle SpectroPolarimetric Imager (AirMSPI), RSP, and HSRL-2 (Table A5) for various aspects of cloud composition, aerosol properties, and the overall cloud/aerosol morphology; and Solar Spectral Flux Radiometer (SSFR) for radiative flux measurements. The aircraft operated as

expected and the average flight duration from Walvis Bay was 8.1 hours. Especially during the second half of the campaign, the ER-2 pilots were extremely accommodating, frequently extending individual flight duration to 9 hours.

### 3.3.3 Choice of Instrumentation

In this section, we discuss the choice of instrumentation for each platform and deployment. Depending on ORACLES deployment year, the P-3 carried 8 to 11 instruments or instrument suites, with the following included for all

deployments: cloud suite (UND/OU), Phase Doppler Interferometer (PDI), Hawaii Group for Environmental Aerosol Research (HiGEAR) in situ measurement suite for aerosols, SSFR/CG-4 for radiative fluxes, 4STAR for aerosol optical depth, cloud and sky radiances, Airborne Third Generation Precipitation Radar (APR-3) for cloud and precipitation observations, Research Scanning Polarimeter (RSP), and CO Measurements and Analysis (COMA) for $CO$, $CO_2$ and $H_2O$ mixing ratios, CCN for cloud condensation nuclei and Water Isotope System for Precipitation and

Entrainment Research (WISPER) for water isotope measurements. In certain years there were targeted additions/deletions:

- The HSRL-2 was added to the P-3 payload for deployments without the ER-2, i.e., 2017 and 2018.
- Advanced Microwave Precipitation Radiometer (AMPR) was included in the first deployment year only (2016), when the ER-2 also carried the HSRL-2 and RSP.

- The PTI (Photothermal Interferometer) was included in the 2016 and 2018 deployments only, as it suffered a failure before the 2017 deployment.
- The Counterflow Virtual Impactor (CVI) was added for 2017 and 2018 as part of WISPER to enable cloud residual aerosol and droplet water isotope ratio measurements.
- An aerosol filter system (AFS) for trapping aerosol particles for post-flight analysis was added in 2017 and 2018.

- In 2017 and 2018, a duplicate Cloud Droplet Probe (CDP) was mounted in a position more forward relative to the leading edge of the wing, in an attempt to determine whether proximity to the leading edge affected cloud particle measurements.
- In 2018, two customized versions of the sunshine pyranometer (Badosa et al., 2014) were added to the P-3, as was a nadir-viewing geo-referenced and radiometrically calibrated fisheye camera.



As part of ORACLES, two new AERONET stations were established, i.e., the "Namibe" site in Namibe, Angola, and the "SEGC_Lope_Gabon" site near Libreville, Gabon. Many other sites in the region were revamped or established with separate funding and are shown in Figure 1 (see https://aeronet.gsfc.nasa.gov/cgi-bin/site_info_v3 for a list of sites).

Tables A4 to A6 in Appendix B provide full payload tables, including instrument names, instrument descriptions,
primary measurements, and derived geophysical observables for each instrument.

### 3.3.4 Experiment Strategy: Dry-run exercise

ORACLES conducted a two-week 'dry-run' activity from September 14-28, 2015, prior to its first deployment in 2016. Project meteorologists, platform scientists, pilots, and the leadership team met by phone and WebEx to examine daily weather forecasts, chemical weather predictions, and flight conditions. In response to these forecasts, detailed
flight plans were developed for the upcoming one or two possible flight days, using a flight planning tool that was specifically designed for multi-aircraft flight operations (LeBlanc, 2018). Longer-range forecasts up to 5 days were used to plan for extended flight strategies relative to overarching flight objectives. Satellite data, primarily Meteosat-10 Spinning Enhanced Visible and Infrared Imager (SEVIRI) visible imagery andCloud-Aerosol-Transport System (CATS) and CALIPSO vertical feature mask and attenuated backscatter profiles were used to evaluate the likely
success of a given flight plan for a given day. The process familiarized the science team with items that impacted real flight planning: aircraft limitations, staff fatigue limits (e.g., down days, crew rest), aviation authority coordination timelines, the availability and latency of meteorological forecasts, and the use of the flight planning tool. This practice time made ORACLES actual deployments more efficient, although the complexity and scope of actual chemical and meteorological forecasts in the field ended up being well beyond the scope of the dry-run exercise. This increased
complexity was undoubtedly the result of lessons learned during the dry-run exercise itself.

### 3.3.5 Experiment Strategy: Forecasting and Flight Planning

The forecasting effort for ORACLES deployments in the field entailed both meteorological and chemical weather predictions. The meteorological forecasting effort for the ORACLES mission consisted of three components: (i) forecasting for flight planning, (ii) nowcasting during flights for real-time flight direction, and (iii) forecasting local
weather for flight operations. Each of the three ORACLES deployments featured daily planning meetings. On non-flight days, the flight planning team met at 08:00 local time to discuss the weather and chemical weather forecasts for a period of up to five days, with special emphasis on the upcoming one or two flight days. On flight days, the team would assemble at 05:00 local time to assess whether the latest forecasts warranted any changes to flight plans made the day prior. Also on flight days, the forecast team would provide in-flight nowcasting that often led to significant
adjustments of flight plans, usually to respond to actual cloud conditions that materialized on a given flight day or in response to changing local conditions.

Clouds were the primary focus of the meteorological forecasting effort for both flight planning and nowcasting. Low clouds (i.e., stratocumulus at the inversion) were of primary scientific interest for their interaction with the African smoke plume. However, middle and high clouds were also important since the presence of these clouds complicated





the radiation measurements of some instruments (e.g., 4STAR, SSFR). Verification studies prior to the ORACLES
       deployments showed that the European Centre for Medium Range Weather Forecasts (ECMWF; Pappenberger et al.,
       2008; ECMWF Newsletter, 2012; Ye et al., 2014) and United Kingdom Meteorological Office (UKMO; Ran et al.,
       2018) global forecast models provided the best performance for cloud forecasts. ECMWF digital data were available
       at 0.125° longitude x latitude resolution, and included the primary meteorological variables (relative humidity and
horizontal winds at 925, 850, 800, 700, 600, 500, 400, 300, 150, and 100 hPa levels; 1000-500 hPa layer thickness;
       surface wind speed; mean sea level pressure; boundary layer height; precipitation; convective available potential
       energy) as well as 3-D ice and liquid water mass. We also used the 2-D ECMWF products of cloud fraction and cloud
       base for the low, middle, and high clouds, which were found to be adequate for our forecasting requirements. The
       ECMWF cloud forecasts were supplemented by low, middle, and high cloud distribution forecasts from the UKMO
global forecast model. In order to forecast the overall circulation over the southeast Atlantic, with an emphasis on
       wind and relative humidity distributions from the surface to 500 hPa, we used a suite of forecast products from
       ECMWF, UKMO, Global Forecasting System (GFS) from the National Centers for Environmental Prediction
       (Environmental Modeling Center, 2003), and the NASA Goddard Earth Observing System, Version 5 (GEOS-5)
       model (Molod et al., 2012).

Our primary nowcasting tool during the flights was the geostationary satellite imagery from the SEVIRI instrument
       aboard Meteosat-10 and Meteosat-11. Raw imagery from the infrared and visible channels was useful for establishing
       the evolution and distribution of clouds during flight. Satellite cloud properties described by Minnis et al., (2008,
       2020) were calculated from SEVIRI raw radiances by NASA Langley including cloud altitude, water path, and
       effective radius, and were also used in real-time flight direction. For forecasting local weather for flight operations,
particularly during the October 2018 deployment in São Tomé, we primarily relied on satellite imagery over the past
       12-24 hours for a short-term (i.e., a day or less) outlook of heavy precipitation at the airport, as precipitation forecasts
       from the models were largely unreliable.

       Chemical forecasts were done using both global and regional models, with the regional models providing a lot of the
       detail required for flight planning on a daily basis. We used three global systems: GEOS5
(https://gmao.gsfc.nasa.gov/GEOS/), the Copernicus Atmosphere Monitoring Service (CAMS,
       https://atmosphere.copernicus.eu/), and a bespoke 3-component aerosol (carbonaceous, mineral dust, and industrial
       pollution) modelling system developed by the UK Met Office for their CLARIFY deployment in 2017. Five-day
       aerosol forecasts provided the expected spatial and vertical location of the main smoke coming from the African
       continent. These models were also useful in identifying times when the smoke was expected to be mixed with dust
aerosols, especially during the 2016 deployment from Walvis Bay.

       For regional model forecasts, two configurations of the Weather Research and Forecasting (WRF) model (Skamarock
       et al., 2008) were employed. One of them used WRF coupled to chemistry (WRF-Chem, Grell et al., 2005) using the
       physics package from the Community Atmosphere Model version 5 (CAM5, Ma et al., 2014), run by a team from the
University of Iowa (WRF-CAM5, PI: G. Carmichael). This model provided daily 72-h aerosol forecasts for similar
       purposes as for the global models by using a full chemistry suite with hundreds of chemical species considered. WRF



was also configured using an aerosol aware microphysics (AAM) scheme (Saide et al., 2016) maintained by a team from NCAR/UCLA (PI: P.E. Saide). The Weather Research Forecasting Aerosol-Aware Microphysics (WRF-AAM) model provided forecasts for lead times of up to 4 days at 12 km resolution. The system included a near-real time

emission constraint using satellite-based aerosol optical depth (Saide et al. 2016), which to our knowledge corresponds to the first near-real time system to perform such tasks. Two simulations were performed per forecasting cycle turning smoke emissions on and off in the model. Since WRF-AAM resolves aerosol-cloud-radiation interactions, these simulations allowed assessment of the effects of smoke on weather in forecasting mode by taking the difference between the two forecasts. The forecasts also included tracers tagged to each day of smoke emissions from the African

continent, which were used to provide a distribution of smoke age based on the tracer concentrations and the days since emissions. Statistics such as mean and mode were extracted from the age distribution and used for flight planning to target plumes with different ages to explore the temporal evolution of aerosol properties.

Another task performed during the planning meetings was near-real time evaluation of the forecasts. These were focused on assessing forecast performance in predicting clouds and the aerosol plume location, and relied mostly on

the latest SEVIRI cloud retrievals, and clear-sky and above-cloud AOD from MODIS. This exercise allowed the team to track forecast failures and successes and provided a sense of reliability when making decisions based on forecasts.

### 3.3.6 Routine Flights vs Target of Opportunity Flights

The ORACLES investigation concept featured a combination of routine flights to facilitate comparisons with climate models and to ensure sampling of a wide range of aerosol loadings and cloud conditions, with other flights addressing

"targets of opportunity". The "routine" flights all took place along a fixed latitude/longitude line with sampling at a range of altitudes and remote sensing of the full column. In 2016, the routine flight track was along a diagonal with endpoints of 20° S/10° E and 10° S/0° E; in 2017 and 2018 the routine flight track extended from the equator to ~15° S along 5° E. See Fig. 11 in Sect. 4.4 for a complete set of flight tracks.

In situ observations of aerosol microphysical and optical properties during the routine flights were envisioned to map

the evolution of BB aerosol radiative properties during transport. HSRL-2 (High Spectral Resolution Lidar) observations from the P-3 or ER-2 helped map the spatial extent of the layers, while SSFR (Solar Spectral Flux Radiometer) and 4STAR (Spectrometer for Sky-Scanning, Sun-Tracking Atmospheric Research) observations provided additional insights into the in situ derived aerosol properties via optical and radiative closure experiments. Measurements to address seasonal variations in direct aerosol radiative effects and their controlling factors were

derived from the routine flights. The routine flight requirements were derived based on the assumption that the statistics of important observed aerosol and cloud properties, given sampling and measurement uncertainties, are sufficiently constrained to distinguish between climate model estimates. For this, we assumed that the variability in aerosol properties at model-relevant scales (100 km$^2$) can be extrapolated from the analysis of Shinozuka and Redemann [2011] to be less than 20%, and that such variability is well below the inter-model differences on such

scales.

Another motivation for the routine flights was to ensure sampling of a wide range of aerosol loadings and cloud conditions. The five to six envisioned routine flights (equaling ~40-50 flight hours per deployment) that comprise the





ORACLES threshold science objectives were intended to yield aerosol and cloud data in about 200 100-km$^2$ climate model grid boxes. Prior to the start of the campaign, we investigated PDFs of MODIS daily 1x1 deg averaged AOD

between 10°-20° S and 5° W-5° E for September 2001 derived from the then-available dark target algorithm. We randomly subsampled the roughly 3000 1x1 deg MODIS AOD boxes with the planned 200 airborne observations and found that the resulting PDFs were a good representation of the parent population of MODIS AOD. We concluded that the number of threshold science flight hours was adequate to compile probability density functions of aerosol properties that allow assessments of climate model differences at these spatial scales. This was confirmed by analysis

after the first deployment (Shinozuka et al., in prep.).

About half of the flight hours in each campaign focused on targets of opportunity, as detailed in Tables A1-A3 in Appendix A. These flights targeted specific science goals (e.g., capturing a range of aerosol ages, or contrasting conditions in terms of aerosol-cloud interactions). Flight patterns (e.g. "radiation walls", square spirals) were optimized to leverage the measurement capabilities of the range of instruments on board the P-3 and to allow for later

comparison of different methods of measuring a common parameter (e.g., aerosol SSA). During the 2017 and 2018 deployments, the target of opportunity flights were planned to be near the routine flight track whenever possible to improve sampling statistics.

### 3.3.7 Considerations for 2016 deployment with ER-2 and P-3

The planning for the 2016 field deployment in Namibia started in early 2015, with multiple site visits to the Walvis

Bay airport, logistics and hotel providers, Namibian science partners and representatives of various Namibian government organizations. This planning started early because the unexpected change to deploy both ORACLES aircraft in 2016 brought along a significant set of challenges due to the large contingency of scientific and aircraft support equipment needed, and this being the first of the three ORACLES deployments. The ORACLES team gratefully acknowledges the help provided by the Honorable Thomas F. Daughton, US Ambassador to Namibia from

2014 to 2017, and the support by the US embassy staff led by Mr. John Kowalski. The US embassy proved instrumental in receiving flight permissions and in arranging the student program in August 2016. The ORACLES team also received invaluable feedback and support from the Namibia University of Science and Technology (NUST), led by its rector, Dr. Tjama Tjivikua, and Dean Lameck Mwewa. In addition to NUST, the Gobabeb Training and Research Center led by Dr. Gillian Maggs-Kolling, the University of Namibia, represented by Dr. Martin Hipondoka

and Dr. Michael Backes, and North-West University (South Africa) represented by Dr. Stuart Piketh provided information and logistics support throughout the 2016 campaign. As we describe in Sect. 3.5 below, these contacts were the springboard for the outreach efforts that led to the deployment of 7 graduate students in the 2016 field campaign, including five students from NUST and the University of Namibia. In their totality, we hope that the efforts expended by the Namibian government and the Namibian and South African science community, as well as the

reciprocating efforts by the ORACLES science team, can be considered a transformational effort in the context of science diplomacy (Annegarn and Swap, 2012), at least in so far as the experience for the individual students that participated in the outreach efforts are concerned.



### 3.4 Linkage with international deployment efforts (LASIC, CLARIFY, AEROCLO-sA)

ORACLES was not the only recent experimental investigation in the southeast Atlantic (see Fig. 1). The UK CLARIFY project, which deployed their BAe-146 plane from Ascension Island in August-September 2017, and the French AEROCLO-sA project, which deployed a Falcon-20 plane from Walvis Bay, Namibia, in August 2017 (Formenti et al., 2019), shared similar science objectives with ORACLES, as did the DOE Atmospheric Radiation Measurement Mobile Facility deployment to Ascension Island from June 2016 through October 2017 for the LASIC project (Zuidema et al., 2015). All four campaigns were active in August-September 2017, and a 'suitcase' flight to Ascension by the NASA P-3 plane included a direct instrument inter-comparison flight with the CLARIFY Bae-146 on 18 August 2017 (Barrett et al., 2020). Collaboration between all four campaigns continued through a joint data workshop held in Paris in April 2019, prior to a joint session at the annual meeting of the European Geophysical Union. The excitement generated from sharing insights and points of view, some similar and some not, from the individual campaigns led to a decision to hold another joint workshop in May 2020, planned for the United States but held virtually due to the COVID-19 pandemic. In one example, a different view of the relationship of the single scattering albedo to aerosol aging was noted, with ORACLES scientists focusing more on the vertical structure (Fig. 12), and CLARIFY scientists interested in investigating the change in SSA with distance from the continent. The latter is an excellent example of the synergism afforded between the two campaigns, with ORACLES sampling air closer to the continent, and the CLARIFY campaign sampling ~1700 km offshore.

### 3.5 Outreach efforts

### 3.5.1 Namibia - 2016

During the field campaign held in Namibia in 2016, the gathering of science data not only benefited the scientists directly involved in the project; through an outreach program the science was extended to the Namibian population (and to some extent the broader southern African region). The outreach effort was multi-tiered and aimed to inform the public, develop young scientists, and encourage children to enter into STEM fields of study. Outreach activities included public lectures, interviews with local radio and newspapers, and open days at the airfield. In addition, ORACLES scientists traveled to northern Namibia for several days to participate in the Ongwediva Annual Trade Fair (OATF) together with students, staff, and faculty from the University of Namibia - UNAM, the Namibia University of Science and Technology - NUST, and the Gobabeb Research and Training Center. The OATF showcased collaborative environmental research from participating research institutes and was attended by Ongwediva-area students, business leaders, and local dignitaries.

In addition to these broader public engagement outreach activities, a targeted science development program was initiated with support from the US Embassy in Namibia and NUST. This 3-week full immersion outreach program was developed to provide promising local and regional young scientists with an opportunity to experience different components of a large complex airborne research field campaign. In total, 7 post-graduate students (Masters and PhD level), from Namibia (5 students) and South Africa (2 students), participated in the student guest program (Fig. 4).



Student guests were exposed to the planning, modelling, and instrumentation used within the ORACLES field campaign. In addition to these broad field campaign skills, they received a solid foundation in basic atmospheric

science through tutorials from the participating campaign scientists, some introductory programming tutorials, and an opportunity to interact with scientists aligned with their field of research. Within the duration of the program they also all had an opportunity to join a science flight. Further regional expansion of this student guest program was planned for the 2017 and 2018 field campaigns in collaboration with the CLARIFY and AEROCLO-sA campaigns, but with the move of the ORACLES field campaigns to São Tomé this expansion outreach effort could not be implemented.

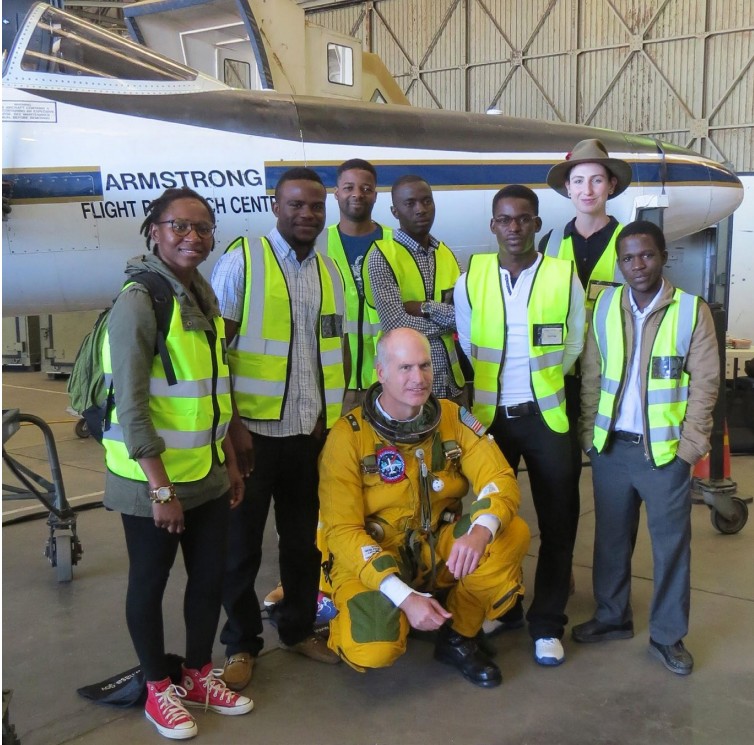


**Figure 4.** The ORACLES guest students, together with ER-2 pilot, James Gregory Nelson, at the Walvis Bay airport.

### 3.5.2 São Tomé and Príncipe (STP) - 2017 and 2018

To understand the challenges of implementing an outreach program as part of a scientific project like ORACLES in
São Tomé and Príncipe (STP), one needs to know a little about the history of this young country of just over 200,000 inhabitants. Previously uninhabited, the STP Archipelago was colonized by Portugal throughout the 16th century, when it served as a warehouse for the slave trade and established itself as a producer of sugar-cane, coffee, and cocoa. STP independence from Portugal came in 1975, keeping Portuguese as its official language, although minority groups also speak at least four other dialects.



During an initial exploratory visit in 2015, built on a previous NSF-sponsored site visit, the ORACLES team contacted the *Instituto Nacional de Meteorologia* (INM) to establish collaborations. The INM operates at STP airport facilities and showed great initial enthusiasm for ORACLES deployments from STP. During the two years of ORACLES operations in 2017 and 2018, INM kindly issued daily weather reports tailored to ORACLES needs. There were several visits from the ORACLES team to INM, and Mr. Aristómenes Amadeu do Nascimento of INM attended various

ORACLES weather briefings.

The only public University in STP and the most important one, the University of STP (USTP), was established only in 2014. The creation of USTP came to address fundamental and emergency problems of the Country, which included training of personnel for the health and education sectors, agriculture, and food production. When the ORACLES team deployed to STP in 2017, USTP had, in its current format, only two years of existence, still consolidating its

vocations and priorities. Nevertheless, the institution represented by Dr. Aires Bruzaca (Dean), Mr. João Pontífice (Vice Dean), and Prof. Manuel do Sacramento Ramos Penhor was enthusiastic about establishing scientific collaborations with NASA.

The ORACLES team organized a series of seminars about ORACLES scientific objectives for the USTP and INM communities. The seminar themes also included the AERONET (AErosol RObotic NETwork) and Pandora NASA

projects, global networks of spectrometers designed to retrieve, respectively, aerosol optical depth and microphysical parameters (Holben et al., 1998, Dubovik and King, 2000), and total columns of ozone and other trace gases in the atmosphere from direct-Sun measurements (Herman et al., 2009, 2015; Tzortziou et al., 2012). All lectures were presented by ORACLES science team members in Portuguese to address potential language barriers.

A Pandora Spectrometer Instrument (PSI) and an AERONET instrument were brought to STP as part of the

ORACLES deployment. The main goal, especially for the PSI deployment, was to assess whether mutual good-will, interest, and capabilities exist for NASA, USTP, and INM to collaborate scientifically long-term. The team was successful in training professors of the USTP to operate the PSI and the AERONET instruments (Fig. 5), and this resulted in additional aerosol measurements beyond the campaign periods. Moreover, it laid the foundation to have STP as one of the sites of the Pandora network, with an official agreement between NASA and the USTP signed in

550 2018.

The ORACLES team found in STP a community open and eager to the establishment of a fruitful scientific cooperation. Our experience points out that involvement with the local community is of extreme importance, not only for the dissemination of scientific knowledge but also to facilitate engagement between the young scientists from both communities. Collaboration with local scientific communities during field deployments such as ORACLES have the

dual benefits of enhancing local scientific capabilities in under-resourced areas of the world and producing tangible benefits for this and future missions in the region.



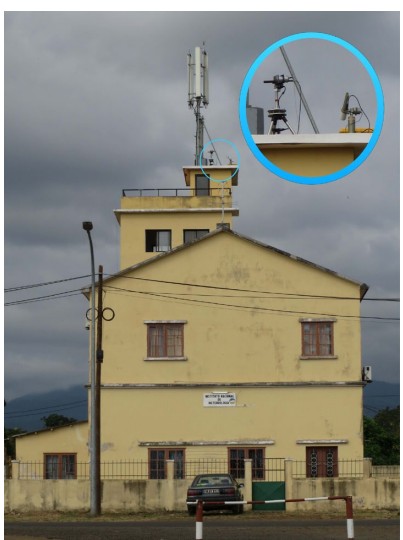

**Figure 5.** A picture of the STP Instituto Nacional de Meteorologia (INM), with an inset of the Pandora and AERONET instruments temporarily installed on the roof-top. Photo credit: James Podolske.

## 4. Description of Deployments

### 4.1 Meteorological context

The key feature of the circulation that transports fire emissions from the African continent over the southeast Atlantic is the easterly flow above about 2 km. Figure 6 shows the 4km flow and relative humidity (RH) from the ERA-Interim reanalysis, along with the southerly limit of significant rainfall from the monthly quarter degree satellite-based 3B43 dataset (Huffman et al., 2007) for the 19-year September mean (2000-2018). The easterly flow maximizes around 8° S, reaching minima near 4° N and 18° S. This flow is maintained by the thermally direct circulation over the continent, which is driven by heating of the elevated African plateau south of 10° S (Adebiyi and Zuidema, 2016). The moister regions to the north are cooler, consistent with the thermal wind relation and the easterly shear below the jet. The jet is similar in character to the African Easterly Jet north of the equator (AEJ-N; consistent with the temperature gradient between the hot Sahara desert and the cooler equatorial region), if not as pronounced. Since there is no heating source over the southeast Atlantic, the jet decreases in intensity as soon as the winds leave the continent. The mean 4 km flow then curls anticyclonically near 10-20°W, and merges with the midlatitude jet, most pronounced in September-October. Smoke associated with this flow has been observed as far away as the South Pacific (Chatfield et al., 2002).



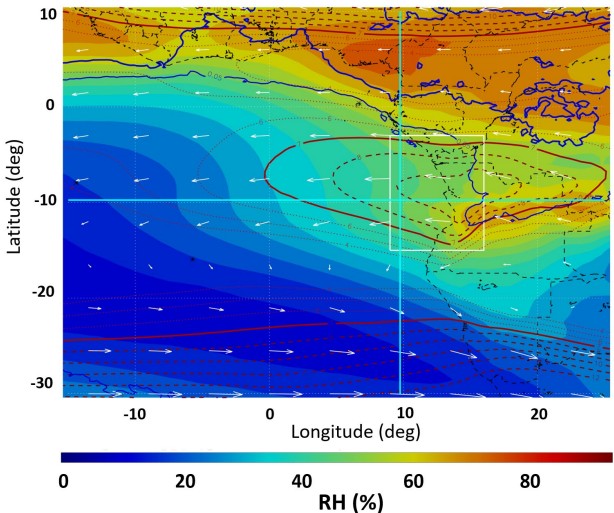

**Figure 6.** September-mean relative humidity (RH – color fill) and horizontal winds at 4km (averaged over 600-650 hPa – red contours and white arrows), and the southerly limit of significant rainfall averaged over 19 years (2000-2018 – blue contour). Heavy solid and dashed red contours indicate elevated wind speeds (7 m/s - solid; 8 and 9 ms$^{-1}$ - dashed). The white rectangle encompasses 8 -16° E and 14° S - 3° S. The cyan lines refer to 9.75° S and 9.75° E, respectively (see Fig. 7). RH and winds are from the ERA-Interim analyses, and the rainfall is from the monthly quarter degree satellite-based 3B43 dataset.

The RH at 4 km altitude provides a useful qualitative indicator of the effects of upward vertical motion (Fig. 6). North of about 3° S, rainfall is substantial (averaging 0.25 mm hour$^{-1}$), and the enhanced RH is almost certainly due to moist convection. The 4 km RH decreases south of 3° S, with a secondary maximum near 10-11° S. This feature is present in other analyses as well (e.g., Modern Era-Retrospective Analysis for Research and Applications MERRA-2, not shown), and occurs at the boundary between the Congo River basin (about 300m) and the elevated African plateau (up to 1500 m). Though moist convection is present in this region, most of the vertical mixing is probably due to dry convection. The effects of this dry convection on the temperature and RH profile have been seen in the occasional radiosondes over south central Africa, and downstream over St Helena Island.

Figure 7 (a and b) shows latitudinal and longitudinal cross sections along the lines indicated in cyan in Fig. 6, also for the 19-year September mean. Figure 7a clearly shows the southern African Easterly Jet (AEJ-S) near 4 km and 8°S, just offshore at 9.75°E. The enhanced RH extends up to about 5-6 km between 10 and 20°S. Notably, even though the flow is easterly (though weaker) above 6 km, the air is dry, indicating that convection is not reaching those levels on a consistent basis. Further north, moist convection is maintaining RH exceeding 80% up to 7 km (and higher, not shown). Another notable feature is the dry tongue extending to 10-15° S at 1-2 km, just above the moist boundary layer. At low levels, there is a strong southerly jet associated with the St. Helena high pressure system. Potential

temperature surfaces slope downward and northward to about 15° S, implying subsidence and drying of the northward

flow (Fig. 6a).

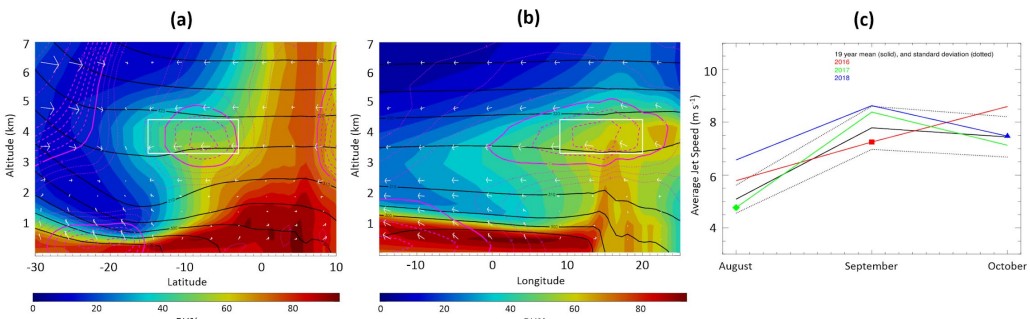

**Figure 7.** September-mean (a) Meridional and (b) zonal cross sections of RH (shading), potential temperature (black contours), and horizontal winds (white barbs), with wind speeds exceeding 7 m/s outlined in magenta contours (7 m s$^{-1}$ - solid; 8 and 9 m s$^{-1}$ - dashed), at a) 9.75° E and b) 9.75° S, respectively, averaged over 19 years (2000-2018). (c)

Strength of the AEJ-S averaged over the volume defined by the white rectangles in panels a and b. The solid lines represent the individual and 19-year (2000-2018) means; the dashed lines represent the standard deviation during the 19 year period.

The longitudinal cross section at 9.75° S, the approximate meridional center of the AEJ-S, indicates that the AEJ-S is

strongest at the coastline. The top of the moist layer is roughly consistent with the top of the daytime boundary layer over the continent near 16° E (Fig. 6b). The RH decreases in the mean as the air flows westward, consistent with the overall subsidence over the SE Atlantic. This subsidence is consistent with modest radiative cooling in this region.

An analysis of the AEJ-S (averaged over a volume the white rectangle in Fig. 6 and the upper left panels in Figs.

7a&b) shown in Fig. 7c reveals that some months during deployment years deviated substantially from the average (such as October 2016). The actual deployment months varied - the AEJ-S strength in Sep 2016 is ~0.7m/s weaker than the climatological mean, the AEJ-S strength in Aug 2017 is weaker than the climatological mean, and the AEJ-S strength in Oct 2018 is very similar to the climatological jet intensity. The individual August-October months of the three ORACLES deployment years are compared to the climatological means of RH and winds at 4 km altitude

(represented as the AEJ-S) in Fig. 8.

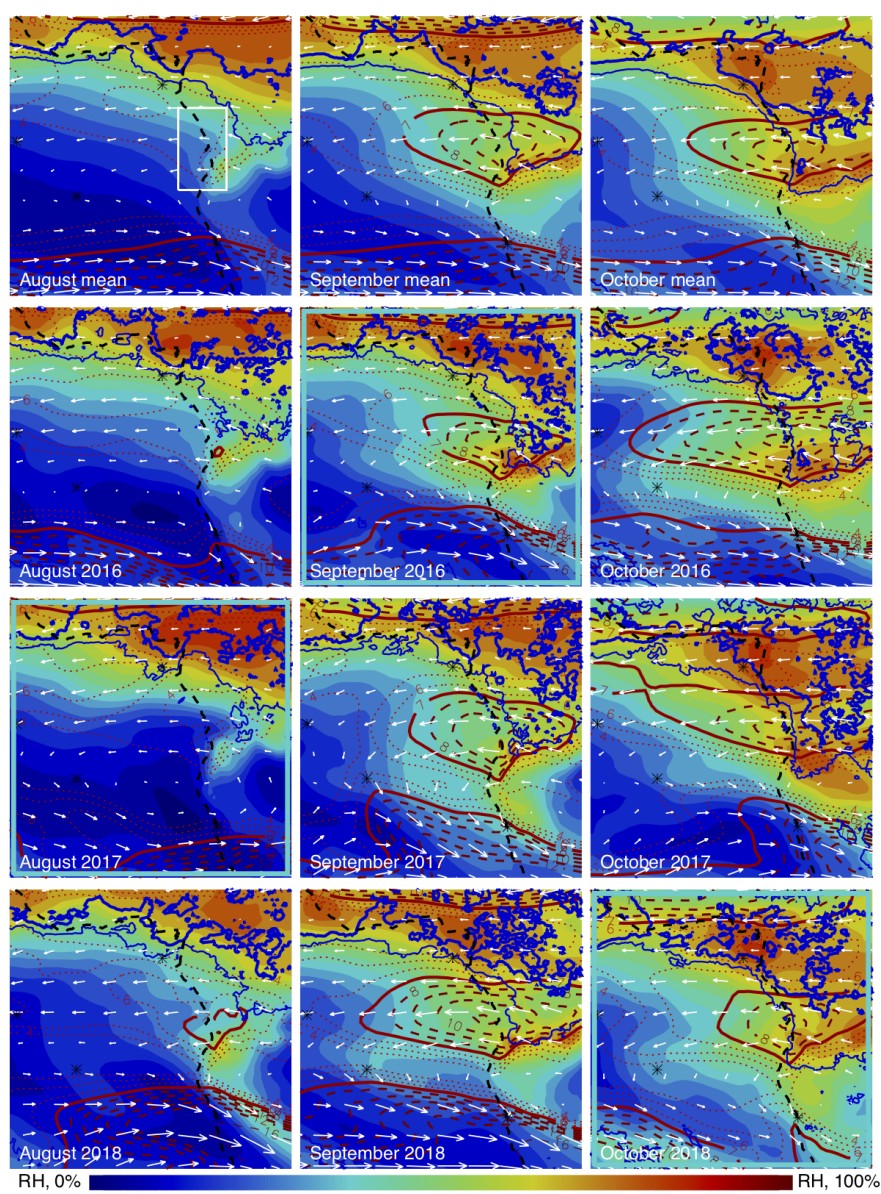

**Figure 8.** The same as Fig. 6 except for the 19-year mean (2000-2018) and 3 years of the ORACLES deployment (2016, 2017, and 2018). The boxes with cyan frames indicate the ORACLES deployment months. The white rectangle represents the area over 8° E - 16° E and 14° S - 3° S.


Clearly apparent is the southward progression of the regions of significant rainfall as the seasons change, and the strengthening of the AEJ-S. Note that the basic westward circulation at 4 km is present for all three months, but is



only about half as strong in August as in the other two months. The other rows represent the three ORACLES years (2016-2018), with the relevant deployment months outlined in cyan. A number of features are apparent. Rainfall in

September 2016 was greater in the AEJ-S region than typical, and RH values were higher. The strength of the AEJ-S in 2016 was about average. The August AEJ-S strength was about average during the 2017 deployment year, though RH values were lower than climatology. The October 2018 AEJ-S strength was also about average, rainfall was about typical, and RH values were lower than the climatology.

Similar to the AEJ-S feature, the southeasterly low-level jet and boundary layer flow intensity during deployment

years are within the range of climatological mean, even though some months deviated substantially from the average. The most important feature of the boundary layer height (BLH) over the area that ORACLES sampled is an overall decrease in BLH from August to October (not shown; Ryoo et al., in prep.).

## 4.2 Aerosol and cloud context

As pointed out above, the interannual variability in fire emissions in Southern Africa was expected to be low [van der

Werf, 2010] and as a result, aerosol loading in the ORACLES region was expected to be repeatable, with analyses of MODIS clear-sky AOD retrievals in the project planning stage implying year-to-year variability through the burning season of only 20% of the mean. Since the beginning of the ORACLES project, a number of MODIS and SEVIRI-based retrieval algorithms for above-cloud aerosol optical depth (ACAOD) have been developed [Meyer et al., 2015; Jethva et al., 2018; Peers et al., 2019; Sayer et al., 2019], allowing a study of the interannual variability of the aerosol

loading above clouds, which is more relevant for ORACLES science objectives than clear-sky AOD. In this section, we describe the interannual variability of MODIS-detected fire counts, ACAOD, and Sc cloud fractions in August, September, and October for the three ORACLES flights years, i.e., 2016-2018. We compare them to the climatologies of the same quantities for the period of 2003 to 2018 for context (Fig. 9). MODIS data are a composite of Terra and Aqua, where available.

Within each panel, Fig. 9 shows monthly averages of fire counts as blue-to-red shading over land, ACAOD as yellow-to-red shading over ocean (Meyer et al., 2015), and low cloud fractions as black contours over ocean, for August, September, and October. The top row of panels shows the 2003 to 2018 climatological means, while the second, third, and fourth row of plots provide the August, September, and October means for the three deployment years, i.e., 2016 to 2018.


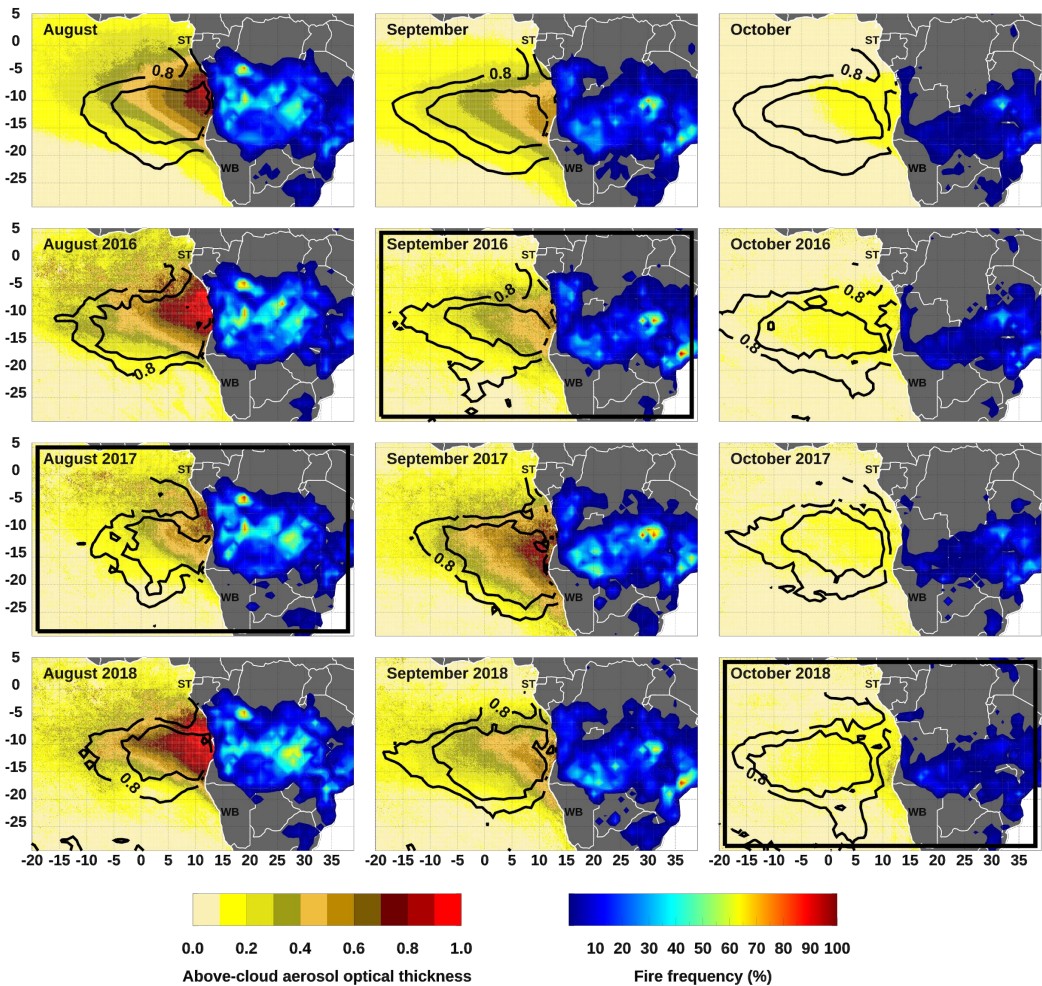

**Figure 9.** Combined Terra+Aqua data for August, September, and October, aggregated at 0.1 degree. Yellow to red shading over water indicates above-cloud AOD, black open contours indicate low-level (cloud-top below 2.5km) cloud fraction of 0.8 and 0.9, and color shading over land indicates MODIS fire frequency for fires with detection confidence above 70%. Top row: climatologies computed over 2003-2018 (coinciding with the Aqua record). Second through fourth row: monthly averages for August through October 2016, 2017, and 2018, respectively. ST and WB stand for São Tomé and Walvis Bay. Bold black frames indicate ORACLES deployment months.

Overall, Fig. 9 corroborates the initial assumption that the interannual variability in fire locations and fire counts is relatively small. While there are differences from month to month, both in a given year and in the climatology, the fire counts in the three deployment years are very similar to the climatologies, with minor differences discussed below.



We attribute this to the interannual consistency in agricultural practices in the various burning regions. The extent of the Sc cloud deck exhibits some interannual variability. The August Sc cloud distributions in 2016 to 2018 appear
quite similar to the August climatologies, while some of the September and October distributions appear to extend slightly farther south or west. The largest interannual differences of any of the quantities shown in Fig. 9 are in the ACAOD. The August 2016 and 2018 plumes appear stronger than the climatological plume, while the August 2017 plume appears weaker than the climatology; the September 2016 and 2018 plumes appear slightly weaker than the climatologies, while the September 2017 plume appears slightly stronger than the climatological plume; the October
plumes in 2016 to 2018 are somewhat reproducible year-to-year, but they all appear slightly stronger than the October climatology. Overall, this supports the conclusion of an earlier and possibly prolonged presence of the BB plume over the SE Atlantic in recent years, relative to the 2003 to 2018 climatology.

As far as the specific ORACLES deployment months are concerned (black outlined panels in Fig. 9), September 2016
shows a slightly weaker ACAOD plume than the climatology. We attribute this mainly to slightly weaker Free Troposphere (FT) winds in the SAEJ and slightly lower RH at plume level (see RH contours, and 7m/s contours not extending as far westward in Fig. 8). Aug 2017 also has a slightly weaker ACAOD plume than the climatology; since fires are as strong or even stronger than the climatology, we again attribute this to a slightly weaker SAEJ than the climatological mean. October 2018 has a slightly stronger plume than the climatology: the SAEJ winds are very similar
to the climatology, but fires in southern Angola appear to be slightly stronger and a more expansive area of elevated RH is present by comparison to the climatology, likely giving rise to the more expansive BB plume in this month.

**4.3 Cloud droplet number concentrations and boundary layer winds**

As ORACLES science objectives encompassed direct, semi-direct, and indirect effects, we were keenly aware of the interannual variability of marine boundary layer pollution levels, as well as boundary layer wind strengths and
directions. In analogy to Sect. 4.2, this section summarizes our current assessment of the interannual variability of boundary layer winds and of cloud droplet number concentrations, $N_c$, the latter as a proxy for BL pollution.

Cloud droplet number concentrations are derived following Painemal and Zuidema (2011) as:

$$N_c = 1.4067 \times 10^{-6}[cm^{-1/2}] \times \frac{(COD)^{1/2}}{(CER)^{5/2}}, \qquad \text{(equation 1)},$$

where COD is the cloud optical depth and CER is the cloud droplet effective radius near cloud top. Equation (1) assumes an adiabatic liquid water content profile and a vertically uniform $N_c$ (Szczodrak et al., 2001). $N_c$ is computed from COD and CER products that have accounted for the above-cloud AOD (Meyer et al., 2015). These products are used instead of those from the MODIS standard product (i.e., MXD06) where CODs are typically underestimated in the retrievals due to the top of atmosphere (TOA) shortwave reflectance absorption by overlying smoke aerosols
(Haywood et al., 2004, Coddington et al., 2010, Meyer et al., 2013).

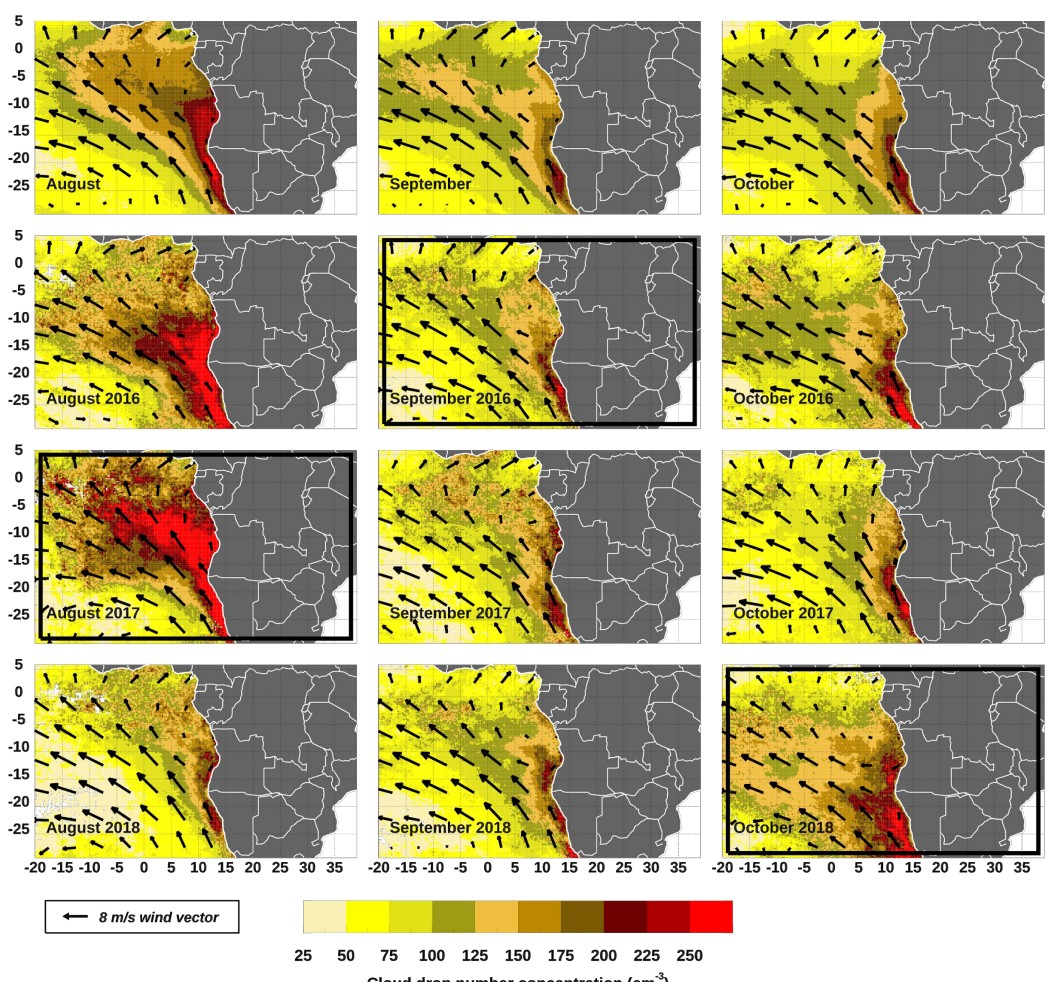

**Figure 10.** As in Fig. 9, but for $N_c$ from combined Terra+Aqua retrievals for August, September and October, aggregated at 0.1 degree. Black arrows denote 925mb wind vectors with 8 m/s wind vector scale shown in the black box. The bold black frames indicate ORACLES deployment months.


Within each panel, Fig. 10 shows monthly averages of $N_c$ calculated from MODIS Aqua and Terra cloud optical depth (COD) and effective radius (CER) retrievals as yellow-to-red shaded contours, along with 925mb wind data from National Centers for Environmental Predictions (NCEP) reanalysis. As in Fig. 9, the top row of plots shows the climatological mean from 2003 to 2018 for August, September, and October, while the second, third, and fourth row

show the monthly means for the same months in 2016, 2017, and 2018, respectively. Overall, there are striking deviations in the $N_c$ averages in every year relative to the climatological mean. August 2016 and August 2017 show significantly larger $N_c$, while August 2018 shows a lower $N_c$ relative to climatology. By contrast, the September $N_c$ values in 2016 to 2018 resemble the climatological mean quite closely. October 2016 and October 2017 are again





quite similar to the October climatology while the October 2018 $N_c$ values are significantly elevated relative to
climatology. MBL winds in most months were similar to the climatological means, with the notable exception of the
2018 October ORACLES deployment, when the MBL winds were somewhat weaker than the climatological means.

To summarize our findings regarding the general plume and MBL pollution levels during the ORACLES deployment
months (September 2016, August 2017, and October 2018), we note that the September 2016 FT plume and boundary
layer $N_c$ values were quite similar to the climatologies of these quantities. August 2017 featured a FT BB plume with
notably lower ACAOD, but an MBL with significantly elevated $N_c$ relative to the climatologies. October 2018 featured
a markedly more expansive FT ACAOD plume and a simultaneously more polluted MBL with significantly elevated
$N_c$. The detailed explanations for these interannual variations are currently the subject of at least one ORACLES-
related investigation [Ryoo et al., in prep.].

**4.4 Description of flights and links to data**

In total, the P-3 aircraft flew 350.6 flight hours in 44 flights for science operations between September 2016 and
October 2018, while the ER-2 flew 97.3 flight hours in 12 science flights in 2016. Both tallies include transit flights
with science data collections into and out of the deployment sites, because on many occasions valuable science data
were collected during these flights. The P-3 flight hours and flight counts exclude an attempt at a transit flight from
Ascension Island in 2017, which had to be aborted due to an aircraft malfunction.
Figure 11 shows the flight tracks of the three P-3 ORACLES deployments in 2016 (light blue), 2017 (orange) and
2018 (dark blue), and the ER-2 flights in 2016 (green). In each of the flight years, about half of the P-3 flights lie on
top of each other along the routine flight track as described above. For clarity, the 2018 P-3 flight tracks have been
offset by 0.1 degrees in longitude to allow distinction from 2017 P-3 flight tracks. In their totality, these flight tracks
cover a vast portion of the climatological SE Atlantic SC cloud deck and the overlying BB plume.

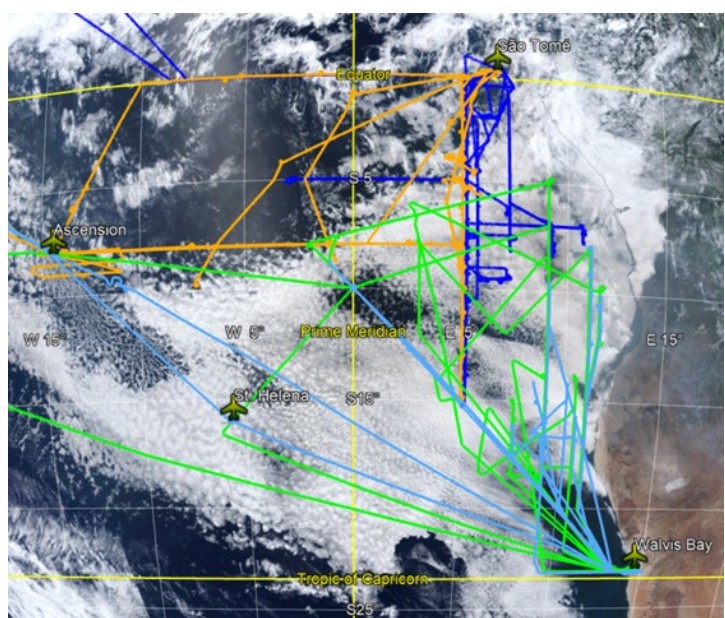

**Figure 11.** Flight tracks of the ORACLES aircraft in 2016 to 2018, overlain on a MODIS-Aqua True Color Image acquired on Sep. 13, 2018. ER-2 flight tracks in 2016 (only deployment year) are shown in green. P-3 flight tracks in 2016, 2017, and 2018 are shown in light blue, orange, and dark blue, respectively. 2018 P-3 flight tracks are offset by 0.1 degrees in longitude to allow distinction from 2017 flight tracks.

Table 2 summarizes eight types of flight maneuvers conducted during research flights and briefly describes the purpose for each maneuver. Figure 12 provides a schematic representation of each of these flight maneuvers.


**Table 5.** P-3 flight maneuvers conducted during research flights, and their primary purpose.

| Flight Maneuver | Primary Purpose |
|---|---|
| 1. Ramp - RA | Continuous profile of thermodynamic state variables, aerosol and cloud properties from in situ measurements, often conducted to maximize the geographic extent of a given flight, while also sampling vertical gradients. |
| 2. Square Spiral - SS | A spiral descent (or ascent) for localized radiation and in-situ profile measurements, modified to include four 20-30s level segments every 90° heading for radiation measurements, giving the pattern the appearance of a box with rounded edges. |
| 3.MBL Leg - ML | A constant-altitude leg in the MBL to assess aerosol properties and trace gas concentrations, usually below cloud. |





| 4. In-Cloud Leg - ICL | A nearly constant-altitude leg, deliberately placed at an altitude of specific cloud interest, e.g., in the thickest part of the cloud or near cloud-top for remote sensing validation. |
|---|---|
| 5. Above-cloud Leg - ACL | A constant-altitude leg, usually just above cloud top, to ascertain the presence of a clear slot and to measure full-column above-cloud aerosol plume with remote sensing/radiation instruments. Sometimes this leg was optimized in altitude to provide sufficient stand-off for radar observations. |
| 6. Sawtooth Leg - STL | A flight segment that continuously profiles from just above cloud top to just below cloud base, to ascertain the cloud vertical structure; also useful for quick above cloud aerosol assessments at the top of each saw-tooth. |
| 7. In-Plume Leg - IPL | A constant altitude leg deliberately placed at a specific altitude to assess aerosol properties with in situ measurements; frequently in the heaviest aerosol loading to provide the largest signal to noise. |
| 8. Above-Plume Leg - APL | A constant altitude leg above the BB aerosol layer, usually intended for full-column lidar assessments with HSRL-2. |

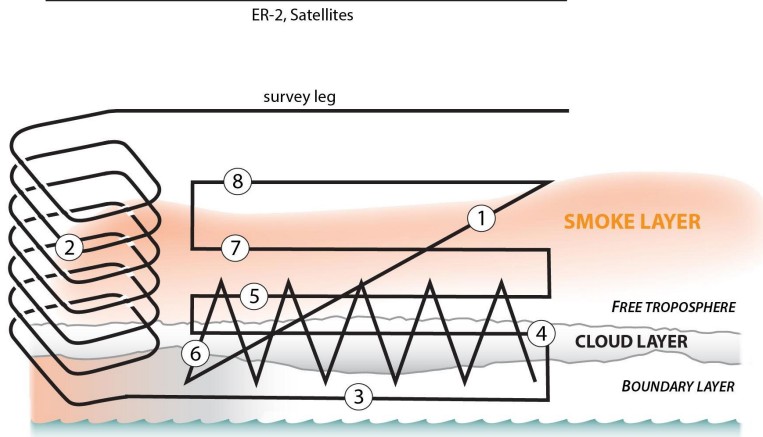

**Figure 12.** Schematic of flight maneuvers during research flight  linked to descriptions in Table 2 and flight synopses in Tables A1-A3.






Tables A1-A3 in Appendix A summarize the flights of both ORACLES aircraft in 2016, 2017, and 2018. They include a brief flight synopsis and the number of each of the eight flight maneuvers described in Table 2 for every flight. Detailed flight reports for each of the flights listed in Tables A1-A3 can be found at the project web-page:

https://espo.nasa.gov/oracles/mission-flight-docs. ORACLES data is archived permanently with separate Digital Objective Identifiers (DOI) for each deployment year and separate DOI for the two aircraft participating in the 2016 deployment (see references for ORACLES Science Team, 2020 - 2016 P3 data, 2016 ER2 data, 2017 P3 data, and 2018 P3 data). A link to images and kmz files for flight tracks, data from ground-based instruments, and other auxiliary information, can be found at https://espo.nasa.gov/home/oracles/content/ORACLES_Science. Video footage from the

P-3 front and nadir cameras are available at https://asp-archive.arc.nasa.gov/Oracles/N426NA/Video/.

## 5. Preliminary science findings and implications

### 5.1 List of golden days for various objectives

The target of opportunity flights, described in Sect. 4.4, usually were designed to address more focused scientific objectives within the context of the general science objectives. Broadly, these objectives can be characterized as

pertaining to (i) the radiative interactions of aerosols and clouds (radiation flights), (ii) the microphysical interactions of aerosols and clouds as they are affected by vertical mixing, drizzle suppression, etc. (cloud microphysics flights), and (iii) the spatiotemporal evolution of aerosol microphysics in the BB plume (plume evolution flights). In addition, a number of flights provided an excellent data set for the evaluation of remote sensing concepts (see Sect. 5.3.7), often on the basis of successful coordination of the P-3 with the ER-2 in 2016 or with satellite assets in 2017 and 2018.

Table 3 provides a list of "golden days" for each observational focus, to be interpreted as the likely best start of exploration of the ORACLES data set for an uninitiated science user. The ordering of flights within each category is based on a preliminary assessment of the utility and quality of the data to address the observational objective. Flights with coordinated P-3 and ER-2 sampling in 2016 are postulated to provide superior data and hence appear before any single-aircraft flights. Future analyses are likely to provide a revised list of useful flight days in the context of detailed

and overarching ORACLES objectives.

**Table 3.** ORACLES golden flight days by observational focus.

| Observational focus | Dates - P-3/ER-2 Flight number |
|---|---|
| Radiation | 09/20/2016 - PRF11Y16/ERF06Y16<br>09/14/2016 - PRF09Y16/ERF03Y16<br>09/27/2016 - ERF10Y16<br>09/02/2016 - PRF03Y16 |
| | 08/13/2017 - PRF02Y17<br>08/26/2017 - PRF09Y17 |





| | 09/30/2018 - PRF02Y18<br>10/05/2018 - PRF05Y18 |
|---|---|
| Cloud microphysics | 09/20/2016 - PRF11Y16/ERF06Y16<br>09/14/2016 - PRF09Y16/ERF03Y16<br>09/06/2016 - PRF05Y16<br>09/25/2016 - PRF13Y16 |
| | 08/28/2017 - PRF10Y17<br>08/13/2017 - PRF02Y17<br>09/02/2017 - PRF13Y17 |
| | 10/03/2018 - PRF04Y18<br>10/02/2018 - PRF03Y18<br>10/23/2018 - PRF13Y18<br>10/12/2018 - PRF08Y18 |
| Plume evolution | 09/18/2016 - PRF10Y16/ERF05Y16<br>09/24/2016 - PRF12Y16/ERF08Y16<br>09/06/2016 - PRF05Y16 |
| | 08/17/2017 - PRF04Y17<br>08/21/2017 - PRF07Y17<br>08/31/2017 - PRF12Y17 |
| | 10/17/2018 - PRF10Y18<br>10/19/2018 - PRF11Y18 |
| Remote sensing test bed data | 09/20/2016 - PRF11Y16/ERF06Y16 |

## 5.2 Examples of significant findings

### 780  5.2.1 HSRL-2 statistics on contact of plume and cloud tops

An important line of inquiry and indeed a major motivation for the ORACLES project was the question of how frequently Sc clouds in the SE Atlantic MBL are in physical contact with the BB plume emanating from the southern African continent. The only prior work in the area using in situ aircraft measurements (Haywood et al., 2003) found frequent separation between cloud and aerosol close to the coast of Namibia and Angola but potential interactions

between cloud and aerosol in the vicinity of Ascension Island. However, the statistical relevance of those findings is



impossible to establish given the scarcity of the available in situ data. For a large number of coincident CALIOP and MODIS aerosol and cloud retrievals in the SE Atlantic region, Costantino and Breon (2013) found that more than half of the vertical profiles they studied indicated "well-separated" aerosol and cloud layers, i.e., an "aerosol-free gap" above cloud top. They went on to attribute the differences in cloud effective radii between separated and unseparated cases to the probable paucity of aerosols in the MBL in the separated cases. ORACLES observations show that the separation of aerosol and cloud layers in an instantaneous profile is a poor indicator for the concentration of aerosol in the MBL, because there are many other pathways for the aerosol to reach the MBL at a given location. However, the physical contact of the BB plume and Sc cloud tops is indicative of active entrainment of BB aerosol into the MBL and is often associated with significant BB aerosol in the MBL. The relatively low signal-to-noise (SNR) in CALIOP vertical profiles after traversing a BB plume with significant optical attenuation, also alluded to in Sect. 2.1, raises the question of how accurate the CALIOP-indicated frequent separation between the BB plume and Sc cloud tops (shown in Fig. 2) really is.

To help address this question, we investigate here how frequently the HSRL-2 aerosol extinction profiles in the three ORACLES deployments indicate separation or lack thereof between the BB plume base and the Sc cloud tops. The left panel of Fig. 13 shows mean aerosol extinction profiles at 532nm for each ORACLES deployment year, along with mean cloud top heights as horizontal solid lines. Because ORACLES-2016 had a somewhat different geographic focus from ORACLES-2017 and ORACLES-2018, Fig. 13 only shows data for the geographic box bounded by 5-15˚S and 2.5-7.5˚E, where there are data present in all three flight years.

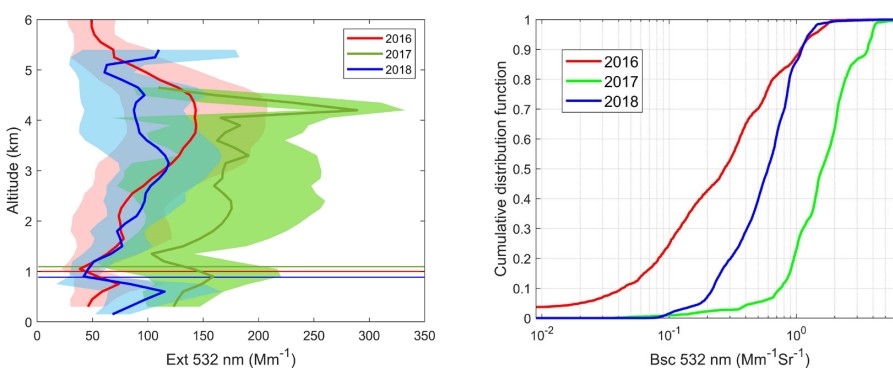

**Figure 13.** Left panel: A composite of vertical profiles of aerosol extinction at 532nm, derived from HSRL-2 measurements for each of the deployment months bounded by 5-15˚S and 2.5-7.5˚E. Shaded regions represent one standard deviation of extinction measurements. Solid horizontal lines indicate the mean cloud top heights for each data set. Right panel: Cumulative distribution function for mean aerosol backscattering in the 300-m layer above cloud top.

The mean extinction profiles show that the BB plume in ORACLES-2017 (August) has a local maximum of aerosol extinction at altitudes between 2-3 km, while the ORACLES-2016 (September) and ORACLES-2018 (October) mean profiles have maximum aerosol loading between 3.5 and 4.5km in altitude, albeit with somewhat lower values. In all



three years, there is a relative minimum in aerosol loading near the location of the mean cloud tops. This suggests that the air mass just above cloud top may indeed be significantly older, more processed and scavenged than the FT above and possibly even the BL below, which may have been subject to entrainment of FT air upstream of the BL flow to the south of the bounding box.

The separation between the BB plume base and the cloud tops for each year is indicated in the right hand panel in Fig. 13. It shows the cumulative distribution of HSRL-derived aerosol backscattering coefficient at 532nm in the 300m layer just above cloud top for the same geographic region as the left panel. The cumulative distribution of aerosol backscatter values near the cloud top varies sigmoidally, without any discontinuity that would indicate an unambiguous lower edge to the aerosol layer. Instead, it is apparent that the assessment of an "aerosol-free gap",

depends entirely on the definition of the aerosol loading. For example, the August 2017 and October 2018 deployments featured a relatively low fraction of profiles with less than 0.25 $Mm^{-1}Sr^{-1}$ aerosol backscatter coefficient in the first 300 m above cloud, i.e., 3% and 15% for 2017 and 2018, respectively. The September 2016 profiles in this region on the other hand show less than 0.25 $Mm^{-1}Sr^{-1}$ aerosol backscatter coefficient, and hence an aerosol-free gap, almost 50% of the time. Regardless of the exact threshold chosen, September 2016 showed by far the highest frequency of

aerosol-free gaps above clouds when compared to the 2017 and 2018 deployments. The representativeness of this September maximum in aerosol-free gaps above clouds is unknown and will need to be explored with future satellite lidar observations of sufficient accuracy. Overall, we found significant spatial and temporal variability in the degree of contact between BB aerosol and Sc clouds in the SEA region, and that the frequency of occurrence of aerosol-free gaps is likely much lower than previously assumed (Costantino and Breon, 2013). These findings need to be

considered when interpreting previous results on aerosol-cloud interactions in the region.

### 5.2.2 Vertical plume structure and chemical composition

The lidar-derived increase in extinction with height for September (2016) in Fig. 13 is accompanied by a similar increase with height of the mean in-situ SSA (derived from the in-situ PSAP absorption paired with nephelometer scattering at 530 nm) (Fig. 14, top row). The increase in mean SSA with height is consistent with a decrease of

refractory black carbon (BC) mass concentration relative to mean organic aerosol (OA) with height (Fig. 14, middle row), as black carbon is the primary absorber of sunlight within BB aerosol (e.g., Bond et al., 2013).

Profiles from August 2017 indicate less of a vertical structure in SSA (and correspondingly in the BC/OA ratio), while the mean SSA from the October (2018) deployment increases even more sharply with altitude than does the mean SSA from September (2016). The vertical profiles in SSA are consistent with the corresponding BC/OA (black carbon

to organic aerosol) ratio for the three deployments. The increase of SSA with height is consistent with the proportional increase in organic aerosol relative to black carbon, as oxidized/bleached organic compounds do not absorb as much sunlight per molecule as does black carbon.

Examples of individual profiles from each year, broken down by aerosol species [black carbon, organic aerosol, nitrate ($NO_3$), ammonium ($NH_4$) and sulfates ($SO_4$)], indicate distinct vertical structures (Fig. 14 bottom row). For example,





the individual profile from the September 24, 2016 flight indicates more nitrate above 3.5km than below it, relative to
the black carbon mass concentration.

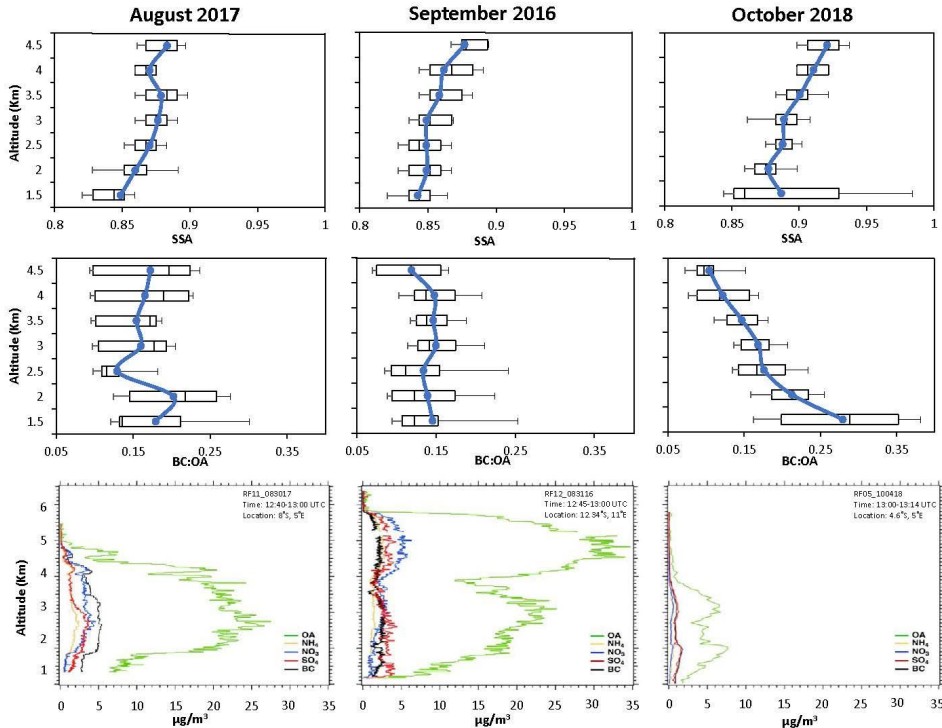

**Figure 14.** Top row: deployment-mean 530nm SSA vertical profiles for each year for the August (2017), September
(2016), and October (2018) deployments. Middle row: the corresponding mean black carbon to organic aerosol ratio
(BC:OA) profiles. Bottom row, example profiles of individual aerosol species, from left to right: August 30, 2017
12:40-13:00 UTC, 8° S / 5° E, September 24, 2016, 12:45-13:00 UTC, 12.34 S° / 11° E, and October 4, 2018, 13:00-
13:15 UTC, 4.6° S / 5° E.


One explanation for the seasonal evolution in the aerosol composition and optical properties can be drawn from the
free-tropospheric winds shown in Fig. 8, in which the winds at 4 km are much stronger in September and October
than in August. This distributes the BB aerosol which reaches that altitude further offshore in September and October
than in August. A slow subsidence will eventually place the older aerosol at the lower altitudes, with stability gradients
within the aerosol layers (e.g., Garstang et al., 1996) explaining the lack of internal mixing within the aerosol layer.
Results from filter-based absorption measurements combined with nephelometer measurements of scattering in
airborne observations have been suggested to give significant biases in SSA (e.g., Lack et al., 2008). However, Davies
et al. (2019), using CLARIFY data and suitable correction algorithms, suggested that their algorithms were able to



reduce biases in absorption. Indeed, the SSA values shown in Fig. 14 are in reasonable agreement with values of ~0.83 for dry conditions in Davies et al. (2019) derived using state-of-the-art photoacoustic and cavity ring-down instrumentation. Pistone et al., (2019) also compared the ORACLES absorption+scattering measurements with SSA derived by several different airborne remote-sensing methods at wavelengths between 400 and 995nm and found reasonable agreement both for specific case studies and for the range of measured spectral SSA over the full ORACLES-2016 deployment.

**5.2.3 Chemical Aging**

Ongoing analysis of the ORACLES dataset is relating the observed vertical structure in aerosol composition and optical properties to aerosol aging. Rather than using an aircraft to track the evolution of a smoke plume as it advects, or using an aerosol chemical component as a 'chemical clock' to determine aerosol age, an alternative approach links a model-derived mean aerosol age to in-situ aerosol characteristics. Particles are marked at the time of the model 880 initialization and tracked in time thereafter (Saide et al., 2016), with the particle ages subsequently extracted along the flight legs, and these modeled ages combined with the in-situ datasets. An example is shown in Fig. 15 for 24 September 2016. Aerosol age comes from one of the ORACLES in-field aerosol forecasting models, WRF-AAM (see Section 3.3.5). The metric f44 is the measured mass to charge ratio mz 44 relative to total organics, with higher fractions of f44 reflecting the formation of carboxylic acids, which coincide with older aerosol ages up to ten days. 885 The ORACLES f44 values shown here are on par with those reported for Siberian biomass burning aerosol after trans-Pacific transport to Alaska. Notable is that younger aerosol overlies older aerosol on this day. Ongoing work is relating the model-derived aerosol aging to a continuous depletion of non-black-carbon aerosols (Dobracki et al., 2020).

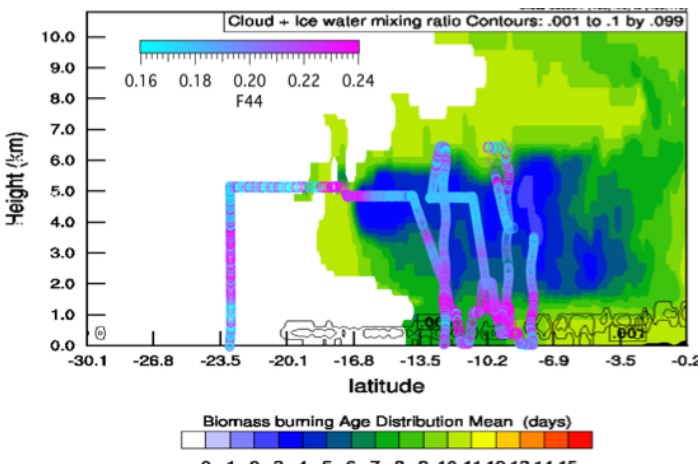

**Figure 15.** The WRF-AAM model-derived mean aerosol age along 10 E as a function of latitude for 24 September, 2016, overlain with a measure of aerosol oxidation derived from in-situ f44 measurements on the P-3 aircraft.



### 5.2.4 Boundary layer clouds - CCN, cloud and rain statistics

ORACLES scientific objective 3 (Table 1) seeks to understand the extent to which smoke aerosol from southern Africa mixes into the MBL and impacts marine low cloud microphysical processes. Smoke may enhance the population of cloud condensation nuclei (CCN), which can lead to increases in cloud droplet concentration and smaller droplet sizes. It can also potentially induce changes in cloud condensate through suppressing precipitation rates, enhancing cloud-top entrainment, and otherwise influencing cloud macrophysical properties. These indirect effects lead to significant increases in albedo in global models, but little in situ data are available over the SE Atlantic to test these models. According to HSRL-2 observations (Fig. 13), a denser, more frequent smoke layer in the FT occurred in 2017 than in 2016 or 2018. CCN measurements made on all three ORACLES deployments (Kacarab et al. 2020) provide evidence that FT CCN concentrations were also higher in 2017 (Fig. 16a). Despite the frequent contact of the smoke with the sub-cloud planetary boundary layer (PBL) top, concentrations of CCN in the PBL are much lower than those in the FT (compare panels a and b in Fig. 16). Also, despite very similar FT CCN concentrations in 2016 and 2018, PBL CCN levels were actually significantly higher in 2018 than in 2016 (Fig 16b), suggesting that the FT CCN concentration alone is not a unique determinant of microphysical properties in the PBL. As noted in Diamond et al. (2018), it is the history of subsidence and entrainment of smoke into the PBL, rather than the instantaneous presence of aerosol immediately above clouds, that sets the MBL CCN level and therefore cloud droplet concentration ($N_c$). Seasonally, satellite evidence (Fig. 10) indicates that the highest $N_c$ values offshore occurred in 2017, and this is consistent with the highest measured PBL CCN concentrations (at supersaturations > 0.25%) occurring during 2017. October 2018 also showed high $N_c$ values offshore consistent with the higher measured CCN concentrations.

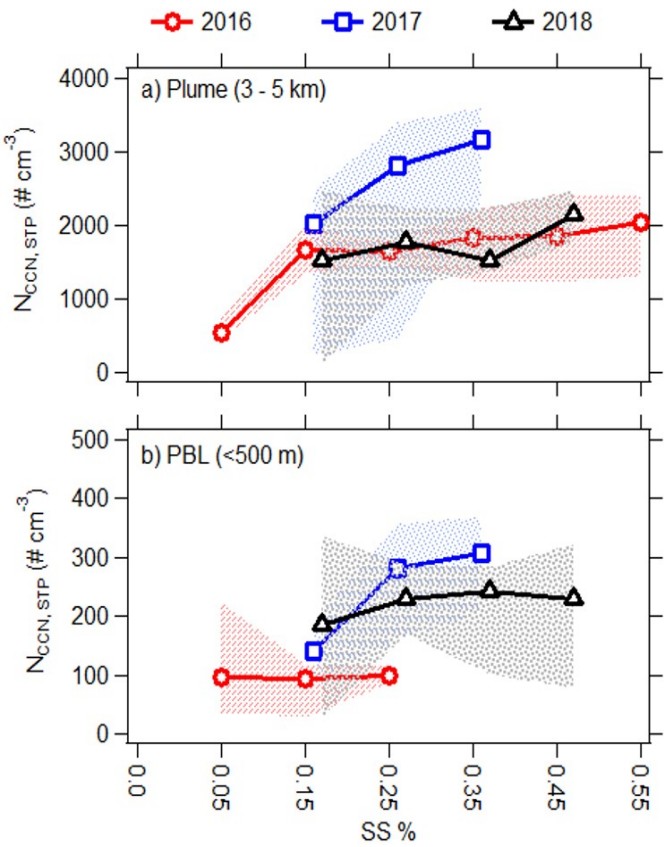

**Figure 16.** Composite CCN supersaturation spectra (concentration of CCN as a function of the applied supersaturation, SS) for (a) the free-tropospheric BB plume, and (b) the MBL within a region (5-15° S and 2.5-7.5° E) sampled during all three deployments.

In addition to entrainment of smoke from the FT, the concentration of CCN and resulting droplet concentrations in the marine PBL are also modulated strongly by coalescence scavenging by light precipitation (Wood et al. 2012). In addition, evidence suggests that precipitation can be suppressed in clouds with high CCN and $N_c$ (Sorooshian et al. 2010). The APR-3 radar on the P-3 is sufficiently sensitive to detect this light precipitation. Measurements of precipitation from the APR-3 indicated significant differences in precipitation during the three campaigns (Fig. 17). Comparing the two campaigns flown out of São Tomé (2017 and 2018), precipitation was significantly lighter in 2017 (Fig. 17), which may indicate suppression due to higher droplet concentrations, but additional analysis of cloud thickness and liquid water path differences between the campaigns is underway to quantify these impacts.

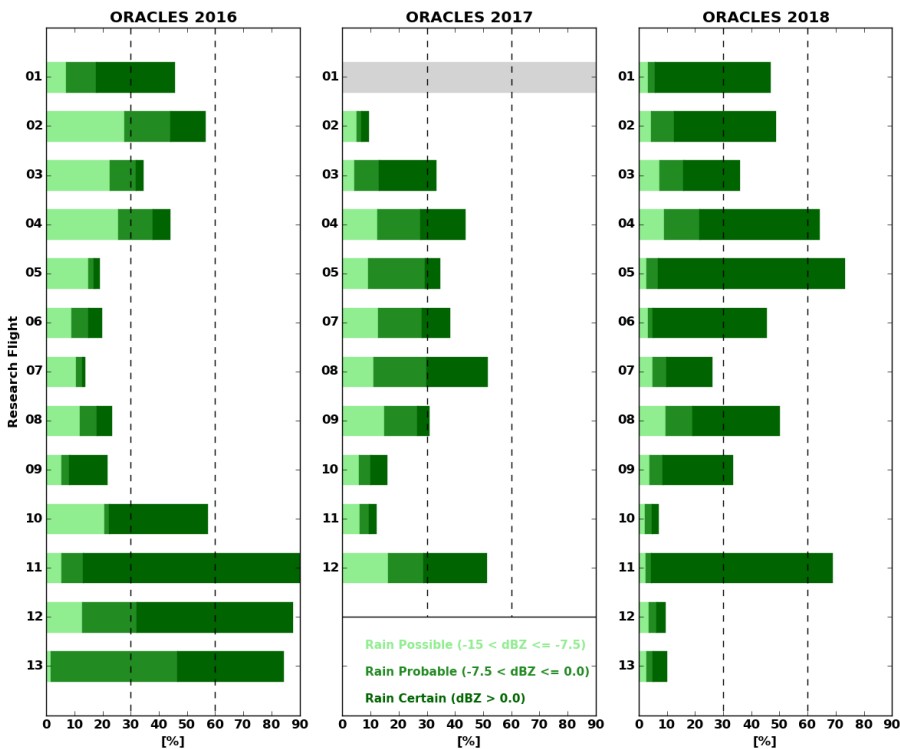

**Figure 17.** The fraction of clouds detected using the APR-3 radar in which rain is possible, probable, and certain, for the ORACLES research flights in 2016 (left), 2017 (center), and 2018 (right).

### 5.2.5 Evidence of aerosol indirect effects

Observations obtained by in-situ probes installed on the NASA P-3 are being used to investigate cloud-aerosol interactions. Over the 3-year period of ORACLES, 397 vertical profiles were flown through clouds during either sawtooth profiles (see Fig. 11) or in individual cloud profiles isolated in time/space from other profiles. The vertical dependence of cloud properties in a common reference frame is examined using a normalized altitude $Z_N$, defined as $(Z-Z_b)/(Z_t-Z_b)$, where $Z$ is altitude, $Z_t$ cloud top altitude and $Z_b$ cloud base altitude. This allows us to composite the

vertical microphysical structure of cloud decks from many different clouds.

We select 397 vertical profiles through the cloud layer when accumulation-mode $(0.1 < D < 3\ \mu m)$ aerosol concentration $(N_a)$ was measured at least 100 m above and below cloud using the Passive Cavity Aerosol Spectrometer Probe (PCASP). Each profile was classified as "Contact" or "Separated" based on whether the layer of enhanced above-cloud aerosol concentration (defined as PCASP $N_a > 500\ cm^{-3}$) was in contact with cloud top or separated from

it by at least 100 m. Each of the "Contact" and "Separated" profiles was further classified based on whether the average PCASP $N_a$ within the 100 m layer below cloud base was greater than or less than 250 $cm^{-3}$. Four different regimes



were thus defined based on the below-cloud boundary layer $N_a$, and by whether or not the cloud layer was in contact with an overlying BBA plume, or separated from it.

Figure 18 shows the mean cloud droplet concentration $N_c$ as a function of $Z_N$ for the four regimes. The average boundary layer $N_a$ for the regimes are as follows: Contact/$N_a < 250$ cm$^{-3}$, 184 cm$^{-3}$; Contact/$N_a > 250$ cm$^{-3}$, 507 cm$^{-3}$; Separated/$N_a < 250$ cm$^{-3}$, 113 cm$^{-3}$; Separated/$N_a > 250$ cm$^{-3}$, 318 cm$^{-3}$. The average boundary layer $N_a$ is a good predictor of the vertical mean $N_c$ across the four regimes, consistent with the primary source for droplets being aerosols ingested into cloud base (Diamond et al. 2018). Although $N_c$ is approximately constant with height for the two Separated regimes and the Contact/$N_a < 250$ cm$^{-3}$ regime, consistent with many previous studies (Nicholls and Leighton 1986, Martin et al. 1994, Miles et al. 2000, Painemal and Zuidema, 2011, Wood 2012), it experiences significant increase with $Z_N$ for the Contact/$N_a > 250$ cm$^{-3}$ case, possibly for reasons hypothesized by Gupta et al. (2020). The highest $N_c$ values occur where the MBL has both high $N_a$ and is in contact with overlying BB aerosol layers. The lowest boundary layer $N_a$ is found where there is separation from the overlying aerosol. It is perhaps surprising that clouds within boundary layers with higher $N_a$ but not overlain by aerosol, or clouds within boundary layers with lower $N_a$ but overlain by aerosol, possess very similar $N_c$. This further suggests that clouds' history of interacting with aerosol within the prior days (Mauger and Norris, 2007; Diamond et al., 2018) can have an impact on $N_c$, but that aerosols entrained from above cloud top also have an impact on $N_c$. These findings, and hypotheses for the processes responsible, are further described by Gupta et al. (2020).

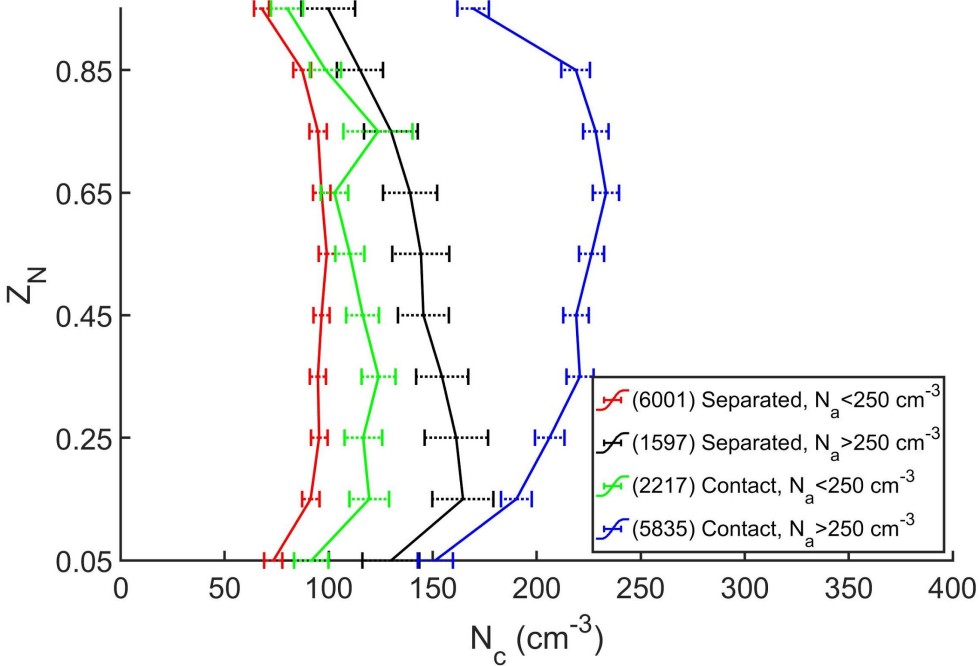





**Figure 18.** Mean cloud droplet number concentration ($N_c$) as a function of normalized cloud depth ($Z_N$), where the mean is computed for 397 vertical profiles flown during the 3 ORACLES campaigns. Different colors correspond to whether the boundary layer below cloud has average aerosol concentration $N_a < 250$ cm$^{-3}$ or $N_a > 250$ cm$^{-3}$ as measured by PCASP and according to whether a layer of BB aerosol is in contact with ($N_a > 500$ cm$^{-3}$ within 100 m of cloud top) or separated from cloud top (no layer with $N_a > 500$ cm$^{-3}$ within 100 m of cloud top). Uncertainties indicated by

horizontal error bars represent the 95% confidence intervals. Numbers in parentheses represent the number of 1 s data points included in each of the profiles. See text for details.

### 5.2.6 Comparisons of models and observations

Approximately one-half of all of the flights were devoted to a routine path, motivated by a desire to facilitate model improvement through an unbiased sampling performed frequently enough to adequately capture the monthly-mean

(see Sect. 3.3.6). An a-priori evaluation based on the random sampling of clear-sky aerosol optical depths rationalized the decision to allocate 6-8 flights to a routinely sampled flight line, on random days during the deployment. The initial model-observation comparison (Shinozuka et al., 2020), which is based on the 2016 measurements only, includes model versions similar to those used for the aerosol forecasts in the field [WRF-CAM5 (Weather Research and Forecasting - Community Atmosphere Model 5) and GEOS-5 (Goddard Earth Observing System, Version 5)], as

well as the UK Meteorological Office Unified Model (UM), the French ALADIN-Climate (Aire Limitée Adaptation dynamique Développement InterNational - Climate, Mallet et al., 2019), the global GEOS-Chem (Goddard Earth Observing System - Chemistry) model, and the E3SM (Energy Exascale Earth System Model) Atmosphere Model (EAM). Measured variables which are compared include the aerosol layer top/bottom boundaries (as determined from lidar and in-situ), aerosol extinction (lidar and in-situ), black carbon and organic aerosol mass concentrations (SP2

and AMS), carbon monoxide, scattering and absorption Ångström exponents, and the single scattering albedo (in-situ). The new datasets allow an extension beyond previous assessments emphasizing the aerosol layer boundaries only (Das et al., 2017; Koffi et al., 2012). Not all of the models include all of these variables, and no effort is made to standardize model features such as the emissions databases and frequency of initialization. The project confronted the issue of how best to compare infrequently-sampled but detailed measurements, to frequently-sampled but coarsely-

resolved model output, through aggregating both measured and model data into approximately 2 by 2 degree grid boxes centered on the routine flight track (see Fig. 19 for an example). This approach is similar to that applied within the AeroCom community (e.g., Katich et al., 2018; Myhre et al., 2013). The smaller grid spacings applied within a larger domain also clarify the ability of models to transport aerosol further offshore. A further study will apply a similar approach to model-observational comparisons based on the 2017 and 2018 deployments.

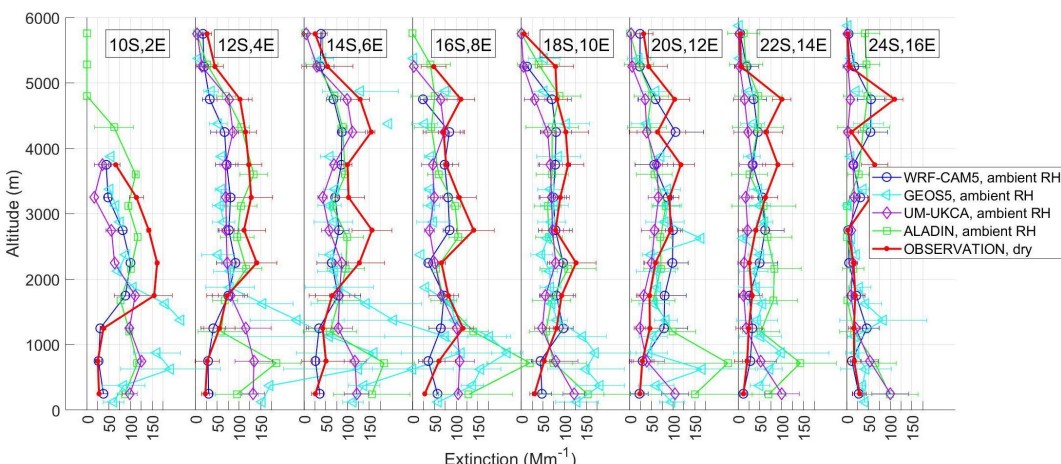

**Figure 19.** Vertical distribution of mid-visible ambient aerosol extinction from four models compared to dry ORACLES observations in 2x2 degree grid boxes (locations indicated) along the 2016 routine flight track.

Shinozuka et al. (2020) assessed modeled aerosol properties through examining three separate layers and concluded that the upper 3-6 km layer generally contains less aerosol in the models than is observed, primarily because the aerosol layer tops are placed too low. Another approach, applied within Doherty et al. (in prep.) is examining the vertical distribution of aerosol (as in Fig. 19) and those cloud properties (cloud fraction and optical depth) important for quantifying the direct aerosol radiative effect. For example, they find significant differences in both the absolute value and vertical structure of SSA in the observations versus in the models, and significant differences between models. Consistent with Shinozuka et al. (2020), the model bias has an altitude dependence that results from the models generally placing the plume at too low an altitude. The covariance in the model bias in extinction and SSA with altitude affects the column SSA – the parameter of interest for determining plume aerosol direct radiative effect. Therefore Doherty et al. also compare observed to modeled extinction-weighted (or column-aerosol) SSA for the smoke plume. This and related analyses in the paper lay the groundwork for determining which modeled parameters are contributing most to biases in modeled aerosol direct radiative effect of the smoke.

### 5.2.7 Illustration of remote sensing testbed

As alluded to in Sect. 2.1, a subordinate, yet important objective for ORACLES was the acquisition of data that can be used for the refinement and testing of retrieval capabilities for instrument concepts that have a potential for deployment to space. Among the remote sensing instruments participating in ORACLES, the following have a link to space-based instrument concepts: HSRL-2 (future HSRL in space), APR-3 (CloudSat's CPR, GPM's DPR, EarthCARE's CPR, and candidate radars for future missions targeting clouds, convection, and precipitation), RSP (APS on Glory), AirMSPI (MAIA), and eMAS (MODIS). Examples of algorithm developments using ORACLES data include Xu et al. (2018), Segal-Rozenhaimer et al. (2018) and Miller et al. (2019), who created new cloud retrieval


algorithms for the AirMSPI and RSP instruments. Additionally, the NASA Ames 4STAR instrument was used to

provide AERONET-like retrievals of aerosol microphysics on the basis of sky radiance measurements from an airborne platform (Pistone et al., 2019), the SSFR instrument provided within-atmosphere spectral radiative flux observations for direct measurements of scene and cloud albedos both from the P-3 and from the ER-2 in 2016 (Cochrane et al, 2019), and HSRL-2 successfully deployed a very stable and accurate density-tuned interferometer as a prototype for space-borne instrumentation, and leveraged the resulting high accuracy in the measurements to infer

aerosol microphysical properties (Burton et al., 2018). In combination, these instruments provide a powerful and unprecedented toolset for retrieving atmospheric trace gases, aerosol, and cloud properties, with simultaneous closure opportunities against the spectral radiative flux observations (see Fig. 20). The resulting data set will be used in the testing of instrument and algorithm concepts for future satellite missions, such as the NASA Aerosol, Clouds-Convection and Precipitation (ACCP) mission, recommended by the 2018 Decadal Survey for Earth Observations

from Space (National Academies of Sciences, Engineering, and Medicine, 2018) and the NASA Plankton, Aerosol, Cloud, Ocean Ecosystem (PACE) Mission (Werdell et al., 2019), due for launch in 2023.

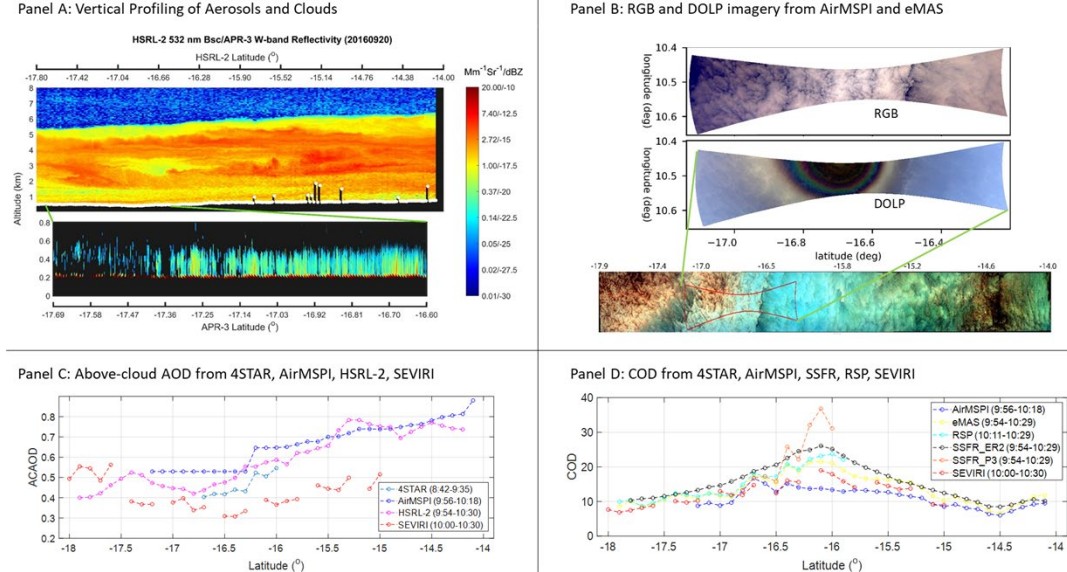

**Figure 20.** Illustrative remote sensing observations in ORACLES-2016. Panel A: Vertical structure of aerosol backscatter and W-band reflectivity from HSRL-2 and APR-3, respectively. Panel B: DOLP and RGB imagery from

AirMSPI (bowtie patterns) and eMAS, respectively. Panel C: Above-cloud AOD (ACAOD) retrievals from various P-3, ER-2, and satellite instruments. Panel D: COD retrievals from various P-3, ER-2, and satellite instruments.

**5.3 A revised schematic view of the system**

Based on the knowledge acquired thus far from ORACLES analysis, we present a revised schematic of the system (Fig. 21) compared with what was described in Sect. 2 as the state of knowledge during ORACLES conception and

planning. Smoke from biomass burning over the southern African continent in Austral winter and spring (July-





October) is emitted into a continental boundary layer that is potentially warmer, and therefore more buoyant, than the cold marine PBL to the west. As this smoke moves westward over the SEA, it must first cool to allow it to subside and be entrained into the PBL. In the free troposphere, clear sky longwave cooling by emission to space helps drive large-scale subsidence. This process is slow, with typical radiatively-driven subsidence rates of 200-400 m/day (Betts

and Ridgway 1988). Solar heating by absorbing aerosol may significantly slow down this process (Sakaeda et al. 2011). As the smoke moves initially westward over the SEA, it typically takes 6-10 days to subside from the jet core to the top of the marine PBL. Typical free-tropospheric wind speeds over the SEA in the southern African Easterly Jet (SAEJ) can reach 5 m/s, so smoke can travel hundreds to a few thousand km offshore in the time taken to descend from the main smoke outflow altitude of 4-5 km. However, FT wind speeds over the SEA are quite variable (see Fig.

15 in Adebiyi and Zuidema 2016) and are modulated by tropical wave disturbances and by incursions of midlatitude systems into the Tropics over the Southern Ocean. Depending upon winds in the FT, smoke trajectories may result in only modest horizontal displacement, or may (as in the case shown in the schematic Fig. 21) be advected as far south as 30° S, especially when midlatitude systems result in re-circulating trajectories that move air back toward the African continent after initial outflow. Thus, smoke aerosol may be entrained into the PBL over a relatively wide geographical

region.

Because subsidence is strongest off the coast of Namibia and southern Angola, this makes it a favored region for entrainment of smoke into the marine PBL, especially from recirculating trajectories like the one shown in Fig. 21. This set-up results in a typical (September) vertical plume structure in the ORACLES sampling region comprising relatively young aerosol aloft, more aged aerosol immediately above cloud, and the oldest aerosol typically in the

marine PBL itself. However, the extensive lateral displacement of the core, and a wind structure with significant vertical shear, means that the presence of smoke immediately above cloud top is intermittent. The PBL over the SEA therefore entrains a considerable volume of clean air, especially from the Southern Ocean. Thus, mean CO levels in the PBL are lower than 100 ppm, while FT plume mean values are typically 150-200 ppm (Shinozuka et al., 2020). With typical unpolluted background levels of 50-70 ppm, this suggests that the majority of the air in the PBL has

origins in regions other than those affected by biomass burning. Although this is the case, smoke plumes in the FT have very high aerosol levels (mass, concentration) compared with those in the pristine marine PBL (see Shinozuka et al. 2020), so even quite small mean elevations of CO in the PBL are associated with large aerosol perturbations.



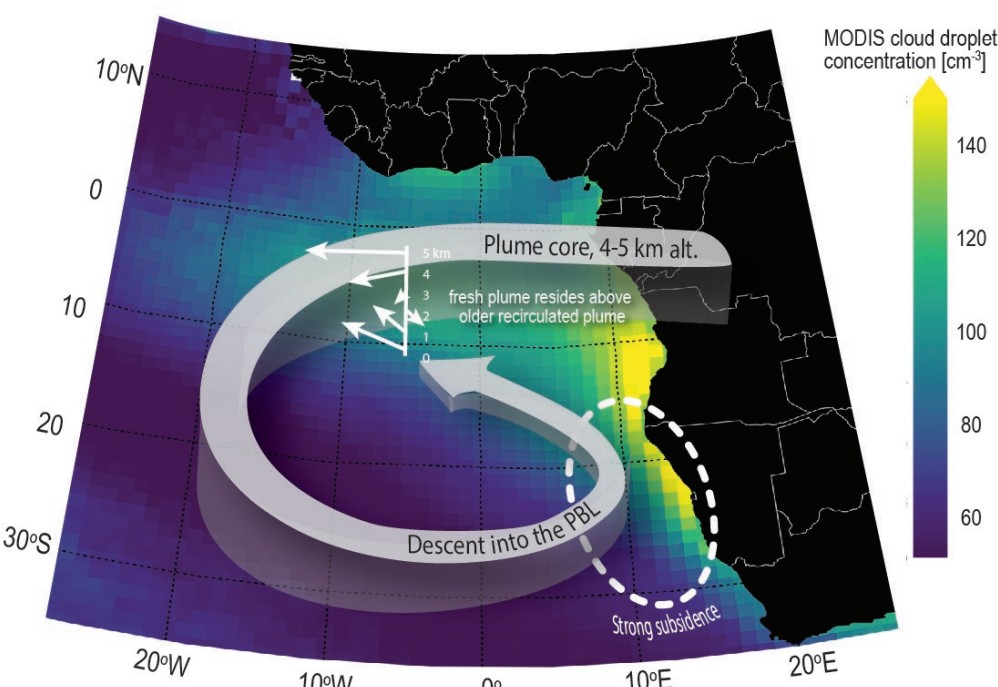

**Figure 21.** A schematic showing an example of a commonly occurring trajectory pathway into the marine PBL for
smoke aerosol ejected from southern Africa in the core (4-5 km altitude) of the African Easterly Jet South (AEJ-S).
This is overlaid onto an August-October mean climatology of cloud droplet concentration (colors) derived using
MODIS (Bennartz and Rausch, 2017). Trajectories over the SEA are variable. Depending upon the winds in the FT,
the trajectory may remain relatively static, or may (as in the case shown here) be advected as far south as 30° S, and
back toward the African continent by the incursion of midlatitude disturbances that lead to westerly lower tropospheric
winds extending into Tropical latitudes.

## 6. Discussion

As with any significant suborbital field deployment, we expect substantial data analysis efforts to extend well beyond
the nominal project end date. In this section, we describe ongoing analyses not previously mentioned. Future work
that may be facilitated by ORACLES data is captured in Sect. 7.

Science objective 1 on direct aerosol radiative effects (DARE) is being pursued with a number of different approaches.
At the finest spatio-temporal scale, these approaches entail instantaneous assessments of DARE on the basis of very
complete measurements of aerosol and cloud radiative properties from designated flight maneuvers on a particular
flight day. On the larger scales, these approaches combine geostationary satellite observations of diurnally varying
cloud properties and above-cloud AOD, both adjusted for aerosol radiative properties measured in situ and possibly





nudged by chemical transport model outputs, with campaign-average models of aerosol intensive properties. A designated group of ORACLES scientists meets routinely to discuss the results of the DARE assessment efforts and to avoid duplication of research efforts. Participation in these telecons can be requested through email to the corresponding authors of this manuscript.

Current work on science objective 2, the semi-direct aerosol effect, focuses on understanding how the vertical structure
in the SSA depicted in Fig. 14 relates to aerosol aging versus source composition (fuel type and flaming/smoldering conditions), and how the aging relates to aerosol transport patterns. Such work is primarily aimed at developing an understanding of the processes affecting the aerosol SSA and is cross-cutting across the different campaigns. Other work is examining how clouds adapt to variations in the absorbing aerosol vertical structure. The discrimination between cloud adaptations to the aerosol shortwave absorption, cloud-nucleating properties, and variability in the
large-scale circulation is inherently complex. While this work will assuredly require focused modeling activities that can control more easily for cause-effect relationships, ongoing analysis of ORACLES datasets will frame the modeling activities. In one example, the multiple routine flights along 5° E, conducted in both August 2017 and October 2018, are being analyzed to determine the dominant thermodynamic, dynamic, and aerosol features. In another example, remote sensing data from the ER-2 platform in September 2016 on the aerosol vertical structure and cloud properties
are being constrained by sea surface temperature and location to assess liquid water path responses to aerosol loading and vertical structure.

Science objective 3 focuses on effects of smoke CCN on the microphysical properties of clouds and precipitation. Concentrations of CCN in the PBL are much lower than those in the FT smoke plume because (a) subsidence and entrainment of smoke into the PBL is relatively slow; (b) there are significant losses in the PBL from both dry and
wet deposition. Work is currently underway to understand the transport pathways from the FT into the PBL using a combination of measurements of water vapor, tracers such as carbon monoxide (CO), and the isotopic composition of both water vapor and liquid water, to understand the smoke contribution to the PBL CCN budget. Cloud droplet concentration closure is being used to quantify the efficacy with which smoke aerosol serve as nuclei for cloud droplet activation. Remote sensing and in situ measurements are being used to understand how smoke both suppresses
precipitation and the extent to which precipitation is a removal process for CCN. The airborne polarimeter and in-situ observations are being used to provide constraints to correct satellite estimates of cloud droplet concentrations that may be affected by overlying smoke. Analyses to assess the Lagrangian evolution of clouds will help quantify how entrainment of smoke aerosol can impact Nd and change liquid water path and cloudiness. One such Lagrangian case, observed by both the P-3 and the UK BAe-146 aircraft over several days, involved a major smoke entrainment event,
and is currently being used to constrain large eddy simulations to evaluate both semi-direct and indirect aerosol effects.

All of the ongoing analyses on the overarching science objectives depend critically on the comparisons of observations and models described in Sect. 5.3.6. This dependence is manifested in two separate ways. The first, and more obvious way, is that the models are being evaluated for their ability to fill in missing pieces of information required for the
assessment of direct, semi-direct or indirect aerosol radiative effects in the SE Atlantic. If models are shown to provide reasonable predictions of relevant parts of the aerosol-cloud system, the predictions of these parts can be used to



extrapolate aerosol and cloud properties beyond the available spatial and temporal domains. The second, and slightly less obvious way, is that the models can be used to study the representativeness of the airborne and even some of the satellite observations. In the case of the airborne measurements, for example, the comparison of a model output

averaged over all time steps coincident with the measurements to the model averaged over all times in a given month may provide an assessment of how representative the relatively sparse airborne observations are for a monthly mean. If the two averages vary significantly, then we may conclude that the airborne observations should not be used directly, but instead should be adjusted by the model results, for the purpose of calculating monthly mean aerosol effects. Relatedly, the in situ measured aerosol properties should only be used to evaluate model results, if both the monthly

average model output and the output subsampled to the airborne observations show the same discrepancy to the model, as such a consistent discrepancy is attributable to a model deficiency rather than a sampling error.

Radiative closure studies will be equally important for progress on the overarching science questions. In a broad sense, these closure studies relate the aerosol and cloud properties measured in situ on the P-3 to remotely measured radiances or irradiances, either from the ER-2 (in 2016), from satellites, or from radiation measurements on the P-3 itself. They

may be as ambitious as exploring the connection between the measured aerosol chemical composition, aerosol and cloud radiative properties, and the radiation field, or simply comparing the aerosol radiative properties measured in situ to those measured from remote sensing instruments. Either way, the knowledge gained from these closure studies will be crucial for assessing large-scale aerosol-cloud-radiation interactions.

## 7. Conclusions and future work

The ORACLES project is a highly successful NASA EVS-2 project that had well surpassed its Level-1 baseline science requirements upon the conclusion of its last field deployment in October 2018. We conducted a total of three flight deployments totaling 350.6 science flight hours with the NASA P-3 aircraft and one flight deployment with the NASA ER-2 totaling 97.3 science flight hours, surveying, probing, and exploring the various features of the SE Atlantic aerosol-cloud-climate system. The ORACLES data set permits the study of aerosol radiative and cloud-

nucleating properties, their vertical distribution relative to clouds, the locations and degree of aerosol mixing into clouds, and cloud changes in response to such mixing. Here, we have only touched upon some of the key findings, leaving the detailed exploration of the various factors determining aerosol-cloud-climate interactions in the SE Atlantic to the individual investigations being conducted as part of, or spawned by, the ORACLES project. A high-level summary of some of the main scientific findings and conclusions from our project includes the following:

- the marine boundary layer of the SE Atlantic is more frequently affected by BB aerosol than previously thought, with the variety of pathways by which BB aerosol reach the BL complex and not yet fully explored (Zuidema et al., 2018); the transport and climate models used in ORACLES flight planning had limited success in forecasting the locations and levels of MBL aerosol pollution;

   - BB aerosol layers appear to be in much more frequent contact with the Sc cloud deck underneath than previously

estimated, but the correlation of cloud drop number concentrations with above-cloud smoke properties is weak (Diamond et al., 2018); in-situ data suggests that cloud droplet number and above cloud aerosol may be



anticorrelated, owing to the anticorrelation of MBL aerosol with above cloud aerosol amount (Kacarab et al., 2020);

- cloud vertical velocity tends to correlate positively with MBL aerosol number; this covariance tends to enhance cloud droplet number considerably beyond what is expected from aerosol changes alone (Kacarab et al., 2020); most of the observed droplet variability in clouds in polluted boundary layers may be driven by vertical velocity and its variability (Kacarab et al., 2020);

- the influence of BB aerosol on cloud microphysics in the MBL may be significant even if the aerosol number is low, as we find that the aerosol hygroscopicity is consistently less than 0.2, much lower than expected for pristine MBL aerosol (Wong et al., in preparation);

- interannual variations in the seasonal evolution of aerosol loading in the FT and cloud drop number concentrations in the MBL over the SE Atlantic during the BB season are greater than previously appreciated (see section 4.2 and 4.3); they are affected by interannual variations in BB location and strength and possibly by variations in the SAEJ (see Sect. 4.1, and Ryoo et al., in prep.);

- direct airborne measurements of the above cloud AOD have shown a slight overestimation by current remote sensing techniques from space borne instruments (MOD06ACAERO, Deep Blue MODIS and VIIRS) and some potential difficulties in cloud masking (LeBlanc et al., 2020; Sayer et al., 2019), but a generally consistent meridionally distribution between the airborne and orbital observations;

- aerosol radiative properties, such as the single scattering albedo and asymmetry parameter, especially in column-integrated values, show reproducible spectral dependence and a fairly well-constrained range of absolute values in each deployment year (Pistone et al., 2019; Cochrane et al., 2020); their vertical dependence appears to be reproducible as well (Doherty et al., in prep.);

- aerosol radiative properties appear to be driven by the chemical evolution of carbonaceous materials during plume aging (Dobracki, et al., in prep.), a process not previously investigated by suborbital means as it requires the sampling of smoke well beyond the near-field fire environment - this sampling of smoke properties after significant aging is a unique accomplishment of ORACLES;

- despite the large aerosol loadings, water vapor contributed significantly to the total heating rate at most altitudes (Mallet et al., 2019; Cochrane et al., 2020);

- there was ample evidence for aerosol-induced modifications of Sc cloud properties (Gupta et al., in prep.), and some evidence for the suppression of drizzle (Dzambo et al., 2019);

- chemical transport models and climate models exhibit a fairly systematic underestimation of aerosol loadings in the SE Atlantic (Shinozuka et al., 2019; Doherty et al., in prep.).

An analogous summary of lessons learned for logistics and planning of large field campaigns to address aerosol-cloud-climate interactions includes the following:

- the ORACLES science objectives and questions provided crucial guidance for flight execution in each of the deployment years;



- progress towards achieving science objectives was facilitated by the book-keeping of flight maneuvers that relate directly to the various detailed science objectives, although such book-keeping proved challenging in the field;
- the joint efforts in developing flight plans by scientists with nominally different flight objectives (e.g., cloud microphysics and radiation) revealed that relatively minor adjustments to flight maneuvers often resulted in data sets that are conducive to addressing a broad range of science questions, well beyond the benefits of flight plans that were developed by any one focused group in isolation;
- the scientific connection to international deployment efforts conducted by the CLARIFY and AEROCLO-sA teams in the same region and timeframe as ORACLES-2017 proved a worthy investment of time - while the connecting science is ongoing, there are already measurable outcomes from the international scientific collaborations in terms of the geographic extension of data sets and related publications (e.g., Mallet et al., 2019; Barrett et al., 2020);
- the routine sampling strategy for about half of the ORACLES flights allowed for a statistical assessment of climate and chemical transport models, unprecedented for suborbital efforts in this field of study;
- the involvement of the modeling community in the conception and development of the ORACLES project in general, and its flight planning specifically, proved to be invaluable for collecting a data set that can be used for addressing model deficiencies that hamper our ability to accurately simulate aerosol-cloud-climate interactions.

Future work will likely need to focus on aspects of the science that remain poorly understood and/or physics that is well understood but not well represented in models. Such topics will likely include long-term changes in biomass burning activity, convection and FT transport in the BB source region, the location and degree of mixing of BB aerosols into the SE Atlantic MBL, and the separation of synoptic-scale variations in the meteorological environment from aerosol-induced changes in Sc cloud properties, to name a few.

We conclude this paper with a figure that represents the complexity of the SE Atlantic aerosol-cloud system well. Figure 22 shows the longitudinal transect of the HSRL-2 scattering ratio curtain collected during the ER-2 transit from Recife, Brazil to Walvis Bay, Namibia, on August 26, 2016. In addition to expected features, such as an increasing MBL height with increasing distance from southern Africa, and the general dilution of the BB plume with distance from shore, the high signal-to-noise in the HSRL-2 data also reveals a complexity of the layering structure within the BB plume that was previously not appreciated. The layering includes horizontal and vertical gradients in loading, and likely in microphysics, with small-scale features that bear explanation. Future studies of the SE Atlantic climate system are well advised to embrace this complexity.



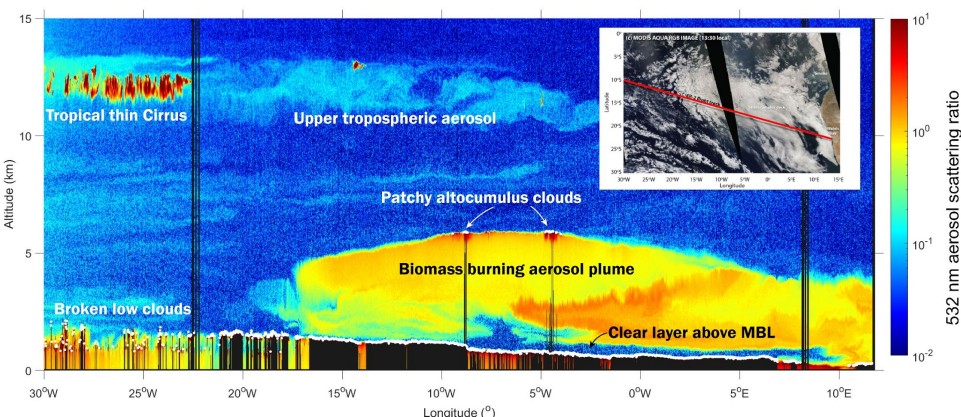

**Figure 22.** Longitudinal cross-section of the HSRL-2 scattering ratio curtain collected during the ER-2 transit from Recife, Brazil to Walvis Bay, Namibia, on August 26, 2016.

## 8. Appendices

### Appendix A - Description of flights

Tables A1a and A1b summarize the P-3 and ER-2 flights in 2016, respectively, while Tables A2 and A3 summarize the P-3 flights in 2017 and 2018, respectively. The entries for each flight are comprised of the flight date and identifiers (Format xRFnnYyy, where x is the platform indicator - P for the P-3 or E for the ER-2; RF is static and stands for Research Flight; yy is the flight number for this platform and year; Y is static and stands for Year, nn is the numerical year - 16, 17 or 18) in the first column. Column 2 indicates the total flight time. Column 3 contains a brief flight synopsis and numbers that indicate the number of specific flight maneuvers indicated in each flight.

**Table A1a.** 2016 flight summary for the P-3 aircraft. The number of each type of flight maneuver (as given in the table header) is shown in parentheses for each flight.

| Date P-3 flight # | Hrs | Flight synopsis (Ramps-RA / Square spirals-SS / MBL legs-ML / In-Cloud legs-ICL / Above-Cloud legs-ACL / Sawtooth legs-STL / In-Plume legs-IPL / Above-Plume legs-APL) |
|---|---|---|
| 08/27 PRF00Y16 | 6.8 | Transit, Ascension Island to Walvis Bay (RA 2 / SS 0 / ML 1 / ICL 0 / ACL 0 / STL 0 / IPL 0 / APL 2) |
| 08/30 PRF01Y16 | 1.6 | Routine (aborted): Upon takeoff, P-3 climbed through overcast stratocumulus; aircraft hydraulic issue was identified during take-off and the mission aborted. Some useful science measurements just offshore within the BB plume at approximately 3500m alt. (RA 1 / SS 0 / ML 0 / ICL 0 / ACL 0 / STL 0 / IPL 2 / APL 0) |





| 08/31<br>PRF02Y16 | 8.1 | Routine: To provide routine mapping along the NW-SE routine flight track from 23S/13E to as far NW as possible given flight time constraints; clouds were present along the entire routine track on the outbound leg, but by the return the clouds had a clear southern edge around 22S; typically, a gap was present between the elevated BB layer and the cloud, corroborated by low RH values associated with clean air just above cloud top.<br>(RA 7 / SS 2 / ML 4 / ICL 3 / ACL 4 / STL 1 / IPL 9 / APL 3) |
|---|---|---|
| 09/02<br>PRF03Y16 | 8.1 | Target: Sample aerosol radiative effects above clouds at 20S/10E where the aerosol plume and low-cloud deck are increasing towards the north and capture CF=100% case. Significant in-situ aerosol and cloud sampling, connection to remote sensing through Terra overpass. Required no Cirrus or mid-level clouds, high AOD, and solid Sc deck (all met). Performed two radiation walls near 16S because cloud conditions were optimal.<br>(RA 9 / SS 2 / ML 2 / ICL 2 / ACL 7 / STL 0 / IPL 7 / APL 3) |
| 09/04<br>PRF04Y16 | 8.0 | Routine: Objective was to reach 10S/0E along routine track with 2 profiles outgoing and returning, offset from each other. Reached 10S with two boundary layer profiles along the way. 2nd BL profile at 13S sampled a decoupled boundary layer reaching 1.5km with high organic/BC mass. The transit back sampled a thick mid-level cloud lying above the smoke layer, with similar aerosol/black carbon concentrations.<br>(RA 4 / SS 0 / ML 2 / ICL 2 / ACL 2 / STL 0 / IPL 8 / APL 8) |
| 09/06<br>PRF05Y16 | 8.0 | Target: Goal was to sample clouds and aerosols along a N-S line in what was anticipated to be three different aerosol conditions (aged aerosols on southern end, break in aerosols in the middle of the track and fresher aerosols towards the northern end of the track). Several -in- and above-cloud legs provided some evidence of forecast conditions. Drizzle evident in APR-3 data.<br>(RA 9 / SS 0 / ML 6 / ICL 4 / ACL 6 / STL 4 / IPL 4 / APL 4) |
| 09/08<br>PRF06Y16 | 8.1 | Routine: Plan was to reach 10S along routine track with a single profile out midway, then on return planned for at least two (2) profiles, followed by stepped ascents. Main objective was to quantify discrepancy in amount and vertical location of aerosol plumes between WRF and GEOS. Succeeded in doing 3 profiles. The aerosol layers were more complicated than modelled; clear slot between aerosol layers, with different aerosol composition in different layers. Some cloud work but mostly over scattered and few cloud areas.<br>(RA 9 / SS 0 / ML 3 / ICL 1 / ACL 4 / STL 1 / IPL 4 / APL 6) |
| 09/10<br>PRF07Y16 | 8.1 | Routine: Goal was a routine flight to 10S/0E, with planned profiling legs (below, within, above cloud, sawtooths through cloud and profiling tropospheric aerosols). Coordination with the ER-2 on the inbound leg. Some predicted Sc clouds did not materialize and some planned cloud work needed to be aborted. Aerosol conditions cleaner than predicted. Coordination between aircraft at 11:30UTC, with P-3 in cloud at exact ER-2 overpass time. |



| | | |
|---|---|---|
| | | (RA 6 / SS 2 / ML 4 / ICL 4 / ACL 4 / STL 1 / IPL 6 / APL 2) |
| 09/12 PRF08Y16 | 8.4 | Routine: Objective was to transit to north and west at high altitude, and then to conduct profiling on the return as time permitted. Loose coordination with the ER-2 on the inbound leg was envisioned. Almost reached 10S/0E and performed cloud work after descent. ER-2 coordination attempted at 18S/8E. In general, boundary layer profiles were quite clean, and clouds and precipitation were observed to reach the ground (APR-3 and in situ) towards the northern end of the track. The biomass burning plume was not being entrained into the cloud. (RA 6 / SS 0 / ML 3 / ICL 2 / ACL 2 / STL 1 / IPL 2 / APL 6) |
| 09/14 PRF09Y16 | 8.1 | Target: Radiation focus - radiation walls at two points with contrasting cloud/aerosol conditions; achieved multiple over-flights by ER-2 during two radiation wall segments on N-S legs near 16-17 S, but with moderate AOD (0.4). Clouds generally quite homogeneous and similar between radiation wall locations. Ultra-clean layers just above cloud top. Full square radiation spiral on leg A - preliminary results indicating significant albedo differences within spiral. (RA 11 / SS 1 / ML 4 / ICL 5 / ACL 7 / STL 0 / IPL 5 / APL 2) |
| 09/18 PRF10Y16 | 8.2 | Target: Young, dense plume studies. Increased likelihood of cirrus, in study region near 12-14S/11E. Predicted plume at about 4 km, only a few days old substantiated; lower parts of the plume were predicted to be older. Flew one extensive radiation wall, with 3 in-plume legs, 2 legs just above cloud, an extended cloud leg, an MBL leg, and one deep profile. 3 ER-2 overpasses captured. Mostly polluted, but not decoupled MBL (RA 3 / SS 1 / ML 2 / ICL 2 / ACL 2 / STL 0 / IPL 5 / APL 4) |
| 09/20 PRF11Y16 | 8.4 | Target: Radiation flight. Objective was to sample aerosol radiative effect, aerosol and cloud properties for two different types of cloud fields (in terms of albedo and/or cloud fraction) in coordination with the ER-2. Very successful flight - two almost complete radiation/ microphysics walls at 10.5E and 9E; mid-level clouds on 9E leg; BB plume reached highest altitudes so far (21kft) and largest AOD (~0.8); apparently fresh aerosol, absorption Ångström exponent higher than on other flights; plume more stratified than on other days; patches of drizzle found in radar and cloud probes. (RA 6 / SS 4 / ML 4 / ICL 3 / ACL 5 / STL 1 / IPL 2 / APL 4) |
| 09/24 PRF12Y16 | 9.2 | Target: Objective was another attempt to find the youngest, densest plume. Went farther north than any other flight, found very polluted layers at altitude. HSRL-2 data exceedingly useful for finding layers. Penetrated a couple of intermediate level clouds, able to get droplet size distributions. |





| | | |
|---|---|---|
| | | Confirmed high altitude plume predicted by WRF; GEOS-5 showed low altitude plumes that weren't there - excellent model testing! One ER-2 overpass. Spent about 30 min in clouds of various sorts. AOD up to 0.9.<br>(RA 5 / SS 3 / ML 4 / ICL 3 / ACL 3 / STL 0 / IPL 5 / APL 5) |
| 09/25 PRF13Y16 | 8.8 | Routine: Objective was to extend routine flight to 9hrs, and coordination with ER-2 at 10S/0E. Sampled BB layer at 14 and 18kft on outbound leg; profiles on outbound legs flown as planned, with extended low level legs. AOD 0.3-0.4, with the exception of 0.6 near turn-around point. Coordination with ER-2 at 13:45UTC, with P-3 in cloud (after a below-cloud leg and before above-cloud legs). Very clean layer above cloud top near 20S on return leg..<br>(RA 7 / SS 0 / ML 2 / ICL 2 / ACL 5 / STL 1 / IPL 8 / APL 5) |
| 09/27 PRF14Y16 | 7.3 | Transit, Walvis Bay to Ascension Island: Mostly high-altitude flight, with some in situ MBL observations and combined radar-in situ observations of precipitating clouds near Ascension Island starting at 11S/10W.<br>(RA 1 / SS 3 / ML 1 / ICL 0 / ACL 2 / STL 0 / IPL 1 / APL 2) |
| **15** | **115.2** | |

**Table A1b.** 2016 flight summary for the ER-2 aircraft. Flights with grey background were coordinated with P-3 aircraft operations.

| Date<br>ER-2 flight # | Hrs | Flight synopsis |
|---|---|---|
| 08/26 ERF00Y16 | 7.9 | Transit from Recife directly to Walvis Bay, very extensive longitudinal cross section with HSRL; RSP without SWIR, eMAS collected no data as plate installed over aperture; RSP, SSFR, AirMSPI, HSRL-2 worked well |
| 08/27 - 09/09: waiting for ER-2 fuel to arrive | | |
| 09/10 ERF01Y16 | 6.3 | Mapping Routine - Planned 9 hour flight to survey smoke transport; High level clouds in target area prevented original plan to work with P-3; RSP, AirMSPI, eMAS, SSFR worked well; HSRL-2 did not operate due to tripped ER-2 circuit breaker/faulty ER-2 coolant pump; flight shortened to attempt coordination with P3 and allow problem troubleshooting; RSP, AirMSPI, eMAS (including SWIR) data for cloud retrievals. |
| 09/12 ERF02Y16 | 8.7 | Routine: Planned 9 hour flight to survey smoke transport ("big triangle"); mid level clouds in target area prevented original plan to work with P-3; HSRL-2, RSP, AirMSPI, eMAS, SSFR worked well |
| 09/14 ERF03Y16 | 8.1 | Target: Planned 8hour flight with P-3 coordination along two N-S legs; RSP, AirMSPI, SSFR worked well; eMAS-VIS/NIR/SWIR worked well - LWIR compromised due to |





| | | Sterling cooler's active balancer failing during flight. HSRL-2 did not operate due to tripped aircraft circuit breaker. Troubleshooting found faulty aircraft coolant pump which was subsequently replaced; very good Terra overpass; Good flight for polarimeter cloud retrievals and intercomparison with P-3 RSP and in situ. |
|---|---|---|
| 09/16 ERF04Y16 | 7.7 | Routine: Mapping/survey flight "little triangle"; HSRL-2 operational again after ER-2 coolant pump replaced; HSRL-2, RSP, AirMSPI, SSFR worked well; eMAS - Good data in Vis-SWIR, no LWIR data available. |
| 09/18 ERF05Y16 | 8.5 | Target: Mapping plume & CALIPSO underflight; S-N leg near the coast (along 11 E) to look at smoke properties close to coast. Northern part of S-N leg (between 10-12 South, 11.5 E) included a that was coordinated with the P-3. Western part of plan included CALIPSO leg (overpass ~13:35 UT); HSRL, RSP, AirMSPI, SSFR, eMAS worked well. |
| 09/20 ERF06Y16 | 7.7 | Target: P-3 coordination, CALIPSO underflight; Coordinated S-N legs with P-3 along 10.5 E and 9 E between 14-18 South Western part of plan included CALIPSO leg; HSRL, RSP, AirMSPI, SSFR, eMAS worked well (eMAS - no 13.9 μm band data) |
| 09/22 ERF07Y16 | 7.9 | Routine & Mapping: Southern survey, St. Helena overflight; RSP, AirMSPI, SSFR, HSRL-2 worked well; eMAS–not operational (data system failure); Southern mapping triangle; Flyover of St. Helena: clouds prevented AERONET aerosol measurements; ER-2 overflight nearly coincident with St. Helena radiosonde launch. |
| 09/24 ERF08Y16 | 8.0 | Target: RSP, AirMSPI, SSFR, HSRL-2 worked well; eMAS–Good data in Vis-SWIR. No LWIR data (bands 26-38) - aircraft pod heater failure toward end of flight. ER-2 leg along 11E between 8-20 South along P-3 leg; ER-2 legs between 8-12 South used to study smoke evolution between 11 E and 3 E. |
| 09/25 ERF09Y16 | 8.7 | Routine: RSP, AirMSPI, SSFR worked well; HSRL-2 no science data due to laser problem; eMAS–Good data in Vis-SWIR, no LWIR data (bands 26-38); Aircraft pod heater failed. Flew "big triangle" and met P-3 on return leg. |
| 09/27 ERF10Y16 | 9.2 | Routine: RSP, AirMSPI, SSFR worked well; HSRL-2 did not collect science data due to laser problem; eMAS–Good data in Vis-SWIR. No LWIR data (bands 26-38) - aircraft pod heater failed. Short (10-15 min leg) on return leg for Aqua overpass for eMAS. Flew "big triangle" again. |
| 09/29 ERF11Y16 | 8.6 | Transit: Walvis Bay to Recife; Initial leg NW over standard leg to 0N/10 S, then over Ascension Island, before continuing to Recife. eMAS collected no data as plate installed over aperture; HSRL-2 did not collect science data due to laser problem. RSP, SSFR, AirMSPI worked well. |





| 12 | 97.3 | |
|---|---|---|

**Table A2.** 2017 flight summary for the P-3 aircraft.

| Date<br>P-3 flight # | Hrs | Flight synopsis<br>(Ramps-RA / Square spirals-SS / MBL legs-ML / In-Cloud legs-ICL / Above-Cloud legs-ACL / Sawtooth legs-STL / In-Plume legs-IPL / Above-Plume legs-APL) |
|---|---|---|
| 08/09<br>PRF00Y17 | 7.8 | Transit: Transit to São Tomé with science en route. Three square spirals, with backtrack at low altitudes. Two of the three with sawtooth cloud sampling, one in clear air with largest aerosol loading of flight. Most instruments worked well, highest AOD of 0.6, above clouds of about 0.46. Polluted MBL both below clouds and in cloud-free columns. No aerosol above clouds at Ascension. Interesting overall gradients.<br>(RA 7 / SS 0 / ML 3 / ICL 0 / ACL 2 / STL 2 / IPL 8 / APL 0) |
| 08/12<br>PRF01Y17 | 8.5 | Routine: Objective is to reach 13S. Multiple layers near São Tomé; highest ACAOD of flight at ~0.45. During S-bound transit, aerosol layer resting on cloud top, then decreasing cloud tops with separation from aerosol at about 2S. At ~6S - 'soft' cloud break then solid deck of small closed cell clouds topped by aerosol layer right on cloud tops. Two sets of spiral descents, cloud sawtooth patterns, sets of backtracking level legs for additional cloud sampling.<br>(RA 3 / SS 3 / ML 3 / ICL 1 / ACL 2 / STL 2 / IPL 5 / APL 2) |
| 08/13<br>PRF02Y17 | 9.1 | Target: Joint cloud-radiation flight, sampling gradient of overcast and broken clouds along the CALIPSO satellite track, aerosol radiative effects in presence of broken clouds, and mixing of aerosols into clouds. Transition between overcast and broken clouds was found along the 7-9˚S line oriented along A-train track. Line was oriented about parallel with surface winds. This allowed both the radiation and microphysics objectives to be addressed. Transition between homogeneous and broken clouds, and some gradient in the mixing mechanism into cloud and the boundary layer. Final part involved sampling at 20 kft during A-train overpass to get HSRL-2/RSP curtain/comparison.<br>(RA 4 / SS 2 / ML 3 / ICL 2 / ACL 3 / STL 1 / IPL 2 / APL 3) |
| 08/15<br>PRF03Y17 | 9.2 | Routine: Flight to 15S, with sampling of Lagrangian start points. Flight plan diverged slightly from that filed - flight altitude first limited due to the heavy fuel load. On way back, after waypoint 18, the 2.5km level leg was backtracked and a 3km level leg stacked on top of that. This was done because of concern about the low-altitude level legs disappearing into the boundary layer before they could be sampled on subsequent flight.<br>(RA 2 / SS 3 / ML 3 / ICL 1 / ACL 0 / STL 2 / IPL 9 / APL 2) |





| 08/17 PRF04Y17 | 9.1 | Suitcase flight São Tomé-Ascension Island: Objective was to resample airmasses from 4 horizontal legs in flight PRF03Y17. Flight path connects midpoints of legs sampled in PRF03Y17 on their 48hr trajectories - first, overflight of parcels for lidar sampling, then re-trace of track at projected parcel height after transport, then forward run along the same track near cloud top. Low clouds had largely cleared in the region between WPs 3 and 5, precluding cloud sampling. Due to significant interest in the BL Cu near ASI, the P3 headed west at 8S (rather than planned 9.7S). Plan was to sample scattered Cu along 8S between ~8W and Ascension. P3 sampled the BB plume to 7.5W, then descended into the MBL. Sawtooths through the boundary layer. Clouds were seen but time in cloud was insufficient for sampling. On approach to Ascension Island we overflew the ARM site. (RA 6 / SS 2 / ML 4 / ICL 1 / ACL 1 / STL 2 / IPL 8 / APL 2) |
| 08/18 PRF05Y17 | 5.5 | Ascension Island local: Coordinated flight with CLARIFY Bae146 to compare aerosol and cloud in situ, and radiation measurements. A highly successful coordinated flight, given the difficult cloud and aerosol forecasts. Bae146 assumed formation during initial climb-out to WP2. Cloud conditions were very broken except for leg between WP 4 and 5, making cloud comparisons limited. Lots of full boundary layer profiling between WP 10 and 11. Extended HSRL run over the ARM site (RA 2 / SS 0 / ML 2 / ICL 2 / ACL 0 / STL 1 / IPL 0 / APL 2) |
| 08/19 PRF06Y17 | 2.0 | Ascension Island-São Tomé: Aborted transit flight, limited set of instrumentation operated. (RA 0 / SS 0 / ML 0 / ICL 0 / ACL 0 / STL 0 / IPL 1 / APL 0) |
| 08/21 PRF07Y17 | 8.3 | Suitcase flight, Ascension Island to São Tomé: Objective was to measure west-to-east transition from aerosol mostly in the boundary layer. Clean above to heavier free troposphere pollution above (and mixing into) low cloud to the east. Also, sampling on Routine flight track. Aircraft maintenance issue reduced possible flight time to ~8hrs. Science focus was on three different plume and cloud regimes along 8S and the routine track (along 5E). Flight featured profiles in very different conditions; low clouds were more broken than forecast. Cirrus (Ci) on the 8S track (near 1E). A blob of mid-level clouds and high ACAOD (0.73) at ~8S/0E. (RA 3 / SS 3 / ML 3 / ICL 2 / ACL 4 / STL 1 / IPL 6 / APL 3) |
| 08/24 PRF08Y17 | 9.4 | Routine: Flight to 15S along 5E and back with some sampling of initial trajectory lines. High cloud contamination of remote sensing on the Northern part of the track. Solid deck at 15-10S, then small popcorn Cu to 5S, then Sc followed by mid-level cloud. Lightly polluted BL. SDI inlet froze. (RA 5 / SS 2 / ML 3 / ICL 3 / ACL 3 / STL 1 / IPL 7 / APL 3) |
| 08/26 PRF09Y17 | 9.7 | Target: Focus on radiation walls over broken cloud decks of varying albedos and relatively invariant aerosol. Targeted a region with broken low clouds, significant aerosol loading, |





| | | |
|---|---|---|
| | | and free of high clouds. Adjusted target area based on morning forecasts and satellite imagery high Ci north of 5S. Two successful radiation wall modules. Long transit prevented the third planned radiation wall. Many adjustments in flight for cloud conditions. During first radiation wall, low clouds were scattered to broken, no Ci. During second wall module, significant Ci streamer in the direct beam for center part of the legs. Good low clouds for most of wall. In-plume legs near 2S along the routine track.<br>(RA 2 / SS 4 / ML 2 / ICL 2 / ACL 3 / STL 0 / IPL 5 / APL 3) |
| 08/28<br>PRF10Y17 | 9.5 | Routine: Flight to 15S, with a simplified radiation wall, followed by cloud work and stacked aerosol sampling. Absence of Ci at 11S allowed good square spiral maneuver over mostly solid cloud. Max ACAOD of 0.76. Extensive cloud sampling.<br>(RA 3 / SS 1 / ML 1 / ICL 2 / ACL 3 / STL 2 / IPL 4 / APL 2) |
| 08/30<br>PRF11Y17 | 8.9 | Routine: Objective was routine flight, but not reaching as far south and allowing for a full radiation wall (less cloud work and more above-cloud legs than on 28 August flight) and include sampling of fresh aerosol and trajectory initialization points. HSRL-2 failure removed need for initial high-altitude leg; instead, sampled within the aerosol plume to 13S. Because of Ci, performed radiation wall at 8S. Sampled the plume south-bound between 4S and 10S at 3.5km altitude, and northbound between 7S and 3S at 3.0km altitude.<br>(RA 2 / SS 3 / ML 1 / ICL 1 / ACL 1 / STL 0 / IPL 8 / APL 2) |
| 08/31<br>PRF12Y17 | 8.3 | Target: Objective was to resample plume sampled on previous day at 3.5km and 3.0km between 4S and 10S along routine track. Plume was projected to be lower than 3km, but at 2.6km were below bottom of the plume. Got remote sensing measurements and cloud measurements in coordination with the A-Train overpass. At the northern end of the in-situ plume leg (corresponding to airmasses sampled at 3-5S on 30 Aug), did a series of stacked legs at 2.7km, 2.9km and 3.0km to check for vertical variations in aerosol properties.<br>(RA 2 / SS 2 / ML 1 / ICL 1 / ACL 1 / STL 0 / IPL 12 / APL 2) |
| 09/02<br>PRF13Y17 | 8.7 | Transit, São Tomé to Ascension Island: objective was to Measure BB aerosol at northern end of study area, possibly affected by wet convection; get AERONET-like retrieval from 4STAR in a mix of biomass burning smoke & dust, supplemented with in-situ and HSRL measurements. Also, sample near ASI where previously sampled airmasses may be present. Got one radiation spiral each without and with some dust present. Near ASI, high-altitude HSRL legs, plume leg and above-cloud leg. Series of legs to study cloudy region just to NE of ASI,<br>(RA 4 / SS 6 / ML 1 / ICL 1 / ACL 5 / STL 1 / IPL 6 / APL 4) |
| **14** | **114.0** | |






**Table A3.** 2018 flight summary for the P-3 aircraft.

| Date<br>P-3  # | Hrs | Flight synopsis<br>(Ramps-RA / Square spirals-SS / MBL legs-ML / In-Cloud legs-ICL / Above-Cloud legs-ACL / Sawtooth legs-STL / In-Plume legs-IPL / Above-Plume legs-APL) |
|---|---|---|
| 09/24<br>PRF00Y18 | 9.3 | Transit, Cape Verde to São Tomé: Mostly transit flight at high altitude, but some in situ sampling near São Tomé.<br>(RA 0 / SS 2 / ML 1 / ICL 1 / ACL 1 / STL 0 / IPL 1 / APL 2) |
| 09/27<br>PRF01Y18 | 8.0 | Routine: Flight to along 5E to 13S. High-altitude transit out, square spiral down at 13S, followed by 3 samples of the cloudy boundary layer on the way back north, the most northern one being at ~5S. Strong aerosol layering south of 5S, fairly clean to the north. Little aerosol right above cloud top.<br>(RA 3 / SS 2 / ML 2 / ICL 1 / ACL 1 / STL 3 / IPL 4 / APL 2) |
| 09/30<br>PRF02Y18 | 7.7 | Target: Objective was radiation work near 7-9S in radiation wall patterns over broken cloud decks of varying albedos and (nominally) relatively invariant aerosol. Coordinated with MISR local mode. Square spiral near 7.5S, well coordinated with MISR overpass, in an area of solid low cloud cover. Hit CALIPSO overpass for RSP; got on their track ~8min after overpass.<br>(RA 4 / SS 1 / ML 3 / ICL 0 / ACL 1 / STL 2 / IPL 5 / APL 4) |
| 10/02<br>PRF03Y18 | 8.5 | Routine: Flight to 10S, setup of BL Lagrangian sampling in PRF04. At approximately 6.7S, found a transition from closed cells to pockets of open cells (POCs). AOD ~0.45. Square spiral at 10.5S. Boundary layer quite clean with a few big particles, low CN, CO < 70 ppb and O3 < 20 ppb. A series of 2.5 dull sawtooths, with clean slot right above the cloud (~200 feet). Several constant altitude legs with high CCN.<br>(RA 3 / SS 3 / ML 2 / ICL 1 / ACL 3 / STL 2 / IPL 3 / APL 4) |
| 10/03<br>PRF04Y18 | 8.5 | Target: Lagrangian resampling of POCs sampled on PRF03; Closed cells present on the transit from São Tomé to 5E, 7.5S. East of 6E along 7.5S, a large region of open cells with significantly lower cloud fraction; FT plume was extensive aloft above both the POC and the surrounding closed cells. Clear evidence of smoke aerosol being present immediately above clouds in closed and open cell regions.<br>(RA 4 / SS 4 / ML 2 / ICL 1 / ACL 2 / STL 2 / IPL 4 / APL 2) |
| 10/05<br>PRF05Y18 | 9.0 | Target: Radiation at high solar zenith; radiation wall between 5.5E and 7W on 9.5S; incl. high-altitude overpass; square spiral from ~6km to surface in mostly-cloudy conditions; above-cloud, in-cloud, below-cloud leg; 3 vertically-stacked in situ sampling legs in the plume. Second similar maneuver in almost clear conditions. Boundary layer most polluted so far in ORACLES 2018. Cloud droplet number concentrations accordingly elevated. |





| | | |
|---|---|---|
| | | Appeared to be a more aged plume than other days in ORACLES 2018. Good case for radiative closure: two square spirals in different conditions, the full radiation wall, moderate RH in the plume. (RA 4 / SS 3 / ML 4 / ICL 2 / ACL 3 / STL 0 / IPL 5 / APL 5) |
| 10/07 PRF06Y18 | 8.4 | Routine: Flight to 15S. During transit to 15S north-south slope in low cloud top heights from 2S to 7S with sloping aerosol layers above. From 10S to 15S mid-level clouds at the top of the outflow plume, embedded in the plume. Boundary layer work between 12S and 9.5S - fairly polluted BL. Extended run at 8kft, Clear slot at 9S for square spiral and radiation work. This spiral happened in the most cloud-free conditions encountered in ORACLES-2018. Extended leg (1hr+) at 8kft, during transit home. (RA 6 / SS 2 / ML 3 / ICL 1 / ACL 2 / STL 1 / IPL 4 / APL 5) |
| 10/10 PRF07Y18 | 8.3 | Routine: Flight to 13S along 5 E. 3 samples of polluted boundary layer allow for possibility of Lagrangian sampling on 12 October. Square spiral to nearly to surface at 13S. Cloud sampling south of 10S. Sampling aerosol layer at 13kft between 9S and 10S. Long boundary layer sequence, starting at 7.5S. Aerosol and boundary layer work near 4.5S. Aerosol at southern end of track up to 19.5 kft. (RA 6 / SS 3 / ML 4 / ICL 2 / ACL 3 / STL 2 / IPL 4 / APL 3) |
| 10/12 PRF08Y18 | 5.3 | Target: Lagrangian follow-up and cloud profiling. Engine issue delayed departure; flight duration shortened by ~3 hours. Outbound transit straight south of São Tomé to initial point at 2.5S, 6.5E where trajectory indicated resampling of boundary layer air. Square spiral at 2S/5.75E. Boundary layer sampling, followed by sawtooth sampling through decoupled cloud layers (stratus above Cu). Cloud patch thicker and precipitating at the north end of the runs. Quite clean conditions just above cloud. More cloud work at 4.5 ° S, 5.5 ° E. Ensuing run northbound crossed distinct boundary between clear and polluted air, with corresponding changes in cloud properties. (RA 2 / SS 2 / ML 3 / ICL 1 / ACL 5 / STL 2 / IPL 2 / APL 1) |
| 10/15 PRF09Y18 | 7.8 | Routine: Flight to 14S. During southbound transit, minimal direct contact between smoke and Sc at 3-9S. Heavily precipitating Sc clouds at 5S. Two regions with plume bottom/cloud top gap (8.5S) and no gap (9.5S) in relative close proximity. Significant drizzle between 11 and 13S. Square spiral at 14S; geometrically thin high smoke loading layer at 13kft, broken Sc. Very clean BL, low cloud bases 500ft. During sawtooth N-bound, clouds thickening, peak above-cloud smoke at 12.4S, dropping to the N. Second set of sawtooth patterns contrasting 9.5 and 8.5S – smoke near cloud top at 9.5 and gap at 8.5S. Circular spiral descent at 5.5 S. (RA 1 / SS 2 / ML 2 / ICL 0 / ACL 3 / STL 2 / IPL 4 / APL 4) |



| 10/17 PRF10Y18 | 8.5 | Target: Young plume near 7S/10E. At 10.5E, 7S still mid-level clouds. Opted to change order radiation wall/spiral module, aerosol in-situ legs first (highest altitude layer to lowest altitude layer); then below, in and above-cloud legs; then square spiral up. Highest aerosol concentrations encountered yet, found in the lower free troposphere. (RA 0 / SS 5 / ML 1 / ICL 1 / ACL 2 / STL 0 / IPL 9 / APL 5) |
|---|---|---|
| 10/19 PRF11Y18 | 8.0 | Target: Objective was to sample young plume near-coast 7S/10E. Moved north-south sampling further north and west to avoid high cirrus. Evidence of very clean air directly above clouds. Did not find fresh plume near Blight of Angola, ACAOD only ~0.27. Two square spirals near each other, one over clear skies, the other over partially cloudy. The bottom of one square spiral had a ship and its plume. Clean MBL, and ultra-clean above clouds. Underflight of partial cloudy skies – high potential of 3D cloud-aerosol radiative effect. Good in situ sampling, while in cloud. (TDMA+CVI), indication of large aerosols. High cloud drop number concentrations near bottom of clouds (opposite of what has been observed in the past). (RA 5 / SS 4 / ML 3 / ICL 1 / ACL 5 / STL 1 / IPL 7 / APL 6) |
| 10/21 PRF12Y18 | 8.2 | Routine: During S-bound transit, FT BB aerosol plume tendril-like structure with numerous layers overlapping; Interesting wave-like structure in low clouds from 1.9-3S, with wavelength of ~100 km; evidence of N-S mesoscale banding in Sc cloud layer below; at 13.5S - square spiral to 200 ft. Plume concentrations all around half of typical values. Ultra-large particles detected on descent. Aerosol well-aged in lower part of plume. Multiple saw-tooth patterns for cloud-work and level legs in plume. (RA 8 / SS 1 / ML 6 / ICL 2 / ACL 2 / STL 2 / IPL 6 / APL 5) |
| 10/23 PRF13Y18 | 8.1 | Target: Survey flight going west along 5S. Only flight with significant boundary layer cloud sampling to the west of 5E. 4 sawtooths through double-layered stratocumulus in which the lower layer cloud droplet number concentrations exceeded those in the upper layer. Generally low ACAOD of 0.19 max. (RA 2 / SS 2 / ML 3 / ICL 0 / ACL 4 / STL 3 / IPL 4 / APL 3) |
| 10/25 PRF14Y18 | 7.8 | Transit - São Tomé to Cape Verde; Survey flight going west. High-altitude along 5E to 5S, then turn west, out to 3W along 5S. Three samples of the cloudy boundary layer on the way back. Four sawtooths through double-layered stratocumulus; lower layer cloud droplet number concentrations exceeded those in the upper layer. Square spiral at 3W. (RA 1 / SS 1 / ML 1 / ICL 0 / ACL 0 / STL 0 / IPL 0 / APL 3) |
| **15/15** | **121.4** | |

Figure A1 shows the distribution of flight times dedicated to the various flight maneuvers described in Table 5 in each of the three ORACLES deployments. While broadly similar, there are a few notable distinctions: 2017 and 2018





featured significantly fewer ramp descents, relatively less time just above cloud top, and more time dedicated to square spiral descents. Also, 2017 and 2018 entailed more sawtooth profiling through clouds than time spent in level legs within clouds. These changes represent an evolution in the thinking regarding the best flight maneuvers to address various cloud- and radiation-related science objectives from the first to the second and third deployment.

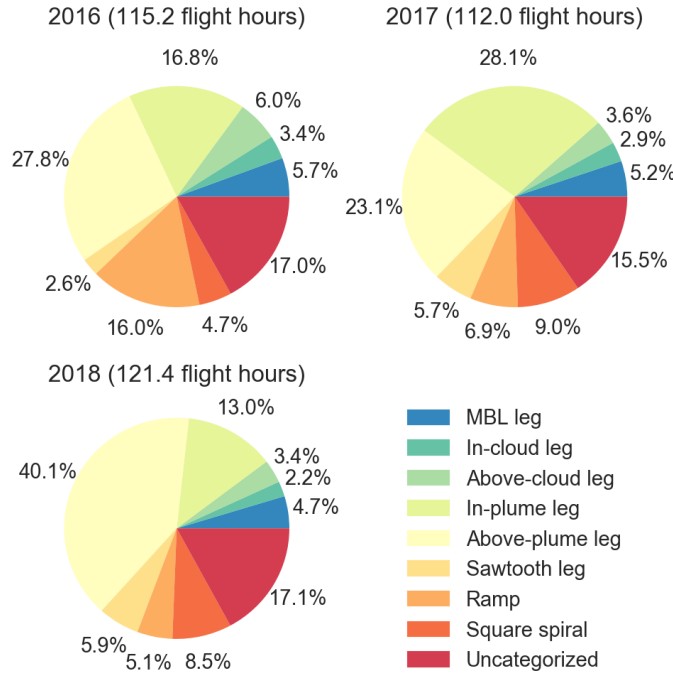

**Figure A1**. Distribution of flight time between flight maneuvers for each ORACLES deployment year.

**Appendix B - Choice of Instrumentation**

**Table A4.** P3 instrumentation in ORACLES (bold entries indicate quantities submitted to the ORACLES archive at: https://espoarchive.nasa.gov/archive/browse/oracles).

| Instrument name / operating organization | Instrument description / key specification | Primary measurement | Measurements / Derived quantities / Inversion products |
|---|---|---|---|
| **Remote sensing** | | | |
| 4STAR / NASA ARC | Hyperspectral sun-/sky-photometer (400-1600nm, ~1nm res.) | 1. Direct solar beam hyperspectral transmittance | 1. **Spectral AOD** |
| | | | 2. **Column O$_3$, H$_2$O, NO$_2$** |
| | | 2. **Sky radiance** | 3. **Aerosol microphysics(e.g., size** |



| | | | distribution, refr. index, absorption, scattering phase function) |
|---|---|---|---|
| | | 3. **Hyperspectral cloud zenith transmittance** | 4. **Cloud Optical Depth, $r_{eff}$, Liquid Water Path**, thermodynamic phase |
| RSP/NASA GISS | Measurements at 410, 470, 555, 670, 865, 960, 1590, 1880, 2260 nm with polarimetric accuracy of ~ 0.15% | **Stokes parameters I, Q and U of reflected light** | 1. **Aerosol microphysics, layer height and AOD** |
| | | **Measurements are over ±60° from nadir** | 2. **Water COD, droplet sd at top, bulk effective radius, top height, physical thickness, $N_c$** |
| | | | 3. **Chl, CDOM conc. and backscatter coeff.** |
| AMPR / NASA MSFC | Advanced Microwave Precipitation Radiometer 4-frequency (10.7, 19.35, 37.1, and 85.5 GHz), cross-track scanning, polarization-variable microwave radiometer | **Polarized Brightness Temperatures** | 1. Precipitation Rate |
| | | | 2. Liquid water path |
| | | | 3. Ocean SST, Winds |
| SSFR, CG-4 / NASA ARC, CU LASP | Solar Spectral Flux Radiometer (350-2100 nm shortwave irradiance, spectral sampling 4-8 nm) CG-4 (longwave irradiance 4-40 μm) | **Spectral Solar Irradiance** | 1. Cloud and aerosol radiative effects |
| | | | 2. Spectral and broadband absorption and heating rate, aerosol SSA from flux divergence |
| | | | 3. Cloud phase, OD, reff (from albedo and transmittance) |
| APR3 | 3-Frequency Cloud and Precipitation Doppler Scanning Radar (Ku, Ka an W-band) | 1. Cloud and Precipitation backscatter | **Rain water content** and **Precipitation Rate** |
| | | 2. Cloud and Precipitation differential backscatter | Hydrometeor Size (Precipitation class) |
| | | 3. Cloud and Precipitation Doppler velocity | Hydrometeor classification (Dominant) |
| | | 4. Path integrated Attenuation | Vertical Air Velocity in precipitation |
| **Cloud in situ** | | | |



| CAPS/ UND | 1.Cloud Imaging Probe - Optical Array Probe (25-1600 µm, 25 µm res.) | Cloud particle images | **Number distribution function, nominally between 25-1600 µm**, particle images from which other parameters can be derived: **total concentration, liquid water content, etc.** |
|---|---|---|---|
| | 2. Cloud and Aerosol Spectrometer (CAS, forward scattering) (0.53-50 µm, 1 µm nominal res.) | Number size distribution | **Number distribution function between 0.53 and 50 µm**, from which liquid water content, effective radius and other parameters can be derived |
| | 3. Liquid Water Content (LWC) sensor (0-3 g/m3), not operational | Liquid water content | Bulk liquid water content |
| CDP/ UND | Forward scattering (2-50 µm, ~2 µm res.) | Number size distribution | **Number distribution function between 3 and 50 µm**, from which liquid water content, effective radius and other parameters can be derived |
| CDP/LARC | Cloud Droplet Probe Forward scattering (2-50 µm, ~2 µm res.) | Number size distribution | |
| CDP/HiGEAR | Cloud Droplet Probe Forward scattering (2-50 µm, ~2 µm res.) | Number size distribution | |
| King/ UND | Hot wire liquid water (0-5 g/m$^3$) | Liquid water content | **Bulk liquid water content** |
| 2-DS/ UND | 2-Dimensional Stereo Probe Optical array probe (10 - 1280 µm, 10 µm res.) | Cloud particle images | **Number distribution function, nominally between 10-1280 µm** and particle images from which other parameters can be **derived (total concentration, liquid water content**, etc.) |
| HVPS-3/ UND | High Volume Precipitation Spectrometer Optical array probe (150 - 19200 µm, 150 µm res.) | Cloud particle images | **Number distribution function, nominally between 150-19200 µm**, and particle images from which **total concentration and rain water content** can be derived |
| FPDR-PDI/ Univ. Hawaii | Cloud droplet size and velocity measurements | Droplet size and arrival time | **1. Droplet size (µm) & arrival time (µs)** |





| | | | 2. $N_c$ (#/cm³) |
| --- | --- | --- | --- |
| | | | 3. Derived LWC (g/m³) |
| | | | 4. Droplet Velocity (m/s) |
| **Aerosol in situ** | | | |
| HiGEAR/Univ. Hawaii | TSI 3321 APS (0.8 to 5 μm aerodynamic) DMT UHSAS (70 to 1000 nm optical) Modified TSI long SMPS (10 - 550 nm) Custom TSI thermal tandem SMPS (10 to 200 nm) | **Dry number size distributions** | **Number, area, volume distributions,** CCN concentration (indirect) |
| | | **Particle volatility** | |
| | TSI 3025A Ultrafine CN counter (1-3000 nm) | **Total particle concentration** | |
| | TSI 3010 CN counters (3-3000 nm, ambient and denuded to 400ºC) | Total particle concentration | Internal/external mixing |
| | TSI 3563 3 wavelength nephelometers Paired Radiance Research M901 nephelometers, one with humidity-controlled inlet. 2 Radiance Research 3 wavelength PSAPs | **Dry particle scattering coefficient, backscattering, @ 450, 550, 700 nm** **Wet vs dry scattering @ 550 nm** **Particle light absorption** | **SSA, Scat. Angstr. Exp., Abs. Angstr. Exp., Extinction (@470, 530, 660 nm)** |
| | DMT SP2 (Single Particle Soot Photometer, 4-channel, 90 - 500 nm) 2016 only | **Refractory black carbon concentration,** | **BC concentration** |
| | | **Refractory black carbon mass** | **BC mass distribution**, BB tracer |
| | Aerodyne HR-ToF-Aerosol mass spectrometer (AMS) | **Non-refractory aerosol composition** | **Sulfates, nitrates, organics, chloride, BB tracer**, pollution tracer |



| | | | |
|---|---|---|---|
| | | | |
| PCASP/ UND | Passive Cavity Aerosol Spectrometer Probe | Forward scattering (0.1-3 μm, ~0.1 μm res.) | **Aerosol number distribution function between 0.1 and 3 μm**, from which total concentration can be derived . |
| AFS/NASA ARC | Aerosol Filter System, collecting various filters for offline analysis (2017-18) | | TEM-EDX/SEM-EDX analysis for single particle size, mixing state and elemental composition, bulk BrC, and bulk soluble ions. |
| PTI/BNL | 2016, 2018: Photothermal Interferometer (532nm) 2017-2018: DMT Single Particle Soot Photometer (SP2; 8-channel, refractory black carbon particle mass 80 nm - 500 nm, mass equivalent diameter) | **Aerosol light absorption at 532 nm; Refractory Black carbon (rBC) particle mass** | 1. **Absorption coefficient (Mm$^{-1}$)**<br><br>2. **Refractory Black carbon (rBC) mass loading (ng/m$^3$) and number size distributions** |
| GIT CCN instrument | CCN concentrations / spectra at cloud-relevant supersaturation | CCN concentration (0.15-0.6%) water vapor supersaturation | CCN spectra, cloud effective supersaturation, aerosol hygroscopicity, droplet growth kinetics. |
| **Gases in situ** | | | |
| COMA/NASA ARC | Trace gas detector | **in-situ measurement of gas phase CO and H2O** | 1. **CO mixing ratio** 2. **H2O mixing ratio** |
| WISPER/OSU Water Isotopes | In situ gas phase cavity ring-down water vapor isotopic analysers (Picarro model L2120-fi) coupled to isokinetic and CVI inlets. | **1. Total H$_2$O mixing ratio, $\delta^{18}$O and $\delta$D.** | 1. Cloud droplet and rain evaporation proportion |
| | | **2. Condensed water content (liquid+ice) (g/m$^3$), $\delta^{18}$O and $\delta$D.** | 2. Bulk air mass mixing state and entrainment rate. |
| | | **3. CVI enhancement for residual aerosol.** | 3. Cloud base and cloud top water mass flux when combined with winds |
| **Winds** | | | |





| Vertical Winds / NASA Langley | Vertical Winds calculated from 5 hole flush port radome system / aircraft inertial navigation system | 1. **fast response (20 hz) vertical winds** | 1. if combined with another fast response measurement, can provide vertical fluxes of that species |


Table A5. ER-2 instruments in ORACLES 2016

| Instrument name / operating organization | Instrument description / key specification | Primary measurement | Measurements / Derived quantities |
|---|---|---|---|
| **Remote sensing** | | | |
| eMAS (Enhanced MODIS Airborne Simulator) | 38-channel multi-spectral line-scanner | Solar reflective and thermal emissive energy in the 0.46-14 micron range | **Cloud optical properties (phase, optical thickness, effective radius, and water path); cloud top properties (temperature, pressure, height, and infrared phase)[a]** above-cloud AOD |
| RSP/NASA GISS | Measurements at 410, 470, 555, 670, 865, 960, 1590, 1880, 2260 nm with polarimetric accuracy of ~ 0.15% | **Stokes parameters I, Q and U of reflected light Measurements are over ±60° from nadir** | 1. **Aerosol microphysics, layer height and AOD from inversion** |
| | | | 2. **Water COD, droplet sd at top, bulk effective radius, top height, physical thickness, $N_c$** |
| | | | 3. **Chl, CDOM conc. and backscatter coeff.** |
| AirMSPI / JPL | Multiangle radiometric/polarimetric imager with bands centered at 355, 380, 445, 470*, 555, 660*, 865*, 935 nm (*polarimetric). | 1. Upwelling radiances (multispectral, multiangle, spatial) | **Liquid cloud droplet effective radius and COD[b]** Aerosol optical depth, particle size distribution, single scattering albedo, refractive index |
| | | 2. Stokes polarization components (Q, U) (multispectral, multiangle, spatial) | |
| SSFR/ NASA ARC, CU LASP | Solar Spectral Flux Radiometer (350-2150 nm, spectral sampling 4-8 nm) | 1. **Spectral Solar Irradiance** | 1a. TOA aerosol radiative effect (BB only) 1b. Aerosol heating rate (w/P3, BB only) 1c. TOA cloud-aerosol radiative effect 1d. Scene incident |



| | | | |
|---|---|---|---|
| | | | irradiance |
| HSRL-2 / NASA LaRC | Multi-wavelength High Spectral Resolution Lidar | **Particulate extinction (355 nm, 532 nm)** **Particulate backscatter (355 nm, 532 nm, 1064 nm)** **Particulate depolarization (355 nm, 532 nm, 1064 nm)** | **Aerosol classification,** aerosol mixing layer height (~ PBL height)**, AOD Aerosol microphysics from inversion (e.g. N,S,V concentrations, effective radius)** |

ᵃeMAS-derived quantities, along with L1B data, are archived and publicly available at the LAADS DAAC (https://ladsweb.modaps.eosdis.nasa.gov/), not the ORACLES archive.

ᵇRSP-derived quantities are publicly available at https://eosweb.larc.nasa.gov/project/airmspi/.


**Table A6.** Ground-based observations supported by ORACLES

| Ground-based | | | |
|---|---|---|---|
| AERONET | Spectral sun and sky ground-based radiometer (340 to 1640 nm) | 1. Direct solar beam transmittance; 2. Sky radiance; Cloud zenith transmittance | 1. spectral AOD, column H2O 2. Aerosol microphysics from inversion (e.g., sd, refr. index, absorption) 3. Cloud phase, OD, reff (inversion) |

**Appendix C - Acknowledgement of all participants**

**Table A7.** Participants in the ORACLES project, 2014-2019, not co-authoring this paper.

| Last Name | First Middle | Organization |
|---|---|---|
| Adebiyi | Adeyemi A. | University of Miami |
| Alexandrov | Mikhail | NASA Goddard Institute for Space Studies, Columbia University |
| Allison | Quincy | Bay Area Environmental Research Institute |
| Alugodhi | Mercy-Thea | |
| Anderson | Bruce Eldon | NASA Langley Research Center |
| Arnold | George Thomas | NASA Goddard Space Flight Center, Science Systems and Applications, Inc. |
| Barrett | Paul Alan | UK Met Office |



| Barrick | John | NASA Langley Research Center |
|---|---|---|
| Bauer | Susanne | NASA Goddard Institute for Space Studies, Columbia University |
| Beach Jr. | Harry Lee | AMA Inc |
| Bennett | Joseph Ryan | National Suborbital Research Center |
| Biswas | Sayak Krishna | Aerospace Corporation, NASA Marshall Space Flight Center |
| Broccardo | Stephen Paul | NASA Ames Research Center |
| Cantrell | Alvin Eric | NASA Marshall Space Flight Center |
| Carnes | Steve Raymond | Jet Propulsion Laboratory |
| Chirica | Dan C. | Bay Area Environmental Research Institute |
| Chowdhary | Jacek | NASA Goddard Institute for Space Studies, Columbia University |
| Chun | William | Jet Propulsion Laboratory |
| Clarke | Antony David | University of Hawaii |
| Colarco | Peter Richard | NASA Goddard Space Flight Center |
| Cook | Anthony L | NASA Langley Research Center |
| Dahlgren | Robert Paul | California State University, Monterey Bay |
| Das | Sampa | NASA Goddard Space Flight Center |
| Delaney Jr. | Michael McFadyen | National Suborbital Research Center |
| Delene | David | University of North Dakota |
| Dobrowalski | Gregg Charles | Jet Propulsion Laboratory |
| Dominguez | Roseanne | University of California, Santa Cruz |
| Drouet | Jeffrey Thomas | University of Colorado, Boulder |
| Dunagan | Stephen | NASA Ames Research Center |
| Dunwoody | Kent Nelson | NASA Armstrong Flight Research Center, USRA |
| Durden | Stephen L. | Jet Propulsion Laboratory |
| Eck | Thomas Frank | NASA Goddard Space Flight Center |
| Ellis | Thomas Ashly | University of California, Santa Cruz |
| Everson | Chad Michael | University of North Dakota |
| Fenn | Marta Angeline | NASA Langley Research Center, Science Systems and Applications, Inc. |
| Finch | Patrick Eugene | Bay Area Environmental Research Institute |



| Fraim | Eric | NASA Ames Research Center, Universities Space Research Association |
|---|---|---|
| Garay | Michael Joseph | Jet Propulsion Laboratory |
| Garland Jr. | Rebecca Maureen | Council for Scientific and Industrial Research |
| Geogdzhayev | Igor | NASA Goddard Institute for Space Studies, Columbia University |
| Giles | David Matthew | NASA Goddard Space Flight Center, Science Systems and Applications, Inc. |
| Grant | Patrick Steven | NASA Ames Research Center, University of California, Santa Cruz |
| Gray | Ellen Theresa | NASA Goddard Space Flight Center |
| Hambloch | Patrick | University of Alabama in Huntsville |
| Harper | David B. | NASA Langley Research Center |
| Heikkila | Ashley Creta | University of Hawaii |
| Henze | Dean | Oregon State University |
| Hildum | Edward Ames | Universities Space Research Association |
| Howes | Calvin Tucker | University of California, Los Angeles |
| Ibrahim | Hani Halim | SpecTIR LLC |
| James | Mark W. | NASA Marshall Space Flight Center |
| Johnson | Roy Robert | NASA Ames Research Center |
| Johnson | Matthew | NASA Ames Research Center |
| Jordan | David Everet | NASA Ames Research Center |
| Kangueehi | Ismael | Stellenbosch University |
| Karol | Yana | NASA Ames Research Center |
| Kindel | Bruce | University of Colorado, Boulder |
| Kittelman | Alan Scott | University of Colorado, Boulder |
| Kraft | Jason Michael | NASA Goddard Space Flight Center |
| Klopper | Danitza | North-West University |
| Lebsock | Matthew David | Jet Propulsion Laboratory |
| Lee | Joseph William | NASA Langley Research Center |
| Lightbourn | Frank | NASA Armstrong (Dryden) Flight Research Center |
| Mace Jr. | Gerald Grant | University of Utah |





| Yang Martin | Meiying Melissa | National Suborbital Research Center |
|---|---|---|
| McFadden | Susan Kimi | NASA Ames Research Center, Bay Area Environmental Research Institute |
| McIlhattan | Elin Arwen | University of Wisconsin–Madison |
| Miller | Rose Marie | University of Illinois at Urbana-Champaign |
| Miller | Daniel John | NASA Goddard Space Flight Center / UMBC Joint Center for Earth Systems Technology |
| Mohrmann | Johannes | University of Washington |
| Nathanael | Benjamin | National Commission on Research, Science & Technology (Namibia) |
| Nghiyalwa | Hilma | Geography Department, University of Namibia |
| Nicholas | Sommer Lynne | Bay Area Environmental Research Institute |
| Olson | Jennifer | NASA Langley Research Center |
| Ottaviani | Matteo | NASA Goddard Institute for Space Studies, Terra Research Inc |
| Purdue | Sara Kisa | University of Miami |
| Quigley | Emmett | NASA Ames Research Center |
| Rheingans | Brian Eugene | Jet Propulsion Laboratory |
| Rodriguez Monje | Raquel | Jet Propulsion Laboratory |
| Schafer | Joel Shannon | NASA Goddard Space Flight Center |
| Schaller | Emily Lauren | National Suborbital Research Center |
| Seidel Caprez | Felix Clemens | NASA Headquarters, now JPL |
| Shimhanda | Senior | Kiyushu Institute of Technology, Japan |
| Shingler | Taylor | NASA Langley Research Center, Science Systems and Applications, Inc. |
| Shipepe | David Michael | Namibia University of Science & Technology |
| Simmons | David Earl | University of Alabama in Huntsville |
| Sims | W. Herb | University of Alabama in Huntsville |
| Sinclair | Kenneth Allan | Columbia University |
| Sinyuk III | Aliaksandr | NASA Goddard Space Flight Center |
| Slutsker | Ilya | NASA Goddard Space Flight Center |
| Smirnov | Alexander | NASA Goddard Space Flight Center, Science Systems and Applications, Inc. |



| Smirnow | Nikolai Brown | University of Hawaii |
|---|---|---|
| Sorokin | Mikhail Grigorievich | NASA Goddard Space Flight Center, Science Systems and Applications, Inc. |
| Stamnes | Snorre | NASA Langley Research Center |
| Swap | Robert John | University of Virginia, NASA Headquarters |
| Tan | Qian | Bay Area Environmental Research Institute |
| Thompson | Andrew | Bay Area Environmental Research Institute |
| Tosca | Mika | School of the Art Institute of Chicago |
| van Diedenhoven | Bastiaan | NASA Goddard Institute for Space Studies, Columbia University |
| Van Gilst | David Patrick | National Suborbital Research Center |
| van Harten | Gerrit | Jet Propulsion Laboratory |
| Vasques | Marilyn | NASA Ames Research Center |
| Wasilewski | Andrzej Piotr | SciSpace, NASA Goddard Institute for Space Studies |
| Williams | Brent Allan | Bay Area Environmental Research Institute |
| Winchester | Cody | University of Hawaii |
| Xu | Feng | JPL, now University of Oklahoma |
| Yates | Brian Avery | NASA GSFC Wallops Flight Facility, Pinnacle/AMOC |
| Zavaleta | Jhony Ronald | NASA Ames Research Center |
| Zhang | Jianhao | University of Miami |
| Zhang | Qin | Bay Area Environmental Research Institute |


## 9. Author contributions

JR, RW, and PZ, designed the original ORACLES observational concept, and co-led the 5-year investigation. SJD was instrumental in conducting the ORACLES investigation. BL coordinated all project management. JMH and PF led the CLARIFY and AEROCLO-sA field experiments, respectively, and their coordination with ORACLES
deployments. SEL, MSD, YS, RY, JMR, AND, AMdS, KML. MSK, CJF, KP, ASA, SEB, AMF, GRC, PES, GAF, SF, BC, BH, KDK, ST, TSL, AMD, OOS, GMM, MRP, SG, JRO, AN, MK, JPSW, JDS-G, KLT, DN, JRP, KSS, PP, HC, SPC, AJS, TJL, ES, MS-R, RAF, SPB, CAH, DJD, SP, JSM, KM, DAS, made critical contributions to the field deployments and post-campaign data analyses. SGH made vital contributions to the experimental design and execution. MM and PS led formal modeling analyses in the post-campaign stage. NMK and SJP coordinated the
outreach and scientific efforts by Namibian and South African partner institutions. IYC and LG created and visualized





results for portions of this manuscript. H. Maring provided crucial administrative support to enable ORACLES field operations.

## 10. Competing interests

PZ, PF, and JMH are guest editors for the ACP Special Issue "New observations and related modelling studies of the
aerosol–cloud–climate system in the Southeast Atlantic and southern Africa regions", that this manuscript is submitted to. The remaining authors declare that they have no conflicts of interest.

## 11. Acknowledgments

ORACLES is a NASA Earth Venture Suborbital 2 investigation, funded by NASA's Earth Science Division and managed through the Earth System Science Pathfinder Program Office. The ORACLES team gratefully acknowledges
the work by the NASA Ames Earth Science Project Office (ESPO), led by Dr. Bernadette Luna and Mr. Dan Chirica. The team is equally grateful for the tireless contributions by the NASA Wallops and NASA Johnson P-3 and ER-2 pilot and flight crews, as well as air traffic control at Walvis Bay airport (Namibia) and the airport in São Tomé. Local authorities in Namibia and São Tomé beyond the ones mentioned in Sect. 3.3.7 and 3.5 played important roles as well, for which the project would like to express their gratitude. P.S. acknowledges funding from the UK NERC project
CLARIFY (NE/L013479/1) and the European Research Council (ERC) project RECAP under the European Union's Horizon 2020 research and innovation program with grant agreement 724602.

The ORACLES project, in spirit and execution, was very much a collaborative effort. It entailed contributions small and large, with many of the smaller and short-term contributions equally crucial to the execution of the project. In
Table A7 in Appendix C we list all participants in the ORACLES project that are not co-authors of this paper. The ORACLES leadership, and indeed NASA as a whole, owe sincere gratitude to any and all of the participants listed there.

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
