# Peer review of "An overview of the ORACLES (ObseRvations of Aerosols above CLouds and their intEractionS) project: aerosol-cloud-radiation interactions in the Southeast Atlantic basin"

_Atmospheric Chemistry and Physics, 2020_

## Referee Comment (RC1) · Armin Sorooshian (Referee) · 30 Jun 2020

This work represents a review of the (ObseRvations of Aerosols above CLouds and their intEractionS) ORACLES project. This expedition took place over Southern Africa, where nearly a third of the planet's biomass burning aerosol particles are produced, with the potential to interact with a large stratocumulus deck over the Southeast Atlantic Ocean. Through a series of three Intensive Observation Periods, numerous findings were obtained focusing on three major topics: a) direct aerosol radiative effects; (b) effects of aerosol absorption on atmospheric circulation and clouds; (c) aerosol-cloud

microphysical interactions. The nature of a paper like this is to not get too deep into any one finding but to familiarize the community with the project and its dataset, in addition to pointing interested readers to more specific studies pertaining to an individual topic if they desire more depth. The paper is a challenging one to design and write as there is so much content for an experiment of this large scale to try to capture in one place; the authors did a good job of summarizing pertinent details and pointing interested readers in the right direction if they want more details about any one specific topic area. I do recommend publication of this work and provide some suggestions for improvement below. Finally, I congratulate this science team for an excellent job done when considering the high degree of difficulty. I look forward to the future science results that still require time to be developed and published.

Comments: Abstract, Lines 75-76: The list of three topics does not qualify in my view as being "science questions" as they are not written as questions. I suggest re-wording.

Line 92-93: when describing the Twomey effect, I suggest to clarify this is at "fixed liquid water" conditions.

Line 103: is "aerosol" considered to be singular since the word "contains" is used? I would assume it would be plural.

Figure 1: Line 120 suggests there is a site called "Principe" that I am not sure I see labeled in Figure 1. Figure 1 generally looks busy and not the easiest to see all the features; I understand a lot of work went in to make this figure though so likely no need to change this unless the authors also feel it is too complicated.

Figure 2: The "See text." addition at the end of the caption seems unnecessary to me. Also, while the blue and red bars are easy to make sense of, the yellow ones seem like a bit too much especially since they are presumably hidden at times behind blue bars. I leave it up to authors to decide whether to break out the yellow bars on a separate axis or figure; perhaps it is desirable to not have more figures and thus they can ignore my suggestion.

Lines 193-198: It would have been nice if the number of science goals listed here (2) mapped on directly to the number of science questions in the abstract (3). Again, up to authors to decide if this is worth addressing or not. It is good to see that the 3 science questions/topics in the abstract map on to Table 1 well.

Line 314: Sounds off to just say "CCN for cloud condensation nuclei". Is the "CCN" supposed to be the instrument name (presumably the CCN counter)? I do not think that "CCN" is the full name of the instrument. Please check.

Line 596 and 863: change "further" to "farther"

-I especially enjoyed reading Section 3.5 and applaud the team for excellent outreach efforts.

Figure 11: I wonder if having a color other than light blue would help with contrast since there is another shade of blue.

Table 3: Nice strategy to provide this. Great idea.

Figure 14: quite challenging to see the text, especially in the bottom 3 panels. Please improve aesthetic quality and the ability of readers to see the information clearly.

Line 883-884: What is the reference for this claim about the "f44" metric? I ask since the reference to "up to ten days" is quite specific and I am curious what study showed that result.

Line 885-886: Interestingly, a 2-part paper series in JGR was inspired by ORACLES to see how similar aerosol-cloud interactions and smoke plume properties would be based on aircraft work off the US West Coast (https://doi.org/10.1029/2019JD031159 and https://doi.org/10.1029/2018JD029134); I leave it up to the authors to decide if it is worth mentioning that one of the many values of the ORACLES dataset is contrasting it with smoke impacting some of the other major SCu decks, such as what was explained in great detail by those two papers above in JGR. In this capacity, I found it interesting that the f44 values reported for smoke in the Mardi et al. (2019) study were very similar

to those in Figure 15, which may be worth mentioning to go along with the Siberian plume comparison. This 2-part paper series interestingly has a lot in common with ORACLES results in other areas, especially those reported by Diamond et al. (2018).

Line 920-921: Note that the paper here is cited for the year 2010 (presumably the JGR paper about precip susceptibility) but the one in the reference list is 2009. Since I am familiar with both papers, I admit either one would work well, but to save yourself time, stick with the one in the reference list (2009 GRL).

Figure 20: I suggest removing the panel titles above each panel figure since they are already in the caption. This is a tough figure to see in terms of clarity and font size. If anything can be done to improve it for the final draft, that would be great since it is a really nice figure idea.

Line 1200-1201: I suggest another word other than "unprecedented". I can think of multiple other campaigns that have applied routine sampling strategies highlighted below (albeit others likely exist). Perhaps I did not interpret the sentence correctly and the authors meant to say something more specific about how their routine strategy was unprecedented and different than other studies like the ones described below? https://doi.org/10.1029/2019JD032346 https://doi.org/10.1175/BAMS-D-18-0100.1

Line 1159: Authors should be more specific about what is actually less than "0.2". What specific hygroscopicity parameter? In fact, I don't remember reading about this in the main body of the paper. I suggest sticking to the policy of not adding new information to the Conclusion section that wasn't reported on in the main body of the paper. In this regard, I was particularly hoping to hear more about the aerosol composition and hygroscopicity in the paper, and especially what was learned from the CVI measurements; I assume the authors prefer readers to look at specific papers about these results, which makes sense.

-The Wong et al reference is missing from the reference list.

[Figure]

Figure 3: can the caption more clearly state the meaning behind "440" and "441" on the y-axes labels?

Table A1a: Some of the entries are a bit difficult to understand; for instance, in PRF03Y16 it say "…and capture CF=100% case". Presumably this means their was interest in sampling in an area of 100% cloud fraction, but I am not sure this was described as clearly as I know it could be. I suggest re-reading some of these entries to make them a bit more clear to those who are relatively newer to the field of airborne science like a beginning graduate student who would want to make sense of these entries. Editing the entries would clean up little issues like a double period at the end of the PRF13Y16 entry.

Table A4: Why are some entries bolded and others are not? Also, there are some acronyms that should be defined such as "droplet sd".

---

## Referee Comment (RC2) · Anonymous Referee #2 · 13 Aug 2020

This paper is a valuable and very comprehensive summary of the ORACLES field campaign. I particularly appreciate the effort to document key elements of the field campaign planning and implementation, including elements that may be innovative. This knowledge is rarely documented in the literature, and therefore only conveyed to the lucky few that get to participate in or observe the management of a complicated field campaign. Some minor revisions, particularly in the presentation of some of the preliminary science results, may be warranted. However, the paper is certainly suitable for publication in ACP. Some detailed comments follow.

The first four sections of the paper are lengthy, but very informative. I wish that the background and context for more of these large field campaigns were so thoroughly documented in the literature. However, the conclusions drawn from figure 9, which are very important to the broader context of the experiment, could be more concretely supported. The notions that the "fire counts in the three deployment years are very similar to the climatologies" and that the analysis "supports the conclusion of an earlier and possibly prolonged presence of the biomass burning plume . . . in recent years" are only supported with small color/contour maps, when the quantitative data presented in those charts could actually be interpreted with proper statistics to determine the magnitudes of any differences between them and how the relative differences or similarities compare to the magnitude of interannual variability. Rather than asking the reader to eye-ball the differences and interpret subtle differences in shading or contour shapes, why not actually reduce the data to statistics that support the conclusion?

At the very end of section 4.3 is a brief summary of the broad conclusions for how the aerosol and cloud properties during the months of the field campaign relate to climatologies and interannual variability, but no conclusion is drawn for what this means for the outcome of the field campaign. Do the authors feel they captured representative conditions from their sampling? Do the differences from climatologies noted for August 2017 and October 2018 have any implications for the resulting dataset in terms of whether the results drawn from the data can be thought of as broadly representative of aerosol/cloud relationships in the region?

The discussion in the paragraph starting in line 820 raises the question of whether any future planned satellite lidar instruments will have the signal-to-noise ratio to provide a better view of the frequency of clear air between aerosol and cloud layers? Can the results of ORACLES help constrain a future mission so that past inferences about the relationships between aerosol and cloud layers can be improved upon with large statistics from satellites in the future?

The discussion between lines 861 and 865 is confusing and seems to be missing some

key elements to the interpretation of the observed SSA and BC:OA ratio. From figure 14, Aug. and Sept. appear similar with Oct. being the outlier, but the discussion draws a distinction, in terms of 4 km winds, between Aug. and the other months. Furthermore, I was not clear on exactly what the higher winds in Sept./Oct. are responsible for. Finally, if I am understanding the argument correctly, the aerosol lower in the column is older and also exhibits lower SSA and higher BC:OA ratio. However, the discussion does not link these aerosol properties to age? Am I to conclude that aging depletes the organic fraction of the aerosol, and that is why the SSA decreases with altitude?

The results presented in figure 18 are intriguing and certainly demand further study. I understand that there is a paper in progress to do just that (Gupta et al. 2020). However, it is rather disappointing to read that there are hypotheses that might explain the results, but not be able to read what they are. Is it not possible to share the hypotheses and then note that they are to be evaluated in the other paper?

Conversely, there are a number of declarative statements in the bulleted list in section 7 of the paper that are not supported in the paper and the supporting citations are to papers that are not yet published. While I understand that the notion here is to summarize some of the findings from the campaign as they presently stand, this does raise a question of whether it would be proper to have this paper on the record citing some declarative results that may either not end up in the peer-reviewed literature, or be altered somewhat after peer-review.

---

## Author Comment (AC1) · 9 Oct 2020

Authors' response to interactive comments on "An overview of the ORACLES (ObseRvations of Aerosols above CLouds and their intEractionS) project: aerosol-cloud-radiation interactions in the Southeast Atlantic basin" by Jens Redemann et al.

Reviewer 1: Armin Sorooshian (Referee), armin@email.arizona.edu

We greatly appreciated Dr. Sorooshian's review of our paper. His praise for certain

parts of the manuscript means a lot, given his PI role for an ongoing Earth-Venture-Suborbital project. Below, we have taken the liberty to number Dr. Sorooshian's comments and attempted to respond to the best of our abilities; we hope that our responses meet with his approval.

Detailed comments:

1. Comments: Abstract, Lines 75-76: The list of three topics does not qualify in my view as being "science questions" as they are not written as questions. I suggest re-wording.

Response: Agreed. We changed the word "questions" to "themes".

2. Line 92-93: when describing the Twomey effect, I suggest to clarify this is at "fixed liquid water" conditions.

Response: We added "at fixed liquid water content" before the Twomey reference.

3. Line 103: is "aerosol" considered to be singular since the word "contains" is used? I would assume it would be plural.

Response: We changed the noun to "aerosols".

4. Figure 1: Line 120 suggests there is a site called "Principe" that I am not sure I see labeled in Figure 1. Figure 1 generally looks busy and not the easiest to see all the features; I understand a lot of work went in to make this figure though so likely no need to change this unless the authors also feel it is too complicated.

Response: This confusion derived from the fact that the country is indeed called "São Tomé and Príncipe", but we were operating from the island of São Tomé. We decided to remove the word "Príncipe" from line 120 so as to eliminate confusion and not clutter Figure 1 with longer labels.

5. Figure 2: The "See text." addition at the end of the caption seems unnecessary to me. Also, while the blue and red bars are easy to make sense of, the yellow ones

seem like a bit too much especially since they are presumably hidden at times behind blue bars. I leave it up to authors to decide whether to break out the yellow bars on a separate axis or figure; perhaps it is desirable to not have more figures and thus they can ignore my suggestion.

Response: We deliberated this carefully amongst lead authors - we would prefer to keep the yellow bars as they speak directly to the physical separation of BB and cloud layers referred to later in the manuscript. We eliminated the words "See text" per the recommendation.

6. Lines 193-198: It would have been nice if the number of science goals listed here (2) mapped on directly to the number of science questions in the abstract (3). Again, up to authors to decide if this is worth addressing or not. It is good to see that the 3 science questions/topics in the abstract map on to Table 1 well.

Response: The overarching questions here were meant to encompass the more specific science questions Table 1. We have tried to clarify by changing the text to read "The overarching ORACLES science goals, which encompass the specific science themes and questions in the abstract and Table 1 below, are:..."

7. Line 314: Sounds off to just say "CCN for cloud condensation nuclei". Is the "CCN" supposed to be the instrument name (presumably the CCN counter)? I do not think that "CCN" is the full name of the instrument. Please check.

Response: We agree and added the word "spectrometer" to read: "...CCN spectrometer for cloud condensation nuclei..."

8. Line 596 and 863: change "further" to "farther".

Response: Done. Thanks.

9. -I especially enjoyed reading Section 3.5 and applaud the team for excellent outreach efforts.

Response: Thank you.

10. Figure 11: I wonder if having a color other than light blue would help with contrast since there is another shade of blue.

Response: We have changed the light blue tracks to red.

11. Table 3: Nice strategy to provide this. Great idea.

Response: Thank you.

12. Figure 14: quite challenging to see the text, especially in the bottom 3 panels. Please improve aesthetic quality and the ability of readers to see the information clearly. Response: We have increased the font and changed the location for the text in this figure.

13. Line 883-884: What is the reference for this claim about the "f44" metric? I ask since the reference to "up to ten days" is quite specific and I am curious what study showed that result.

Response: We added text that specifies that this statement referred to figure 15. We have addressed the comment on specificity by adding "approximately".

14. Line 885-886: Interestingly, a 2-part paper series in JGR was inspired by ORACLES to see how similar aerosol-cloud interactions and smoke plume properties would be based on aircraft work off the US West Coast (https://doi.org/10.1029/2019JD031159 and https://doi.org/10.1029/2018JD029134); I leave it up to the authors to decide if it is worth mentioning that one of the many values of the ORACLES dataset is contrasting it with smoke impacting some of the other major SCu decks, such as what was explained in great detail by those two papers above in JGR. In this capacity, I found it interesting that the f44 values reported for smoke in the Mardi et al. (2019) study were very similar to those in Figure 15, which may be worth mentioning to go along with the Siberian plume comparison. This 2-part paper series interestingly has a lot in common with ORACLES results in other areas,

especially those reported by Diamond et al. (2018).

Response: We agree with the interesting similarity between the absolute values in f44 between our study and the Mardi et al. paper. However, the latter does not make a quantitative connection between f44 and physical smoke age, and we would hence prefer to not reference the Mardi paper here.

15. Line 920-921: Note that the paper here is cited for the year 2010 (presumably the JGR paper about precip susceptibility) but the one in the reference list is 2009. Since I am familiar with both papers, I admit either one would work well, but to save yourself time, stick with the one in the reference list (2009 GRL).

Response: Change made.

16. Figure 20: I suggest removing the panel titles above each panel figure since they are already in the caption. This is a tough figure to see in terms of clarity and font size. If anything can be done to improve it for the final draft, that would be great since it is a really nice figure idea.

Response: We removed the panel titles from this figure and did our best to improve legibility.

17. Line 1200-1201: I suggest another word other than "unprecedented". I can think of multiple other campaigns that have applied routine sampling strategies highlighted below (albeit others likely exist). Perhaps I did not interpret the sentence correctly and the authors meant to say something more specific about how their routine strategy was unprecedented and different than other studies like the ones described below? https://doi.org/10.1029/2019JD032346 https://doi.org/10.1175/BAMS-D-18-0100.1

Response: We added text to specify what the term unprecedented referred to: "...unprecedented for suborbital efforts at geographical scales of up to 2,000km (Shinozuka et al., 2020);".

18. Line 1159: Authors should be more specific about what is actually less than "0.2".

What specific hygroscopicity parameter? In fact, I don't remember reading about this in the main body of the paper. I suggest sticking to the policy of not adding new information to the Conclusion section that wasn't reported on in the main body of the paper. In this regard, I was particularly hoping to hear more about the aerosol composition and hygroscopicity in the paper, and especially what was learned from the CVI measurements; I assume the authors prefer readers to look at specific papers about these results, which makes sense.

Response: We removed the bullet point with the conclusion about the kappa parameter (which was referred to with the numerical value of 0.2) and eliminated all references to papers that have not yet been submitted. Fortunately, except for the bullet point with the kappa parameters, all other conclusions are supported by figures in this paper and added references to sections 4.2, 4.3, 5.2.2 and 5.2.3 and Figure 18, where necessary.

19. -The Wong et al reference is missing from the reference list.

Response: We removed this reference.

20. Figure 3: can the caption more clearly state the meaning behind "440" and "441" on the y-axes labels?

Response: The "440" and "441" referred to the wavelengths of the AERONET retrievals. We have changed the y-axis labels to read "AOT (440/441nm)" and "SSA (440/441nm)", respectively, and added the wavelengths in the caption.

21. Table A1a: Some of the entries are a bit difficult to understand; for instance, in PRF03Y16 it say "...and capture CF=100% case". Presumably this means their was interest in sampling in an area of 100% cloud fraction, but I am not sure this was described as clearly as I know it could be. I suggest re-reading some of these entries to make them a bit more clear to those who are relatively newer to the field of airborne science like a beginning graduate student who would want to make sense of these entries. Editing the entries would clean up little issues like a double period at the end

of the PRF13Y16 entry.

Response: This point is well taken - we were trying to strike a balance between highlighting the main components of the flights and keeping our statements brief. This resulted in some entries that were difficult to comprehend. We have made numerous changes in all four table (A1a, A1b, A2, and A3): for all flight entries, we added a statement describing the primary science objective or at least the general plan for the flight; we added verbs to make some sentences more comprehensible, and, where short-hand notation was not conducive to the reader's understanding, we expanded the short-hand notation to full sentences; we replaced most jargon and acronyms; and we fixed a number of punctuation issues beyond the one mentioned in this review comment.

22. Table A4: Why are some entries bolded and others are not? Also, there are some acronyms that should be defined such as "droplet sd".

Response: As the table caption indicated, the bold entries refer to measurements that were archived in the official ORACLES archive. We have changed the Table caption to clarify this: "P3 instrumentation in ORACLES (bold entries indicate quantities submitted to the ORACLES archive, see ORACLES Science Team (2020), in the list of references)."

---

## Author Comment (AC2) · 9 Oct 2020

Authors' response to interactive comments on "An overview of the ORACLES (ObseRvations of Aerosols above CLouds and their intEractionS) project: aerosol-cloud-radiation interactions in the Southeast Atlantic basin" by Jens Redemann et al.

Anonymous Referee #2

We greatly appreciated this reviewer's scrutiny of our paper. We feel that their comments stem from great knowledge and experience with investigations similar to OR-

[Figure]

ACLES. We have taken the liberty of numbering their comments and attempted to respond with the diligence that the thoughtfulness of the comments inspired.

Detailed comments:

1. The first four sections of the paper are lengthy, but very informative. I wish that the background and context for more of these large field campaigns were so thoroughly documented in the literature. However, the conclusions drawn from figure 9, which are very important to the broader context of the experiment, could be more concretely supported. The notions that the "fire counts in the three deployment years are very similar to the climatologies" and that the analysis "supports the conclusion of an earlier and possibly prolonged presence of the biomass burning plume ...in recent years" are only supported with small color/contour maps, when the quantitative data presented in those charts could actually be interpreted with proper statistics to determine the magnitudes of any differences between them and how the relative differences or similarities compare to the magnitude of interannual variability. Rather than asking the reader to eye-ball the differences and interpret subtle differences in shading or contour shapes, why not actually reduce the data to statistics that support the conclusion?

Response: A more careful analysis indicated that the statement about "... an earlier and possibly prolonged presence of the biomass burning plume ...in recent years" is overly simplistic and that it depends strongly on the observational period considered. As the analysis of historical context is beyond the scope of this overview paper and somewhat tangential, we decided to just remove the sentence on burning season length from the discussion following figure 9 and leave the analysis to a future publication. Our added text in response to this reviewer's suggestion 2 below partially addresses the context issue.

2. At the very end of section 4.3 is a brief summary of the broad conclusions for how the aerosol and cloud properties during the months of the field campaign relate to climatologies and interannual variability, but no conclusion is drawn for what this means
for the outcome of the field campaign. Do the authors feel they captured representative conditions from their sampling? Do the differences from climatologies noted for August 2017 and October 2018 have any implications for the resulting dataset in terms of whether the results drawn from the data can be thought of as broadly representative of aerosol/cloud relationships in the region?

Response: We have added the following text to the end of section 4.3: "Taking all three years together, mean ACAOD values were slightly lower than their climatological mean values for August-October and Nc values slightly higher than climatology. These deviations are substantially smaller than the day-to-day variability sampled in the three campaigns, and we therefore consider the ORACLES measurements to have captured representative conditions overall. Detailed assessment of the representativeness of the actual aircraft observations, which only sampled a fraction of the days within each of the three measurement months, has been undertaken in the model-observation intercomparison studies (Shinozuka et al., 2020). This assessment indicates that the airborne sampling provides averages sufficiently representative of the monthly means to be able to characterize and test model skill at representing geographical gradients in climatological mean plume structure. The wide range of varying aerosol-cloud vertical structures sampled are sufficient for addressing all of the originally-postulated objectives."

3. The discussion in the paragraph starting in line 820 raises the question of whether any future planned satellite lidar instruments will have the signal-to-noise ratio to provide a better view of the frequency of clear air between aerosol and cloud layers? Can the results of ORACLES help constrain a future mission so that past inferences about the relationships between aerosol and cloud layers can be improved upon with large statistics from satellites in the future?

Response: The ORACLES results can indeed be used to study the information content and capabilities of future satellite missions. A manuscript led by co-author F. Xu that studies joint lidar+polarimeter retrieval capabilities on the basis of ORACLES RSP and

HSRL-2 observations in the context of the NASA observable study ACCP (Aerosols, Clouds, Convection and Precipitation) is close to submission and will be added to the reference list if it is submitted before this manuscript needs to be finalized. The question on required SNR to unambiguously detect the clear-air layers in the SE Atlantic depends on very specific instrument characteristics. For now, we have added the following text to the manuscript at the end of section 5.2.1: "ORACLES data provide a useful testbed for algorithm development in support of future satellite missions, for example NASA's ACCP (Aerosols, Clouds, Convection and Precipitation) mission. For instance, ORACLES observations are currently being used to develop joint polarimeter+lidar retrievals of aerosol and cloud properties. Whether such observations will successfully detect features such as the clear-air layers in the SE Atlantic will depend on specific instrument characteristics, but the ORACLES measurements should provide useful benchmarks for the testing of candidate observing concepts."

4. The discussion between lines 861 and 865 is confusing and seems to be missing some key elements to the interpretation of the observed SSA and BC:OA ratio. From figure 14, Aug. and Sept. appear similar with Oct. being the outlier, but the discussion draws a distinction, in terms of 4 km winds, between Aug. and the other months. Furthermore, I was not clear on exactly what the higher winds in Sept./Oct. are responsible for. Finally, if I am understanding the argument correctly, the aerosol lower in the column is older and also exhibits lower SSA and higher BC:OA ratio. However, the discussion does not link these aerosol properties to age? Am I to conclude that aging depletes the organic fraction of the aerosol, and that is why the SSA decreases with altitude?

Response: The discussion surrounding Fig. 14 in lines 840-865 has been modified to more clearly reflect what is shown in the figure and the inferences we make from it, with the rewritten paragraphs included here: "The lidar-derived increase in extinction with height for September (2016) in Fig. 13 is accompanied by a similar increase with height of the mean in-situ SSA (derived from the in-situ PSAP absorption paired with

nephelometer scattering at 530 nm) from 0.84 to 0.87 (Fig. 14, middle of top row). The SSA values are in reasonable agreement with values of ∼0.83 for dry conditions in Davies et al. (2019) derived using state-of-the-art photoacoustic and cavity ring-down instrumentation. Pistone et al., (2019) compare the ORACLES absorption+scattering measurements with SSA derived by several different airborne remote-sensing methods at wavelengths between 400 and 995nm and found reasonable agreement both for specific case studies and for the range of measured spectral SSA over the full ORACLES-2016 deployment. Black carbon is the primary absorber of sunlight within BB aerosol (e.g., Bond et al., 2013). Although a corresponding decrease with height of the refractory black carbon (BC) mass concentration relative to the mean organic aerosol (OA) mass concentration is not clearly apparent in Fig. 14, (middle of middle row), an example from an individual profile from the September 24, 2016 flight indicates more nitrate and organic aerosol above 3.5km than below it, relative to the black carbon mass concentration (Fig. 14, middle of bottom row). This is consistent with an increase in SSA with height. Examples of individual profiles are shown from each year, broken down by aerosol species [black carbon, organic aerosol, nitrate (NO3), ammonium (NH4) and sulfates (SO4)], indicate distinct vertical structures (Fig. 14 bottom row). Profiles from August 2017 also indicate some vertical structure to the SSA without a clear mapping to the BC/OA ratio, while the mean SSA from the October, 2018 deployment increases even more sharply with altitude than does the mean SSA from September (2016). For October, 2018, the increase of SSA with height is clearly consistent with the proportional increase in organic aerosol relative to black carbon. Work is ongoing to attribute changes in SSA to both thermodynamically-driven changes in gas-particle phase partitioning (e.g., Wu et al., 2020) and more irreversibly driven changes related to forms of photodegradation in this near-equatorial, sun-exposed environment. Overall the SSA values are less than has been previously assumed, with that implication further explored within the modeling study of Mallet et al., (2020).

Some of the differences between the individual profiles in Fig. 14, bottom row, can be related to differences in the prevailing meteorology shown in Fig. 8. The distinctive two-layer aerosol structure profiled on 24 September, 2016 reflects the ability of strong winds at 4 km (see also Fig. 6) to disperse aerosol westward, with the aerosol lower down, at 2-3 km, resulting from an anticyclonic circulation (Fig. 8). The strong 4km zonal winds are much less apparent in August, consistent with a lower-lying, less layered aerosol vertical structure. The free-tropospheric winds remain strong into October, but by then the fire emissions have reduced considerably and less aerosol appears to reach the altitude at which the zonal winds are strongest. "

5. The results presented in figure 18 are intriguing and certainly demand further study. I understand that there is a paper in progress to do just that (Gupta et al. 2020). However, it is rather disappointing to read that there are hypotheses that might explain the results, but not be able to read what they are. Is it not possible to share the hypotheses and then note that they are to be evaluated in the other paper?

Response: This point is well taken; after considerable discussion among co-authors and mild disagreement on the hypotheses that best explain the findings, we would prefer to eliminate any mention of specific hypotheses and instead refer the reader to Gupta et al., 2020. Consequently, we have made the following changes to the paper:

a) We have changed the sentence "...it experiences significant increase with ZN for the Contact/Na>250 cm-3 case, possibly for reasons hypothesized by Gupta et al. (2020)." from lines 950-951 of the original manuscript with "...it experiences a significant and unexpected increase with ZN for the Contact/Na>250 cm-3 case."

b) We have replaced ""These findings, and hypotheses for the processes responsible, are further described by Gupta et al. (2020)." with "These findings are further described by Gupta et al. (2020)." on lines 957-958 of the original manuscript;

c) For the bullet point in the conclusion section (lines 1179-1180) we replaced "there was ample evidence for aerosol-induced modifications of Sc cloud properties (Gupta et al., in prep.), and some evidence for the suppression of drizzle (Dzambo et al., 2019);" with "there was ample evidence for aerosol-induced modifications of Sc

cloud properties (Fig. 18) and those of mid-level clouds (Adebiyi et al., 2020); a new cloud/rain dataset produced by a joint multi-wavelength cloud radar and multi-angle/multi-wavelength polarimeter will allow for further investigation for the suppression of drizzle by aerosol (Dzambo et al. 2019, 2020);"

6. Conversely, there are a number of declarative statements in the bulleted list in section 7 of the paper that are not supported in the paper and the supporting citations are to papers that are not yet published. While I understand that the notion here is to summarize some of the findings from the campaign as they presently stand, this does raise a question of whether it would be proper to have this paper on the record citing some declarative results that may either not end up in the peer-reviewed literature, or be altered somewhat after peer-review.

Response: See our response to point 18 raised by reviewer #1 - We removed the bullet point with the conclusion about the kappa parameter (which was referred to with the numerical value of 0.2) and eliminated all references to papers that have not yet been submitted. Fortunately, except for the bullet point with the kappa parameters, all other conclusions are supported by figures in this paper; we added references to sections 4.2, 4.3, 5.2.2 and 5.2.3 and Figure 18, where necessary.
* * *